**High peatland methane emissions following permafrost thaw: enhanced acetoclastic**
**methanogenesis during early successional stages**
Liam Heffernan[1,2*]★, Maria A. Cavaco[3*]★, Maya P. Bhatia[3], Cristian Estop-Aragonés[4],
Klaus-Holger Knorr[4], David Olefeldt[1]
[1] Department of Renewable Resources, University of Alberta, Edmonton, AB T6G 2H1,
Canada. [2] Evolutionary Biology Centre, Department of Ecology and Genetics/Limnology,
Uppsala University, Norbyvägen 18D, 752 36, Uppsala, Sweden. [3] Department of Earth and
Atmospheric Sciences, University of Alberta, Edmonton, AB T6G 2H1, Canada. [4] Institute of
Landscape Ecology, Ecohydrology and Biogeochemistry Group, University of Münster,
Münster, Germany
[*]Corresponding authors: Liam Heffernan (liam.heffernan@ebc.uu.se) and Maria A. Cavaco
(cavaco@ualberta.ca)
★ These authors contributed equally to this work

**Abstract**

Permafrost thaw in northern peatlands often leads to increased methane ($CH_4$) emissions, but the underlying controls responsible for increased emissions and the duration for which they persist have yet to be fully elucidated. We assessed how shifting environmental conditions affect microbial communities, and the magnitude and stable isotopic signature ($\delta^{13}C$) of $CH_4$ emissions along a thermokarst bog transect in boreal western Canada. Thermokarst bogs develop following permafrost thaw when dry, elevated peat plateaus collapse and become saturated and dominated by *Sphagnum* mosses. We differentiated between a young and a mature thermokarst bog stage (~30 and ~200 years since thaw, respectively). The young bog located along the thermokarst edge, was wetter, warmer and dominated by hydrophilic vegetation compared to the mature bog. Using high throughput 16S rRNA gene sequencing, we show that microbial communities were distinct near the surface and converged with depth, but lesser differences remained down to the lowest depth (160 cm). Microbial community analysis and $\delta^{13}C$ data from $CH_4$ surface emissions and dissolved gas depth profiles show that hydrogenotrophic methanogenesis was the dominant pathway at both sites. However, mean $\delta^{13}C$-$CH_4$ signatures of both dissolved gases profiles and surface $CH_4$ emissions were found to be isotopically heavier in the young bog (-63 ‰ and -65 ‰, respectively) compared to the mature bog (-69 ‰ and -75 ‰, respectively), suggesting that acetoclastic methanogenesis was relatively more enhanced throughout the young bog peat profile. Furthermore, mean young bog $CH_4$ emissions of 82 mg $CH_4$ m$^{-2}$ day$^{-1}$, were ~ three times greater than the 32 mg $CH_4$ m$^{-2}$ day$^{-1}$, observed in the mature bog. Our study suggests that interactions between the methanogenic community, hydrophilic vegetation, warmer temperatures, and saturated surface conditions enhance $CH_4$ emissions in young thermokarst bogs, but that these favorable conditions only persist for the initial decades after permafrost thaw.

**Keywords**
Permafrost, peatland, thermokarst, 16S RNA, isotope, methanogenesis, microbial
community, methane emissions

## 1. Introduction

Methane ($CH_4$) emissions in northern peatlands are typically thought to be driven by
environmental and ecological conditions such as temperature, water table position, and
vegetation community (Bellisario et al., 1999). However, $CH_4$ emissions are ultimately the
result of microbial activity and understanding the interactions between environmental
conditions and microbial processes is key to understanding the impact of disturbances on
peatland $CH_4$ emissions. Increased disturbances such as permafrost thaw are transforming
northern latitude peatlands (Helbig, Pappas & Sonnentag, 2016), through the disruption of the
frozen landscape and environmental conditions responsible for the regional accumulation of
large peatland carbon (C) stores. Rapidly rising northern air temperatures (Mudryk et al.,
2018) are predicted to lead to widespread gradual thawing of permafrost (Schaefer et al.,
2011) and subsequent thermokarst development in high C density permafrost peatlands
(Olefeldt et al., 2016). Thermokarst formation in ice-rich permafrost peatlands is
characterized by ground subsidence and surface inundation (Camill, 1999). This exposes
previously frozen C to anaerobic microbial decomposition and potential mineralization into
greenhouse gases (Schuur et al., 2015). Redox conditions following thermokarst formation
are an important control of decomposition, with 3 – 4 times greater C mineralization
occurring as aerobic respiration compared to anaerobic respiration (Schädel et al., 2016).
Increased emissions of methane ($CH_4$) due to thermokarst formation are projected to result in
a positive feedback with climate warming (Turetsky et al., 2020). However, the magnitude of
peatland $CH_4$ emissions and the metabolic pathways responsible for these emissions in
response to permafrost thaw remain uncertain, as does the period for which these conditions
and emissions persist.

Methanogenesis, conducted by methanogenic archaea belonging to phylum

Euryarchaeota, is one of the most prominent microbial processes contributing to the
anaerobic decomposition of organic matter in water-logged permafrost soils (Cai et al., 2016;
Knoblauch et al., 2018). Methanogenesis occurs primarily via two pathways: acetoclastic
methanogenesis and hydrogenotrophic methanogenesis (Whiticar et al., 1986; Whiticar,
1999). Acetoclastic methanogenesis involves the cleavage of acetate into $CH_4$ and $CO_2$ and
when considering these two species, causes less apparent fractionation than the
hydrogenotrophic methanogenesis pathway. This results in acetoclastic methanogenesis
yielding comparatively isotopically heavy $\delta^{13}C$-$CH_4$ ($\delta^{13}C$ = -65 to -50‰). The reduction of
$CO_2$ and $H_2$ in hydrogenotrophic methanogenesis typically produces $CH_4$ lighter in $^{13}C$ ($\delta^{13}C$
= -110 to -60‰) (Hornibrook et al., 1997, 2000). While the two pathways are
stoichiometrically equal (Conrad, 1999; Corbett et al., 2013), the activity of acetoclastic and
hydrogenotrophic methanogens are governed by different extrinsic controls (Bridgham et al.,

2013).

Hydrogenotrophic methanogenesis is thought to be the main pathway of $CH_4$

formation in northern peatlands (Hornibrook et al., 1997; Galand et al., 2005). However, the
acetoclastic pathway can dominate in the upper layers of more minerotrophic, nutrient rich
peatlands (Popp et al., 1999; Chasar et al., 2000) where there are sufficient levels of acetate
(Ye et al., 2012). During the initial decades following thaw, surface runoff of nutrients from
surrounding intact peat plateaus (Keuper et al., 2012; 2017) and increased connectivity to
regional hydrology (Connon et al., 2014), can result in more minerotrophic conditions. Such
shifts in hydrology, temperature, nutrients, redox conditions, and vegetation communities
following permafrost thaw have been shown to increase the prevalence of acetoclastic
methanogenesis and $CH_4$ emissions (Hodgkins et al., 2014; McCalley et al., 2014). However,
this potential post-thaw enhancement of acetoclastic methanogenesis needs to be considered
in context of the existing methanogenic community that developed in the peat profile before
thaw. For example, historical environmental conditions have been shown to have a legacy
effect on the methanogenic community following thaw and can therefore be a key constraint
on methanogenic community structure and activity post-thaw (Holm et al., 2020; Lee et al.,
2012). Overall, an understanding of the methanogenic community's response following thaw
to shifts in both surface conditions and exposure to previously frozen organic matter is key to
estimating $CH_4$ emissions from thermokarst peatlands.

Environmental conditions following permafrost thaw in peatlands are characterized

by a drastic shift in water table position and increased wetness, increased soil temperatures,
and a change in vegetation community associated with increased labile inputs (Beilman,
2001; Burd et al., 2020; Camill, 1999). These shifts may provide optimal conditions for $CH_4$
production and emissions, particularly in the initial decades following thaw. Peatland $CH_4$
emissions are constrained by water table position (Huang et al., 2021; Strack et al., 2004),
and surface inundation leads to increased $CH_4$ emissions (Tuittila et al., 2000). Methane
production and emissions are positively influenced by soil temperatures (Hopple et al., 2020;
Olefeldt et al., 2017), and peatland $CH_4$ emissions have been shown to increase when both
water table position and temperatures are high (Grant, 2015). The colonization of vegetation
associated with fresh, labile inputs has also been shown to increase both the magnitude and
temperature sensitivity of $CH_4$ emissions in peatlands (Leroy et al., 2017; McNicol et al.,
2019). As such, many studies have focussed on the relationship between water table position,
soil temperature and vegetation communities in determining $CH_4$ fluxes following thaw
(Johnston et al., 2014; Turetsky et al., 2007; Wickland et al., 2006). However, while these
environmental conditions are key drivers of $CH_4$ emissions, they are unable to fully account
for the variability in permafrost peatland $CH_4$ emissions (Juottonen et al., 2021; Kuhn et al.,
2021). Some of this unaccounted variance may be in part explained by microbial activity, as
changes in the composition and abundance of methanogenic community members can
contribute significantly towards peatland $CH_4$ emissions (Fritze et al., 2021). Relatively few
studies have assessed how shifts in environmental conditions and ensuing changes in
methanogenic community structure influences $CH_4$ emissions following thaw (McCalley et
al., 2014), an interaction that may be significant both at the local and circumpolar scale.

In this study we assess the impact of permafrost thaw on peatland methanogenic

community composition and $CH_4$ emissions along a space-for-time thaw gradient that
includes an intact peat plateau and an adjacent thermokarst bog with areas that have thawed
~30 and ~200 years ago (herein referred to as young bog and mature bog, respectively).
Thermokarst formation has resulted in distinct environmental conditions at each stage along
this thaw gradient. We herein define these distinct environmental conditions as water table
position and surface wetness, soil temperatures, and vegetation community. Along this
gradient we assessed methanogenic community structure down to 160 cm. We hypothesize
that: (1) shifting environmental conditions along the permafrost thaw gradient results in a
successional microbial community and a restructuring of the methanogenic community, and
(2) the warmer conditions and hydrophilic vegetation community in the young bog, along
with the exposure of previously frozen peat, will result in a greater relative abundance of
acetoclastic methanogens throughout the depth profile, and subsequently greater overall $CH_4$
emissions. In the young bog and mature bog, we measured the concentration and $\delta^{13}$C-
signature of dissolved $CH_4$ and $CO_2$ down to 245 cm, and the rates and $\delta^{13}$C-signature of both
$CH_4$ and $CO_2$ land-atmosphere fluxes. The combined approach of measuring dissolved gas
depth profiles and surface emissions, in tandem with assessing the structure of the
methanogenic community along a depth profile, allows us to determine how changing
environmental conditions following thaw impacts methanogenic pathways and community
composition. Utilizing this approach, we can subsequently gain further insight into how long
elevated surface $CH_4$ emissions may persist post-thaw. Furthermore, this approach highlights
that while environmental conditions are important in determining $CH_4$ emissions, microbial
community composition, and changes in the methanogenic community structure are likely to
significantly influence $CH_4$ emissions following thaw.

## 2. Methods


### 2.1 Study Site and Design


The Lutose peatland study site (59.5°N, 117.2°W; Figure 1) is located on the Interior
Plains of western Canada, within the zone of discontinuous permafrost (Brown et al., 1997;
Heginbottom et al., 1995). The climate is continental with a monthly average summer high
temperature of 16.1 °C (July), winter low of -22.8 °C (January), and annual average air
temperature of -1.8 °C (Climate-Data.org, 2019 – data from site located ~50 km south of
Lutose). Annual average precipitation is 391 mm, of which three quarters fall as rain between
May and September. In the discontinuous permafrost zone of the Interior Plains in boreal
western Canada, ~40% of the landscape is covered by permafrost peatlands that have
between 2 and 6 m deep peat deposits (Gibson et al., 2018; Vitt et al., 2000). The peatland
complexes in this area are a fine-scale mosaic of permafrost peat plateaus, and permafrost-
free ponds, fens, and bogs (Zoltai, 1993; Bauer et al., 2003; Vitt et al., 2000; Pelletier et al.,
2017), and they are similar to those found in the Hudson Bay Lowlands (Kuhry, 2008) and
Alaska (Jones et al., 2017). The Lutose peatland complex is representative of the peatlands
found in the discontinuous permafrost zone of the Interior Plains in western Canada
(Heffernan et al., 2020). The site has 5 – 6 m deep peat and has transitioned through multiple
developmental stages since it began accumulating organic matter ~8,800 years ago. It
transitioned from a marsh, through a fen and a bog stage prior to permafrost aggradation
~1,800 years ago (Heffernan et al., 2020). Peatlands in the Interior Plains in western Canada
are one of the three largest stores of organic carbon found in peatlands within the permafrost
zone, the other two being the Hudson Bay Lowlands and the West Siberian Lowlands
(Hugelius et al., 2020; Olefeldt et al., 2021). Within the sporadic and discontinuous
permafrost zone of our study region >15% of the total peat plateau area has thawed and
formed thermokarst bogs in the last 30 years (Baltzer et al., 2014; Gibson et al., 2018).
Projections for this area suggests total permafrost lost from plateaus by 2050 (Chasmer and
Hopkins, 2017).

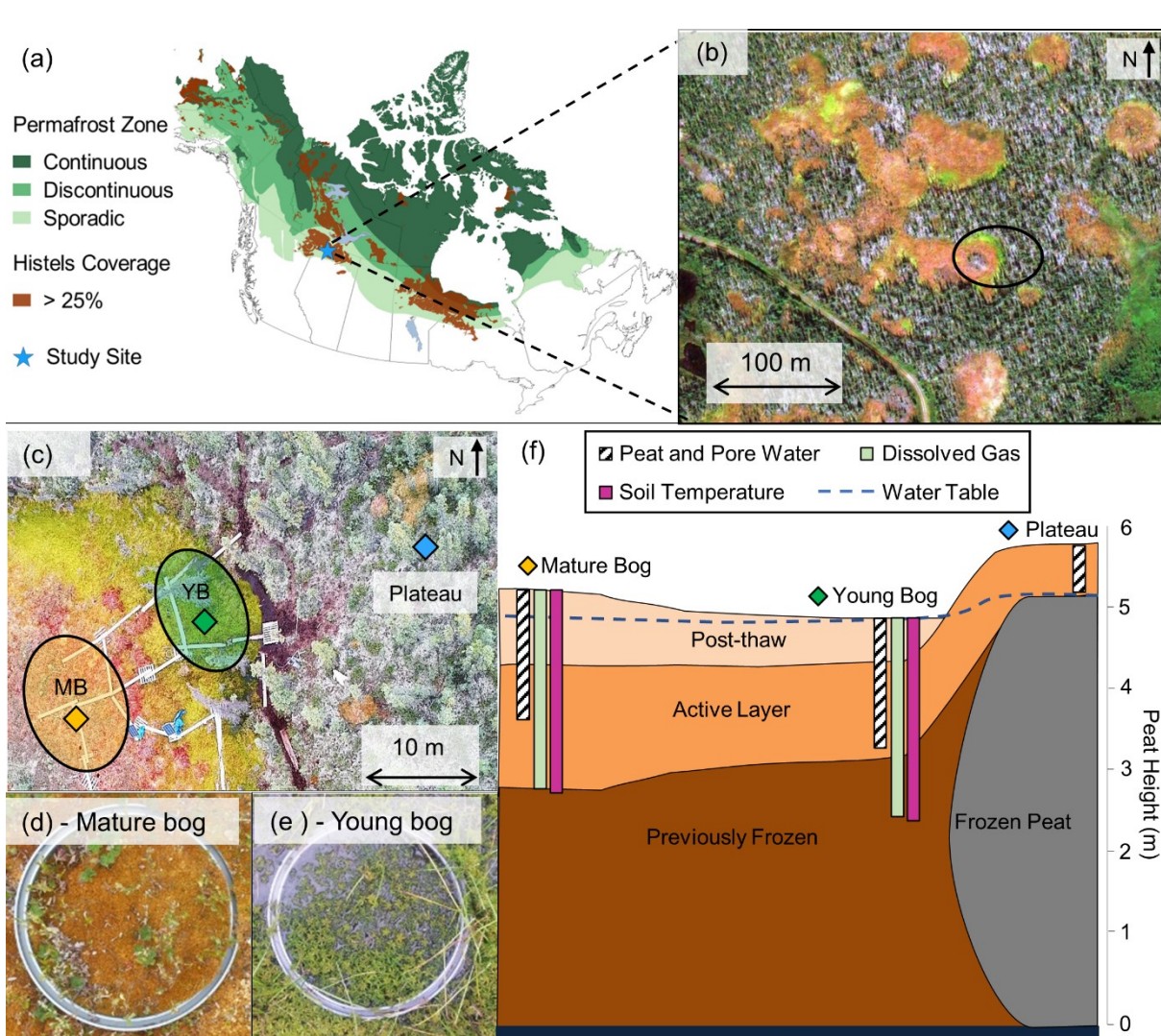


**Figure 1.** Lutose peatland site location and study design. (a) Site location (Lutose, Alberta, Canada 59.5°N, 117.2°W) in boreal western Canada. Green shading represents permafrost zonation (Brown et al., 1997) and brown shading represents areas with >25% permafrost peatland (histels) extent (Hugelius et al., 2014). (b) Geoeye satellite image of study site (image from https://zoom.earth/), 0.46 m resolution. Circle represents the area where sampling took place. (c) Aerial image of study transect, locations of peat and dissolved gas sampling in the plateau (blue diamond), young bog (green diamond), and mature bog (orange diamond), and area where collars for gas flux measurements were located in the young bog (YB, green) and mature bog (MB, orange) (Aerial photo credit: Olefeldt, David). (d, e) Surface vegetation in the mature bog and young bog (f) Soil profile of thaw transect based on (Heffernan et al., 2020). The transition to Post-thaw peat occurs at 29 cm and 71 cm in the young bog and mature bog respectively. Peat (core) and pore water (pore water peepers), including microbial community, sampling depth profile 0 – 160 cm shown as white column with diagonal black lines. Dissolved gas (diffusive samplers) sampling depth profile 0 – 245 cm shown as light green column. Soil temperature depth profile 0 – 250 cm shown as purple column. Average water table depth shown as dashed blue line.

The studied transect represents a space-for-time gradient of permafrost thaw that includes three thaw stages: a permafrost peat plateau, and a young (~30 years since thaw) and mature (~200 years since thaw) part of an adjacent thermokarst bog. The timing of permafrost thaw was previously determined by $^{14}$C dating the shift in macrofossil vegetation indicative of thaw, at 29 cm in the young bog and at 71 cm in the mature bog (Figure 1f) (Heffernan et al., 2020). The peat plateau has an active layer thickness of ~70 cm and its surface is raised 1 – 2 m above the adjacent thermokarst bog due to the presence of excess ground ice, resulting in relatively dry surface conditions where the water table generally follows the deepening of the seasonally thawed peat layer (Zoltai, 1972). This thaw stage is characterized by a stunted, open black spruce (*Picea mariana*) canopy and ground cover of lichens (*Cladonia* spp.), *Sphagnum fuscum* hummocks, and low-lying ericaceous shrubs as is characteristic of the peat plateaus in the area (Vitt et al., 1994). The young bog stage is narrow (<5 – 10 m wide) and is located next to the actively thawing area of the peat plateau. The young bog has an average growing season water table position of 1.3 ± 4.9 cm below the peat surface. These inundated conditions result in the dominance of a hydrophilic vegetation community (Figure 1e) consisting of *Sphagnum riparium*, bog-sedge (*Carex limosa*), and rannoch rush (*Scheuchzeria*

*palustris*). The mature bog is ~10 – 15 m from the young bog and is      drier, compared to the
young bog, with an average growing season water table position of 22.9 ± 9.3 cm below the
surface. The dominant vegetation reflects these drier conditions and consists of *Sphagnum*
*fuscum, Sphagnum magellanicum,* leather leaf (*Chamaedaphne calyculata),* cloudberry
*(Rubus chamaemorus), Eriophorum vaginatum* tussocks, and some black spruce (*Picea*
*mariana)* regrowth (Figure 1d). The mature bog is located >10 – 20 m from the thawing
plateau edge.
*2.2  Site Preparation and Monitoring of Environmental Conditions*
The Lutose peatland study site was established in 2015 and a boardwalk was constructed
to minimize disturbances along the peat plateau - thermokarst bog transect. Three collars for
surface greenhouse gas flux      (39 cm diameter) measurements were permanently installed to
a depth of 20 cm in both the young and mature thermokarst bog stages. The top of each collar
was aligned with the peat surface. PVC wells (2 cm diameter) were installed directly next to
each collar and were used to manually monitor the water table position during each gas flux
measurement. We monitored soil temperature (°C) at 10, 30, 50, 75, 100, 150, 200, and 250
cm every 30 min from May – September 2018 using permanently installed loggers (Hobo 8k
Pendant Onset Computer, Bourne, MA, USA) in both thermokarst bog stages. Temperature
depth profiles were established centrally among collars in each thermokarst bog stage, in
areas that had similar vegetation, water table position, and distance from the thawing edge as
the collars.
Custom made plexiglass pore water suction (Heffernan et al., 2021) and diffusive
equilibration gas sampling devices (Knorr et al., 2009) were installed in July 2016 in the
young and mature bog. These devices were installed in both thermokarst bog stages ~1 m
from the nearest flux measurement collar. Pore water suction devices were installed to a
depth of 160 cm and consisted of 15 sampling depths, with each sampling depth connected to
the surface via silicone tubing. This allowed for repeated non-destructive pore water
sampling. Three diffusive gas sampling devices were installed in each thermokarst bog stage,
where two collected dissolved soil gas samples from 5 – 95 cm deep and a third from 115 –
245 cm. Each diffusive gas sampler consisted of a PVC pipe with a 10 cm long sampling
section centred at each sampling depth. Sampling sections consisted of ~2 m of silicon tubing
(3 mm i.d., 5 mm o.d.) wrapped around the PVC pipe and kept in place by PVC-spacers at
the top and bottom of each interval. Silicone tubes were sealed at one end whereas the other
end was connected to polyurethane tubing (1.8 mm i.d.) that ran back up inside the PVC tube
to reach the peat surface where it was sealed with a three-way stopcock. Silicone tubing has
been shown to be permeable to gases such as $CO_2$ and $CH_4$ within a number of hours, while
remaining impermeable to water, making it suitable for sampling of dissolved soil gases
(Kammann et al., 2001).
*2.3  Pore water chemistry and peat enzyme activity*
Pore water dissolved organic matter (DOM) chemistry and peat enzyme activity
presented in this study have previously been published (Heffernan et al., 2021), and are
briefly described here. Pore water samples for DOM chemistry were taken monthly from
May – September 2018 using the previously described pore water suction devices in the
young bog and mature bog. Three 60 mL samples were taken from all 15 measurement
depths by applying a vacuum at the surface and collecting water with syringes via a three-
way stopcock. Each water sample was immediately filtered through 0.7 μm pore size glass
fiber filters (GF/F Whatman) into two acid-washed amber glass bottles, with one sample
acidified with 0.6 mL 2N HCl to prevent further microbial activity. Pore water samples were
transported in a cooled container and stored at 4 °C prior to analysis. Pore water DOM was
analyzed for pH, phosphate ($PO_4^{3-}$; µg $L^{-1}$), dissolved organic carbon (DOC; mg $L^{-1}$), total
dissolved nitrogen (TDN; mg $L^{-1}$) concentrations, phenolic contents, specific UV absorbance
at 254 nm (SUVA, L mg $C^{-1}$ $m^{-1}$; Weishaar et al., 2003) and spectral slope between 250 – 465
nm ($S_{250-465}$, $nm^{-1}$; Helms et al., 2008). SUVA and $S_{250-465}$ values are used to indicate
aromaticity, with high SUVA indicating a high aromatic content and lower $S_{250-465}$
indicating low molecular weight and decreasing aromaticity (Hansen et al., 2016).
Peat cores extracted to a depth of 160 cm were stored at 4 °C for less than one week in the
laboratory before homogenization to determine potential soil enzyme activities. We
performed hydrolytic enzyme assays for four enzymes; phosphatase, β-N-glucosaminidase, β-
glucosidase, and β-cellobiosidase using fluorogenic 4-methylumbelliferone labelled
substrates (Dunn et al., 2014). We assayed oxidative enzyme activity by measuring laccase
activity using syringaldazine (Criquet et al., 2000; Jassey et al., 2012). We summarized the
activity of all enzymes using a multi-functionality index based on *z*-scores (Allan et al., 2015;
Heffernan et al., 2021).
*2.4 Surface Land-Atmosphere Gas Fluxes*
We measured surface land-atmosphere greenhouse gas fluxes ($CH_4$ and carbon dioxide;
$CO_2$) monthly from May – September 2018 at the 3 collars in each peatland stage using the
static chamber method (Carroll & Crill, 1997). The chamber used to capture land-atmosphere
fluxes was a transparent cylindrical Plexiglass chamber with a basal area of 0.12 $m^2$, height
of 0.40 m, and volume of 47.8 L. The chamber was equipped with three fans (Micronel
Ventilator D341T012GK-2, BEDEK GmbH, Dinkelsbühl, Germany) to mix air during
measurements and a temperature sensor (Hobo RH Smart Sensor, S-THB-M002, Onset
computers, Bourne, USA) that was shaded from direct sunlight (Burger et al., 2016). An
airtight seal was formed between the chamber and collar by pouring water in a ~1.5 cm deep
well around the upper circumference of each collar. Land-atmosphere fluxes of $CO_2$
(ecosystem respiration) and $CH_4$ were captured simultaneously in darkened conditions by
covering the chamber with a reflective shroud. Gas concentrations were determined at a
temporal resolution of 1 s using an Ultraportable Greenhouse Gas Analyser (Los Gatos
Research, CA, USA) and real-time fluxes were monitored using the VNV® Viewer
(RealVNC® Limited, UK) application with an iPad mini 2 (Apple Inc.).

The rates of $CH_4$ and $CO_2$ land-atmosphere fluxes (*Flux*) were calculated using the ideal

gas law following:
$$Flux = slope\,\frac{P.V}{R.T.A} \qquad (1)$$
where slope is the linear rate of change of gas concentration ($\mu mol\ mol^{-1}\ second^{-1}$) over the
measurement period inside the chamber; P is an atmospheric pressure (atm) constant of 0.96
atm; V is chamber volume (L); R is the universal gas constant ($L\ atm\ K^{-1}\ mol^{-1}$); T is the
average temperature (K) inside the chamber during the measurement; and A is the chamber
basal area ($m^2$). Chamber closure for each flux measurement was 5 minutes with the first 2
minutes discarded to ensure fluxes (i.e., change in concentration over time) with $R^2 > 0.75$.
We report $CO_2$ fluxes in g $CO_2$ $m^{-2}$ $day^{-1}$ and $CH_4$ fluxes in mg $CH_4$ $m^{-2}$ $day^{-1}$, with positive
values indicating fluxes to the atmosphere. To quantify the proportion of C being emitted as
$CH_4$, we standardized our $CO_2$ and $CH_4$ fluxes per g C emitted. The proportion of C emitted
as $CH_4$ ($CH_4$:C emissions) was calculated as
$$CH_4{:}\,C\ emissions = \frac{CH_4\ m^{-2}\ day^{-1}}{CH_4\ m^{-2}\ day^{-1} + CO_2\ m^{-2}\ day^{-1}} \qquad (2)$$
*2.5  $\delta^{13}C$-signature of $CH_4$ emissions*

We assessed the $\delta^{13}C$-$CO_2$ and $\delta^{13}C$-$CH_4$ signatures of ecosystem respiration ($CO_2$) and

$CH_4$ emissions. This was done similarly to regular measurements of $CO_2$ and $CH_4$ fluxes, but
using a smaller, opaque chamber of 31.1 L and discrete syringe-samples for $\delta^{13}C$ analysis in
combination with the continuous monitoring of gas concentrations described above. Gas
syringe samples were taken using a 20 mL syringe via a three-way stopcock placed between
the sealed chamber and gas inlet port on the Ultraportable Greenhouse Gas Analyser. Gas
samples were then injected into a 37.5 mL sealed glass-vial that had been flushed with
nitrogen gas prior to sealing. Chamber enclosure time ranged from 30 – 50 minutes with 4 – 5
samples being taken during this time. Samples were taken either every 10-minutes or once a
minimum change in $CO_2$ (30 μmol mol$^{-1}$) and $CH_4$ (1 μmol mol$^{-1}$) concentrations was
observed. An atmospheric gas sample was used as a time-zero measurement when assessing
the change in concentration over time. Glass-vials containing samples were stored at 4 °C
until analysis. These measurements were taken in September and October 2016 from 1 collar
in both the young and mature bog, with each collar measured twice.

We measured the $\delta^{13}C$ values of gas samples from both the chamber fluxes and

atmospheric background. To assess whether the gas concentration of each sample fit within
the measurement range required for $\delta^{13}C$ analysis we measured $CO_2$ and $CH_4$ concentrations
using 1 – 3 mL from each vial. Following these concentration measurements, the remaining
sample (17 – 19 ml) was diluted with nitrogen gas to a final volume of 20 mL and injected
into a Small Sample Introduction Module (SSIM, Picarro, California, USA) system to
measure $\delta^{13}C$ signatures. The $\delta^{13}C$-$CO_2$ and $\delta^{13}C$-$CH_4$ signature was measured in-line with a
cavity ring-down spectrometer (G2201-L, Picarro, California, USA) that had been calibrated
using certified standards.

We then used the time-series of $\delta^{13}C$-$CH_4$ and $CH_4$ concentrations to estimate the $\delta^{13}C$-

$CH_4$ signature of the $CH_4$ released to the atmosphere using Keeling plots (Keeling, 1958).
Using this approach, the $\delta^{13}C$-$CH_4$ signature of gas in each sample is plotted on the *y*-axis
against the inverse of $CH_4$ gas concentrations (1/[$CH_4$]). The *y*-axis intercept of the linear
regression represents the mean isotopic signature of the $CH_4$ source (Fisher et al., 2017).
While fractionation during diffusive transport may influence these estimates, it has been
shown in similar systems to be of minor importance compared to other contributing processes
(Preuss et al., 2013; Nielsen et al., 2019).
*2.6  Dissolved gas depth profiles*

Dissolved gas samples were collected using  diffusive equilibration gas sampling

devices. Samples were taken from the following 15 depths: every 10 cm down to 95 cm
starting at 5 – 15 cm, and then at 115 cm, 140 cm, 165 cm, 195 cm, and 245 cm. Once a
month from May – September 2018 a ~7 mL gas sample was drawn from each depth using a
10 mL plastic syringe. These gas samples were immediately injected into a 10 mL sealed
glass-vial that had been flushed with nitrogen gas prior to sealing, and then were stored at 4
°C until analysis. A total of 214 $CO_2$ and 211 $CH_4$ dissolved gas concentration measurements
were made by injecting 1 – 3 mL of gas into a gas chromatograph with an FID and $CO_2$
methanizer (8610C Gas Chromatograph, SRI Instruments, California, USA). We measured
$\delta^{13}C\text{-}CO_2$ and $\delta^{13}C\text{-}CH_4$ signatures using the previously mentioned cavity ringdown
spectrometer and SSIM system. As with surface chamber gas samples, dissolved gas samples
were diluted with $N_2$ to 20 ml. However, dissolved gas concentrations were considerably
higher than gas concentrations found in the surface chambers, and some were well above the
optimal concentration range required for accurate $\delta^{13}C$ analysis for the SSIM system even
after dilution. To fit within measurement range of the system, further dilution resulted in $CO_2$
concentrations below detectable limits. As such, we were able to obtain 90 and 75
measurements of $\delta^{13}C\text{-}CH_4$ in the young and mature bog, respectively, and 93 measurements
of $\delta^{13}C\text{-}CO_2$ in both.

We used the $\delta^{13}C\text{-}CO_2$ and $\delta^{13}C\text{-}CH_4$ signature of each gas sample to calculate the

apparent fraction factor $\alpha_c$, where $\alpha_c = [^{13}C\text{-}CO_2 + 1000]/[\,^{13}C\text{-}CH_4 + 1000]$. The $\alpha_c$ can serve
as an isotopic indicator of the pathway of methanogenesis, with typical values of 1.060 –
1.090 observed for hydrogenotrophic methanogenesis and 1.040 – 1.060 for acetoclastic
methanogenesis (Chanton et al., 2005).

*2.7 Peat and pore water sample collection for microbial community composition*

*analyses*

Microbial community composition was characterized in both peat and peat pore water
samples from depths between 0 – 160 cm in the young bog and mature bog. Focusing on peat
samples, microbial community composition in the active layer of the peat plateau was
assessed from depths between 0 – 30 cm. Peat cores were extracted in June and September
2018. Near-surface cores were extracted using a cutting tool to 30 cm deep in the peat plateau
and young bog, and 50 cm deep in the mature bog. Surface cores were limited to 30 cm in the
plateau due to the presence of ground ice during sampling in June. Surface core depths
differed between the young bog and mature bog due to differences in the water table position.
Deeper core sections (down to 160 cm) in the young bog and mature bog were extracted
using a Russian peat corer (4.5 cm inner-diameter, Eijkelkamp, Giesbeek, The Netherlands).
Cores were extracted from two boreholes located ~20 cm apart, alternating between
boreholes to avoid disturbance contamination from the 10 cm corer tip during the coring
process. To do so, 50 cm long core sections were taken alternatively from each borehole, with
each core having a 10 cm overlap with the previous core taken from the adjacent borehole. In
the field, immediately after the entire core was extracted, cores were divided into 15
subsections. The first two subsections contained peat from 0 – 5 cm and 5 – 10 cm, followed
by 10 cm increments down to 120 cm, and two further subsections from 130 – 140 cm and
150 – 160 cm. Peat from each interval was sub-sampled using sterilized forceps and placed
directly into Whirl-Pak® bags, and frozen within 3 hours of sampling for transportation back
to the laboratory. Once samples reached the laboratory, they were frozen at -80 °C until
analysis.
We also sampled peat pore water at all 15 peat sampling depths in September 2018 from
the pre-installed pore water suction sampling devices mentioned above. We extracted 60 mL
pore water samples by applying a vacuum at the surface and collecting water with new plastic
60 mL syringes. Pore water was immediately filtered through sterile 0.2 µM pore size
Polyvinylidene difluoride (PVDF) membrane sterivex filters (MilliporeSigma). Microbial
cells were retained on the filter, and remaining porewater in the sterivex was removed via
extrusion using a 60 mL sterile syringe. Sterivex filters were then immediately flash-frozen at
-80 °C in a liquid nitrogen dry-shipper to preserve microbial community members until
analysis could take place.
*2.8 DNA extraction*
Genomic DNA was extracted from all peat and pore water samples using the DNeasy
PowerSoil kit (Qiagen) and the PowerWater DNeasy kit (Qiagen), respectively, to assess the
differences in microbial community structure. Extraction of DNA from both sample types
was followed as described by the manufacturer (Qiagen), with two modifications: (i) for peat
samples, prior to mechanical lysis using bead beating, the prepared samples were chemically
lysed by incubation at 70 °C for 10 minutes in the provided lysis solution, and (ii) sterivex
(pore water) samples were incubated with rotation at 37 °C following addition of lysis buffer.
These modifications were made to increase total DNA yield. The amount of isolated DNA
from each sample was then determined using a Qubit fluorometer (model 2.0, using the 1×HS
dsDNA kit), with concentrations ranging between ~0.1 and 22.4 ng µL$^{-1}$. This extracted DNA
served as the template for polymerase chain reaction (PCR) analyses described below.
*2.9 Sequencing and computational analyses*
We amplified 16S rRNA genes using universal prokaryotic primers 515F (Parada,
Needham & Fuhrman, 2016) and 926R (Quince et al., 2011). Each primer also contained a
six-base index sequence for sample multiplexing (Bartram et al., 2011). The PCR mix (25μL
total volume) contained $1 \times$ Q5 reaction buffer, 0.5 μM forward primer, 0.5 μM reverse
primer, 200 μM dNTPs, 0.500 U Q5 polymerase (New England Biolabs, Ipswich,M.A,
U.S.A) and 2.5 μL of genomic template. Genomic extracts with DNA concentrations of
greater than 2 ng μL$^{-1}$ were diluted 1:100 in nuclease-free water. The PCR was performed as
follows: 95 °C for 3 minutes, 35 cycles of 95 °C for 30 seconds, 60 °C for 30 seconds, 70 °C
for 1 minute and a final extension of 70 °C for 10 minutes. Pooled 16S rRNA gene amplicons
were purified using Nucleomag beads and a 4.5 pM library containing 50% PhiX Control v3
(Illumina, Canada Inc., NB, Canada) was sequenced on a MiSeq instrument (Illumina Inc.,
CA, USA) using a $2 \times 250$ cycle MiSeq Reagent Kit v3 (Illumina Canada Inc) at the
Molecular Biology Service Unit (MBSU, University of Alberta). The MiSeq reads were
demultiplexed using MiSeq Reporter software version 2.5.0.5. Each read pair was assembled
using the paired-end assembler for Illumina sequences (PANDAseq; Masella, Bartram &
Truszkowski, 2012) with a quality threshold of 0.9, dictating that 90% of overlapping reverse
and forward reads must match in order to assemble reads into read pairs. Assembled reads
were analyzed using the Quantitative Insights Into Microbial Ecology II pipeline (QIIME2;
Boylen et al., 2020). Sequences were clustered into amplicon sequence variants (ASVs) with
chimeric sequences, singletons and low abundance ASVs removed using DADA2 (Callahan
et al., 2019). All representative sequences were classified with the Greengenes reference
database, using the most recent release (version 13.8; McDonald et al., 2012). Although
Greengenes is not updated as frequently as the SILVA database, we chose to use it to classify
our ASVs as a comparison of both databases revealed that they captured a similar number of
archaea (total of 51187 methanogenic read counts attributed to SILVA *versus* 51141
methanogenic read counts attributed to Greengenes). The taxonomic resolution between both
databases was also similar, identifying the same kinds of phyla, families and genus, and
methanogens (e.g., Methanoregula, Methanosarcinales, etc.). Given these similarities, and the
fact that methanogen nomenclature has not changed significantly over time, we ultimately
chose to use Greengenes because it was able to resolve more methanogenic families
belonging to Methanocelalles and Methanomassiliicoccaceae particularly, compared to
SILVA. The Greengenes database is also still commonly used to explore methanogenic
archaeal communities in current literature (Vanwonterghem et al., 2016, Lin et al., 2017,
Carson et al., 2019). Furthermore, since 1021 methanogenic reads were captured per sample,
on average, using Greengenes and are comparable to other studies (Vishnivetskaya et al.,
2018; Holm, et al., 2020) we believe that our approach is sufficient for covering methanogen
diversity.
*2.10       Statistical analyses*
All statistical analyses were carried out in R (Version 3.4.4, R Core Team, 2015) using
the *nlme*, *vegan*, *factoextra*, *ggplot2*, *VariancePartition* and *ggpubr* packages (Pinheiro et al.,
2017; Oksanen et al., 2013; Kassambara & Mundt, 2017; Wickham, 2016; Hoffman &
Schadt, 2016; Kassambara, 2018). For Analysis of Variance (ANOVAs), distribution of the
data was inspected visually for normality along with the Shapiro-Wilk test. We tested
homogeneity of variances using the *car* package and Levene's test (Fox and Weisberg, 2011).
We report uncertainty as ± 1 standard deviation, except for land-atmosphere greenhouse gas
fluxes which we report as ± 95% confidence intervals. We here define the statistical
significance level at 5%.
We used ANOVAs and Bonferroni post-hoc tests on linear mixed effects models to
address our second hypothesis and to evaluate significant differences and seasonal trends in
greenhouse gas fluxes and dissolved gas depth profiles. We performed these tests to assess
whether thaw stage (young bog or mature bog) influenced greenhouse gas fluxes and
dissolved gas depth profiles. This approach was used to test for significant differences in $CH_4$
fluxes, ratio of $CH_4$:C emissions, and source $^{13}C$-$CH_4$ signature intercepts of Keeling plots
between young bog and mature bog stages. In each linear mixed effect model, sampling
month and peatland stage were defined as fixed effects whereas sampling collar was defined
as a random effect. Similarly, we tested for significant differences between the young and
mature bog depth profiles with respect to dissolved $CH_4$ and $CO_2$ concentrations, $\delta^{13}C$-$CH_4$
and $\delta^{13}C$-$CO_2$ values, $\alpha_c$ values, and pore water chemistry. In these models, sampling month
and peatland stage were defined as fixed effects while sample depth was defined as a random
effect.
Following microbial 16S rRNA gene sequencing, sample reads were rarefied to the
lowest read count of 28,129 for all subsequent analyses. These sequences represent whole
microbial community data that was used to determine whether there was evidence of changes
in microbial community structure representing the successional peatland stages following
permafrost thaw throughout the 160 cm depth peat profile. In addition, to address our first
hypothesis, we assessed differences in community composition across both peat and pore
water and to determine whether seasonality impacted microbial community structure in both
sample matrices. Here, Bray Curtis dissimilarity matrices for overall microbial community
data were used, at 999 permutations, to identify distinct groupings assessed at the 95%
confidence interval in NMDS ordinations. These distinct groupings were further evaluated for
significance using the non-parametric permutational analysis of variance (PERMANOVA)
test.
To further test our first hypothesis, methanogens were selected at the order level from
our whole community data using Greengenes-assigned taxonomy. Utilizing their assigned
taxonomy, the pathways through which identified methanogens conduct methanogenesis was
determined by comparing our findings with the literature (Berghuis et al., 2019; Stams et al,
2019; Kendall & Boone, 2006; Zhang et al., 2020). Focusing on the methanogenic
community allowed us to specifically assess how permafrost thaw affects the microbial
community responsible for $CH_4$ production and net $CH_4$ emissions following thaw. We
utilized our methanogenic community data to construct redundancy analyses (RDA) and
relative abundance bar plots. RDAs were conducted using a Hellinger-transformed
methanogenic community. Explanatory variables (i.e., dissolved concentrations of $CO_2$, $CH_4$,
DOC, temperature, enzymatic activity estimate, thaw stage, depth, and distance to water
table) were scaled about the mean. These explanatory variables had variance standardized,
were checked for collinearity (parameters with variance inflation value > 10 were removed)
and selected for significance using backward selection, set at 1,000 permutations. The
significance of the RDA model, and of each axis was tested using ANOVAs, set at 999
permutations. Variance partitioning analyses were conducted to assess the contribution of
significant environmental parameters (i.e., thaw stage and distance to water table) on the
structuring of the Hellinger-transformed methanogenic community. Distance from water table
reflects the distance (in cm) a certain sample is from the water table in different stages of
thaw (young bog and mature bog). Due to the smaller size of our methanogenic community
relative to the total community, and the lack of some data at certain depths, we combined
pore water and peat samples together for these analyses. Relative abundance, which measures
how common or rare a particular microorganism is relative to the entire microbial
community, of methanogenic orders related to acetoclastic or hydrogenotrophic
methanogenesis processes were plotted according to depth. Significant differences in
methanogenic community composition between depths were assessed using the non-
parametric Kruskall-Wallis test with a Benjamini-Hochberg correction for multiple
comparisons, after running a Wilcox rank sum test.

## 3. Results

*3.1 Site environmental conditions*
The young bog was wetter and warmer than the mature bog throughout the May –
September 2018 study period. In June, following snowmelt, the water table was at its highest
at $2.2 \pm 0.6$ cm above the surface in the young bog. The highest water table position in the
mature bog was $17.5 \pm 1.9$ cm below the peat surface and observed in July. The water table
dropped during the season and in September was $5.7 \pm 2.2$ cm and $27.3 \pm 1.2$ cm below the
peat surface, in the young bog and mature bog respectively. In the plateau, the seasonally
thawed layer gradually deepened during the growing season, with an active layer depth of
$79.5 \pm 13.7$ cm measured in September. The water table in the peat plateau followed the
deepening of the seasonally thawed layer.
Soil temperatures followed the seasonal climate but were dampened and had temporal
lags in deeper peat layers (Figure S1a). The highest young bog and mature bog soil
temperatures at 10 cm depth occurred in July, at 14.3 and 14.1 °C, respectively. At 100 cm
depth the maximum temperatures occurred in August and September, at 8.6 and 6.9 °C,
respectively for the young and mature bog. Soil temperatures at 250 cm were still rising at the
end of September, peaking at 4.1 and 3.2 °C in the young bog and mature, respectively. The
young bog was consistently warmer than the mature bog throughout the study by on average
$0.9 \pm 0.9$ °C, $1.8 \pm 1.0$ °C, and $0.5 \pm 0.4$ °C at 10 cm, 100 cm, and 250 cm depths,
respectively.
Across all depths and sampling occasions, average pH was higher (ANOVA: $F_{(1, 77)} =$
$35.2$, $P < 0.001$) in the young bog than in the mature bog at $4.1 \pm 0.2$ and $3.9 \pm 0.2$
respectively. In contrast, DOC at $69.2 \pm 18.4$ and $53.8 \pm 5.4$ mg C $L^{-1}$ (ANOVA: $F_{(1, 82)} =$
38.7, $P < 0.001$) and total dissolved nitrogen at $1.5 \pm 1.4$ and $0.9 \pm 0.1$ mg $L^{-1}$ (ANOVA: $F_{(1,}$
$_{82)} = 12.8$, $P < 0.01$) were higher in the mature bog than in the young bog, respectively.
Average SUVA values were higher (ANOVA: $F_{(1, 82)} = 103.5$, $P < 0.001$) in the young bog
($3.2 \pm 0.4$ L mg $C^{-1}$ $m^{-1}$) compared to the mature bog ($2.6 \pm 0.4$ L mg $C^{-1}$ $m^{-1}$), indicating
DOM with a greater aromatic content in the young bog. However, average spectral slope
($S_{250 – 465}$) values were also greater (ANOVA: $F_{(1, 81)} = 6.9$, $P < 0.05$) in the young bog (-
$0.016 \pm 0.002$ $nm^{-1}$) compared to the mature bog ($-0.017 \pm 0.003$ $nm^{-1}$), indicating lower
molecular weight and decreasing aromaticity. Average phenolics ($0.6 \pm 0.2$ and $0.6 \pm 0.2$ mg
$L^{-1}$) and phosphate ($PO_4^{3-}$: $9.0 \pm 14.3$ and $6.7 \pm 3.0$ µg $L^{-1}$) were similar between the young
bog and mature bog, respectively, across all depths and sampling occasions. Full details of
DOM chemistry results can be found in Heffernan et al., (2021). Of note is the fact that the
pore water chemistry was compared across all depths in this study, in contrast to Heffernan et
al., (2021) in which pore water found above and below the transition indicating permafrost
thaw was compared.
*3.2 Concentrations and isotopic signatures of dissolved gases*

Dissolved $CH_4$ increased with depth below the water table in both the young and

mature bog (Figure 2a). Dissolved $CH_4$ concentrations in the young bog increased with depth,
from19 µmol $L^{-1}$ at 5 cm depth, to a peak of 5,400 µmol $L^{-1}$ at 195 cm. Dissolved $CH_4$
concentrations in the mature bog remained low above the water table (<6 µmol $L^{-1}$ below 25
cm), but then increased to $4,100 \pm 1,700$ µmol $L^{-1}$ between 115 and 250 cm depth and peaked
at 6,800 µmol $L^{-1}$. Dissolved $CO_2$ concentrations followed a very similar pattern to $CH_4$,
increasing with depth in both the young and mature bog (Figure 2b). Again, the mature bog
had overall higher concentrations, with mean average values ranging from 340 – 1,295 µmol
L$^{-1}$ and peaking at 1,500 µmol L$^{-1}$ at 85 cm. In contrast, the young bog had average values
ranging from 113 – 960 µmol L$^{-1}$ and peaked at 1,200 µmol L$^{-1}$ at 95 cm (Figure 2b).

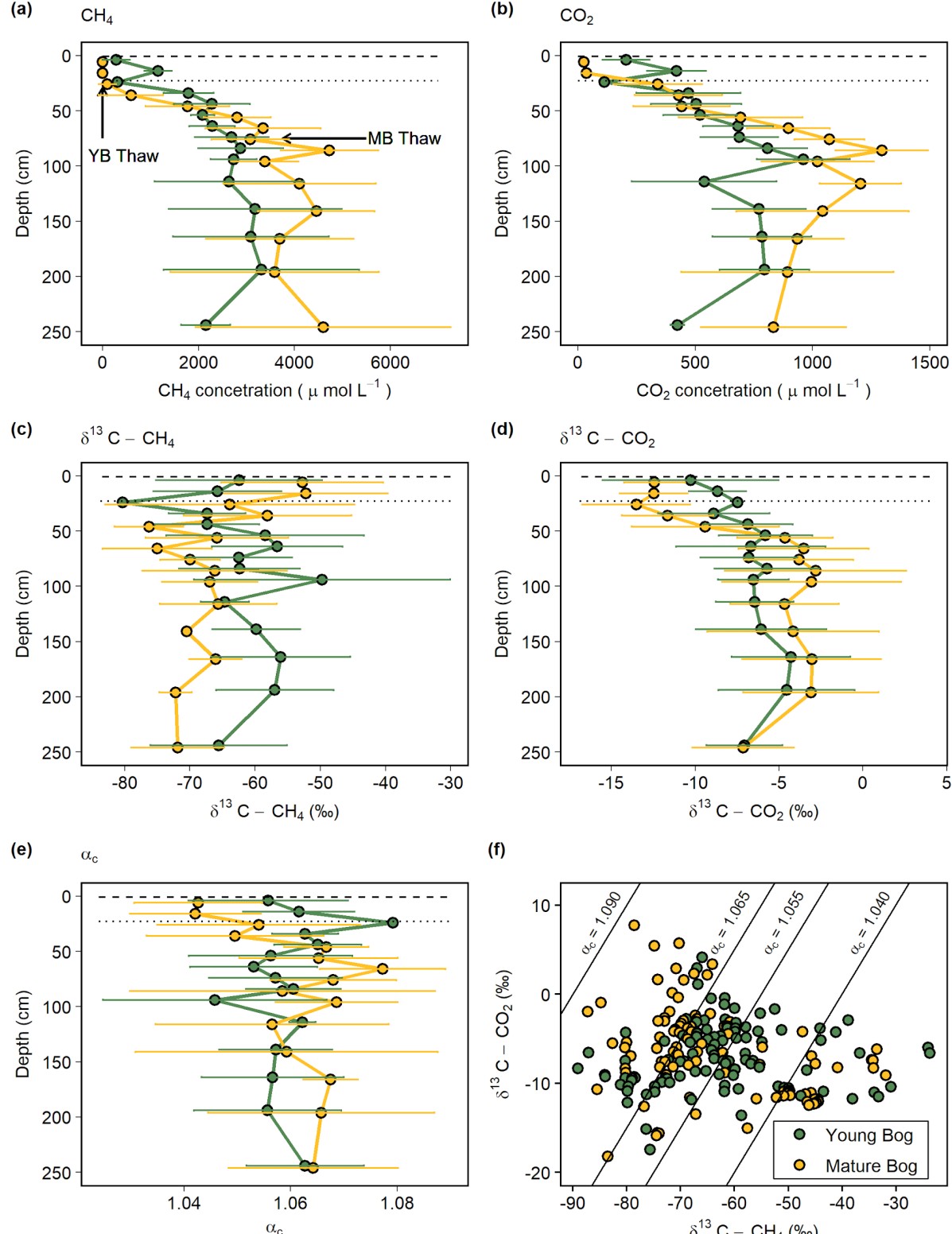


**Figure 2.** Average seasonal (May – September) depth profiles in the young (green, black
circles) and mature (yellow, black circles) bog of (a) dissolved $CH_4$ concentration ($\mu$mol $L^{-1}$),
(b) dissolved $CO_2$ concentration ($\mu$mol $L^{-1}$), (c) $\delta^{13}$C-$CH_4$ (‰), (d) $\delta^{13}$C-$CO_2$ (‰), and (e)
apparent fractionation factor ($\alpha_c$) between dissolved $CH_4$ and $CO_2$. (f) Cross-plot of
corresponding $\delta^{13}$C-$CH_4$ and $\delta^{13}$C-$CO_2$ values (‰) in the young bog and mature bog, from
raw data used in panels (c) and (d). Diagonal lines represent different $\alpha_c$ where $\alpha_c$ 1.040 –
1.065 represents acetoclastic methanogenesis, and $\alpha_c$ 1.055 – 1.09 represents
hydrogenotrophic methanogenesis (Whiticar, 1999). (a) – (e) Dashed and dotted horizontal
lines represent water table depth in the young (YB) and mature bog (MB) respectively.
Arrows in panel (a) represent depth of thaw transition in both the young (29 cm) and mature
bog (71 cm), i.e., the transition from deep peat (accumulated prior to thawing) and shallow
peat (accumulated post thawing).

The young bog and mature bog had distinct profiles of $\delta^{13}$C values for both $CH_4$ and $CO_2$
(Figure 2c, d). The young bog had no apparent trend with depth for both $\delta^{13}$C-$CH_4$ (ANOVA;
$F_{(14, 45)} = 1.75$, $P = 0.08$) and $\delta^{13}$C-$CO_2$ (ANOVA; $F_{(14, 46)} = 1.79$, $P = 0.07$), averaging -62.4
$\pm$ 7.0 ‰ and -6.8 $\pm$ 1.6 ‰, respectively (Figure 2c, d). In the mature bog we observed
significant depth trends for both $\delta^{13}$C-$CH_4$ (ANOVA: $F_{(14, 43)} = 3.19$, $P < 0.01$) and $\delta^{13}$C-
$CO_2$ (ANOVA: $F_{(14, 49)} = 6.22$, $P < 0.001$). These significant depth trends are due to
isotopically heavy $\delta^{13}$C-$CH_4$ and light $\delta^{13}$C-$CO_2$ above the water table, which suggests an
influence from $CH_4$ oxidation. When comparing $\delta^{13}$C depth profiles between the thermokarst
bogs we focused on those values taken from under the water table to avoid the effect of $CH_4$
oxidation observed above the water table in the mature bog. Under the water table, $\delta^{13}$C-$CH_4$
values in the mature bog were significantly lighter (ANOVA: $F_{(1, 64)} = 18.72$, $P < 0.001$)
compared to the young bog at an average of -68.7 $\pm$ 5.0 ‰ and -62.4 $\pm$ 7.0 ‰, respectively.
Conversely, the mature bog had isotopically heavier $\delta^{13}$C-$CO_2$ than the young bog below the
water table (ANOVA: $F_{(1, 71)} = 13.86$, $P < 0.001$).
The apparent fractionation factor ($\alpha_C$) is a robust parameter to characterize the relative
contribution of $CH_4$ production pathways, with values of 1.040 – 1.060 indicating
acetoclastic methanogenesis and 1.060 – 1.090 for hydrogenotrophic methanogenesis
(Chanton et al., 2005). Similar to the gas $\delta^{13}$C depth-profiles, we found no clear trend with
depth for $\alpha_C$ values in the young bog (ANOVA; $F_{(14, 44)} = 0.87$, $P = 0.59$) with an average of
$1.058 \pm 0.012$ and range of $1.018 - 1.079$ (Figure 2e). In the mature bog, we found a clear
depth trend in $\alpha_C$ values (ANOVA: $F_{(14, 43)} = 5.71$, $P < 0.001$). Similar to the $\delta^{13}C$ depth
profiles in the mature bog, this significant depth trend in $\alpha_C$ is due to the influence of $CH_4$
oxidation above the water table, with the lowest $\alpha_C$ values being those from samples collected
above the water table at 5, 15, and 25 cm. The average $\alpha_C$ beneath the water table in the
mature bog was $1.064 \pm 0.017$ and ranged from $1.015 - 1.094$. When comparing $\alpha_C$ values
from beneath the water table between the young and mature bog we found that $\alpha_C$ values were
significantly lower in the young bog (ANOVA: $F_{(1, 63)} = 30.8$, $P < 0.001$).
In the isotopic ratio cross-plot of $\delta^{13}C$-$CH_4$ and $\delta^{13}C$-$CO_2$ (Figure 2f), most of the young
bog had $\alpha_C$ values of between $1.055 - 1.065$ (29 in total), with a greater number of samples
(21) between $\alpha_C = 1.040 - 1.055$, compared to the mature bog (15). In contrast, a greater
proportion of the mature bog samples had $\alpha_C > 1.065$ (42 in the young bog and 52 in the
mature bog). There was no clear depth trend in the $\alpha_C$ values and no samples in this study had
$\alpha_C > 1.090$. Several samples (13) from the young bog and mature bog had $\alpha_C$ values of $<$
1.040, likely due $CH_4$ oxidation (Knorr et al., 2009).
*3.3 Magnitude and isotopic signature of land-atmosphere gas fluxes*
The young bog had almost three times greater average $CH_4$ fluxes than the mature bog
during the May – September study period, at $82.3 \pm 21.9$ mg $CH_4$ m$^{-2}$ day$^{-1}$ and $30.8 \pm 10.6$
mg $CH_4$ m$^{-2}$ day$^{-1}$, respectively (Figure 3a). Fluxes of $CH_4$ in the young bog were greatest
between June and August, ranging from $80.6 \pm 40.3$ mg $CH_4$ m$^{-2}$ day$^{-1}$ to $100.9 \pm 63.1$ mg
$CH_4$ m$^{-2}$ day$^{-1}$. The lowest young bog $CH_4$ fluxes were observed in September at $55.0 \pm 17.7$
mg $CH_4$ m$^{-2}$ day$^{-1}$ (Figure S3a). Mature bog $CH_4$ fluxes were greatest in September ($55.8 \pm$
$21.1$ mg $CH_4$ m$^{-2}$ day$^{-1}$) and lowest in May ($5.6 \pm 2.7$mg $CH_4$ m$^{-2}$ day$^{-1}$). Ecosystem
respiration ($CO_2$ emissions measured with dark chambers) was significantly lower in the
young bog than mature bog, with study period averages of $0.6 \pm 0.3$ and $1.9 \pm 0.3$ g $CO_2$ m$^{-2}$
day$^{-1}$, respectively (Figure S3). Maximum ecosystem respiration in the young bog occurred in
August (1.6 g $CO_2$ m$^{-2}$ day$^{-1}$) and was much lower during the other four months (monthly
averages of 0.2 to 0.4 g $CO_2$ m$^{-2}$ day$^{-1}$). Ecosystem respiration rates in the mature bog were
elevated from June to August (monthly averages between 2.1 and 2.6 g $CO_2$ m$^{-2}$ day$^{-1}$),
and decreased in September (0.8 g $CO_2$ m$^{-2}$ day$^{-1}$). The proportion of total C emissions (sum
of $CH_4$ and $CO_2$ emissions) released as $CH_4$ were an order of magnitude greater in the young
bog than mature bog stage, at 18 and 2% respectively. This was a result of both higher $CH_4$
emissions and lower ecosystem respiration (Figure S3) in the young bog. The $\delta^{13}$C-$CH_4$
signature of $CH_4$ emissions (intercept values from Keeling plots), in the young bog were
significantly greater than those observed in the mature bog (Figure 3c; ANOVA: $F_{(1, 4)} =$
20.67, $P < 0.05$). The average $\delta^{13}$C-$CH_4$ signature of $CH_4$ emissions in the young bog (n = 4)
was -66.5 $\pm$ 1.4‰ (95% CI) and 78.5 $\pm$ 5.6‰ (95% CI; Figure 3c) in the mature bog
emissions (n = 4).

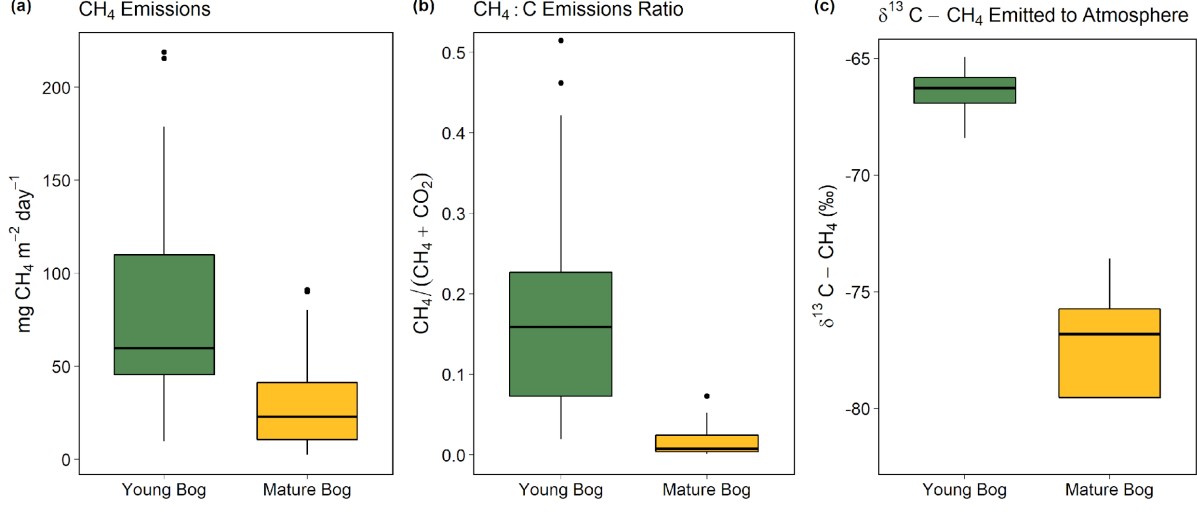


**Figure** 3. Magnitude and isotopic signature of greenhouse gas fluxes from the young bog
(green) and mature bog (yellow) shown as boxplots. Boxes represents the interquartile range
(25 – 75%), with median shown as black horizontal line. Whiskers extend to 1.5 times the
interquartile range (distance between first and third quartile) in each direction, with outlier
data plotted individually as black dots (a) The magnitude of net land-atmosphere $CH_4$
emissions as measured by soil chambers. (b) The ratio between $CH_4$ emissions and the sum of
$CO_2$ emissions (ecosystem respiration) and $CH_4$, both standardized to per g C. (c) Intercept
values of Keeling plots indicating the $\delta^{13}C$-$CH_4$ signature of $CH_4$ emissions. Isotopically
heavier (i.e., less negative) $\delta^{13}C$-$CH_4$ is produced via acetoclastic methanogenesis, whereas
isotopically lighter (i.e., more negative) $\delta^{13}C$-$CH_4$ is produced via hydrogenotrophic
methanogenesis,  The $CH_4$ and $CO_2$ land-atmosphere fluxes shown in (a) and (b) were
measured once a month from May – September 2018. The $\delta^{13}C$-$CH_4$ of $CH_4$ emitted to the
atmosphere was measured in September and October 2016 (see methods for details and
Figure S4 for Keeling plots).

*3.4 Microbial community structure along the permafrost peatland thaw gradient*
We used NMDS ordinations to assess differences in microbial community structure
between solid peat and pore water samples, between sampling depths, and between the
plateau, young bog, and mature bog. The only exception was the plateau, where only peat
samples were collected (i.e., no pore water samples). Microbial community structure in peat
was determined to be significantly different from porewater microbial communities
(PERMANOVA, $R^2 = 0.13$, $P < 0.05$, Figure 4). The differences observed in the microbial
community structure between peat and pore water samples could be a function of the
different extraction methods used to extract DNA (Carrigg et al., 2007). Among the pore
water samples, distinct microbial communities were found to be associated with the young
bog and mature bog. Similarly, microbial community structure in peat was found to be
significantly distinct between the three successional stages (plateau peat, young bog and
mature bog; Figure 4; PERMANOVA, $R^2 = 0.18$, $P < 0.05$). There is also a common trend in
vertical community structuring for all sample matrices according to depth. Changes in overall
microbial community composition in both peat and pore water, across a vertical profile (to a
maximum depth of 160 cm), illustrate a confluence in microbial community structure with
depth in both the young and mature bog (Figure 4). In other words, community structure was
most dissimilar at depths closer to the surface (Figure 4, Figure S2; PERMANOVA; $R^2 =$
0.16, $P < 0.05$). This trend was particularly evident in the porewater samples (Figure 4). In
the peat samples, though microbial communities did not fully converge, deeper young bog
peat (i.e., 90 – 160 cm) communities did become more similar to communities found in the
mature bog at intermediate depths (i.e., 30 – 70 cm), based on the nearness of sample points
on the NMDS (Figure 4). We also observed that the mature bog near-surface peat samples
were located closer to the plateau peat on the NMDS (Figure 4, PERMANOVA, $R^2 = 0.4$, $P =$
0.1). It was not possible to assess the presence of this cyclic succession (from young bog to
mature bog to plateau) in the pore water samples since we did not characterize the microbial
community in the plateau pore water. Finally, we also assessed the effect of seasonality on
microbial community structure and found no effect with regards to sampling month
(PERMANOVA; $R^2 = 0.02$, $P = 0.090$).

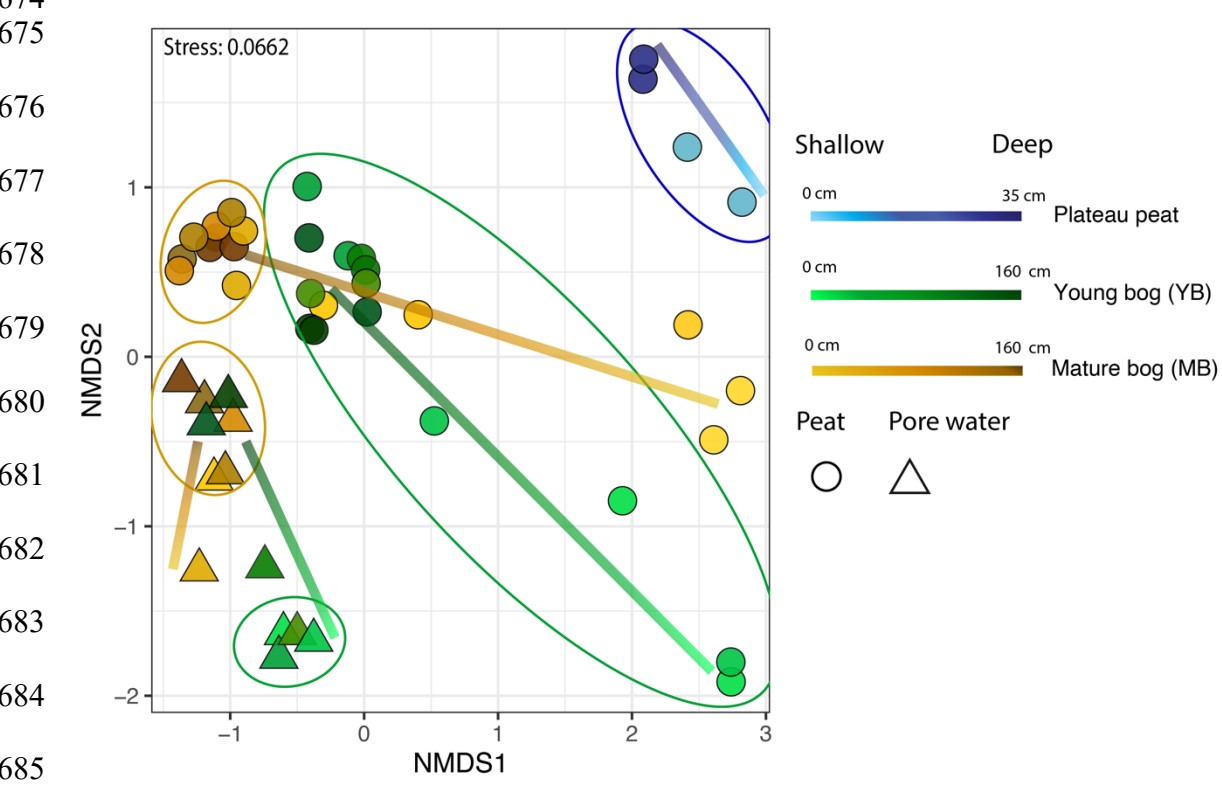

**Figure 4.** Microbial community distribution according to stage of peat/pore water. NMDS
ordinations of amplicon sequencing variant (ASV) data demonstrate significant community
dissimilarities according to thaw stage for both pore water (shown by the triangles) and peat
(shown by the circles) samples, encircled by 95% confidence intervals. Colour gradient and
lines demonstrate the shift in microbial community structure along vertical depth profiles
where lighter shades indicate samples closer to the surface.

The total archaeal community comprised 6% of the entire microbial dataset.

Methanogen-related orders comprised 54% of this archaeal dataset and demonstrated marked
differences in the relative abundance of acetoclastic-related methanogens according to thaw
stage and depth in both peat and pore water samples (Figure 5; Figure S2). In the young and
mature bog peat samples, hydrogenotrophic-related methanogens were ubiquitously present
throughout both depth profiles (Figure 5a). In comparison, acetoclastic-related methanogens
exhibited a relatively restricted presence, only present at specific depths (Figure 5a). These
communities were most abundant (>25% of the total methanogenic community) near the
surface in the young bog, just above and below the thaw transition zone (Figure 5a). In the
pore water, hydrogenotrophic methanogens were also dominant throughout depths in both
stages of thaw (Figure 5b). However, in contrast to peat samples, acetoclastic methanogens
were virtually absent in the pore water, although minimally present (i.e., $\leq 10\%$ relative
abundance) at depths between 35 and 155 cm, all found below the thaw transition zone
(Figure 5b).






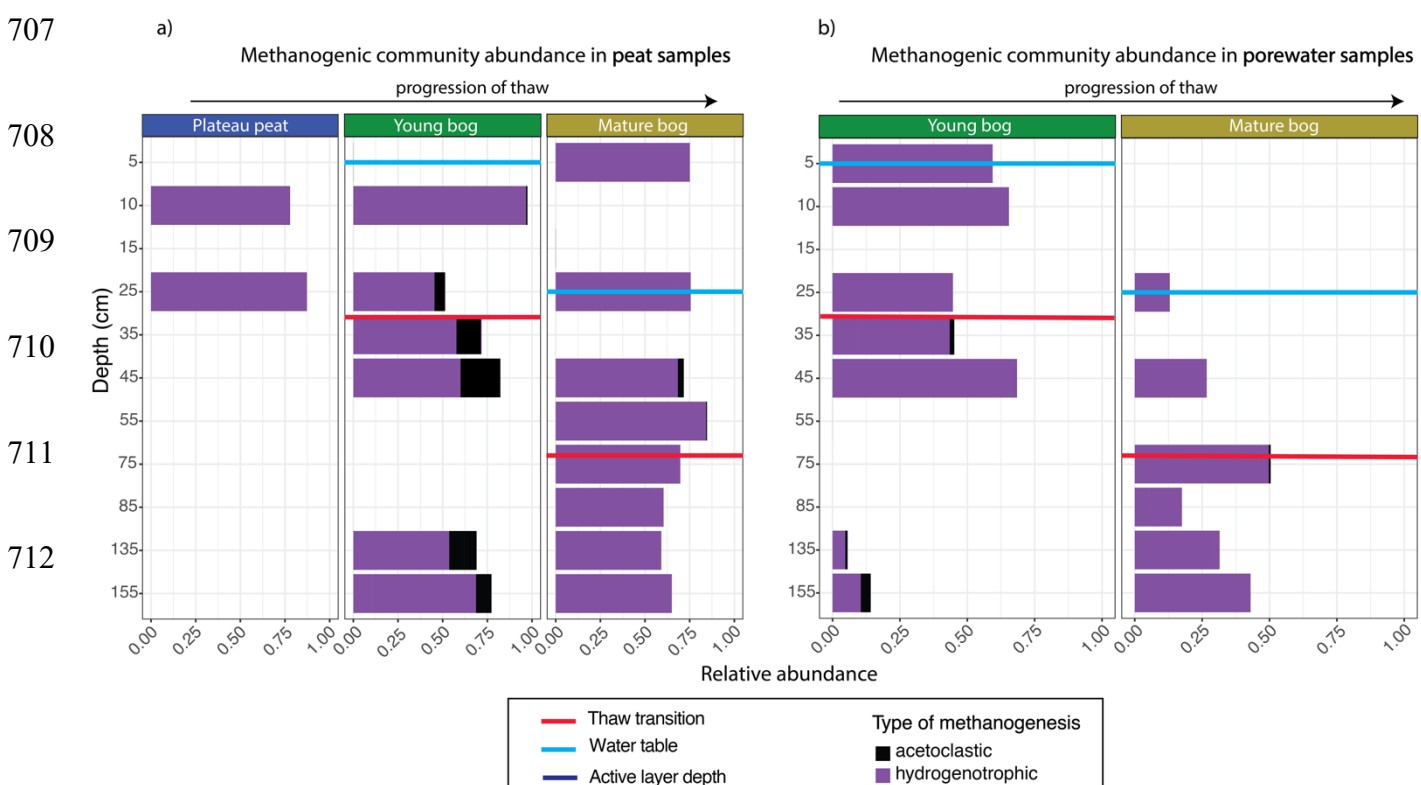




**Figure 5.** Relative abundance of archaeal orders according to putative methanogenic
capability, along a depth profile for peat and pore water samples. Samples are arranged
according to depth (y axis), with the relative abundance of methanogenic archaea resolved
shown on the x axis. Note that the y axis does not uniformly progress in 10 cm increments.
Progression of thaw is shown from plateau peat to young bog to mature bog at the top of the
figures, with position of water table shown in blue for each panel. Red lines demonstrate
thaw transition zone for the young bog and mature bog. (a) Stacked bar plot of methanogenic
Archaea for all peat samples. Samples demonstrate significant differences in putative
methanogen composition between all stages (Kruskall-Wallis test & Wilcox rank sum test,
with Benjamini-Hochberg corrected p-values, $P < 0.05$). (b) Stacked bar plot of
methanogenic Archaea for all pore water samples. Samples do not demonstrate significant
differences in putative methanogen composition between stages (Kruskall-Wallis test, with
Benjamini-Hochberg corrected p-values, $P = 0.965$).

Using a redundancy analysis (RDA, Figure 6) we found that 27.6% of variation in the

methanogenic community was explained by two variables: thaw stage (ANOVA, $P < 0.05$)

and depth from the water table (ANOVA, $P < 0.05$). Although these were the only two

parameters that were identified as significant variables impacting microbial community

structure when using a backward stepping model, it should be noted that there may be more

variation in the community that our experimental design does not take into account as a result

of unconstrained variation represented by plant-microbe and/or microbe-microbe interactions

(Boon et al., 2014). Nonetheless, the 27.6% variation explained is in accordance with other

studies conducted in permafrost impacted regions using similar methods, where the

percentage of explained variation falls between 6% (low) to 43% (high) (Comte et al., 2015;

Hough et al., 2020). Next, we used variance partitioning to assess the extent to which thaw

stage and depth from the water table (i.e., the significant environmental variables identified

by the RDA) explained the variation in only the methanogenic community structure (Figure

6). Based on this analysis, thaw stage explained 18.4% and distance to the water table

explained 4.3% of methanogenic community variation, respectively.






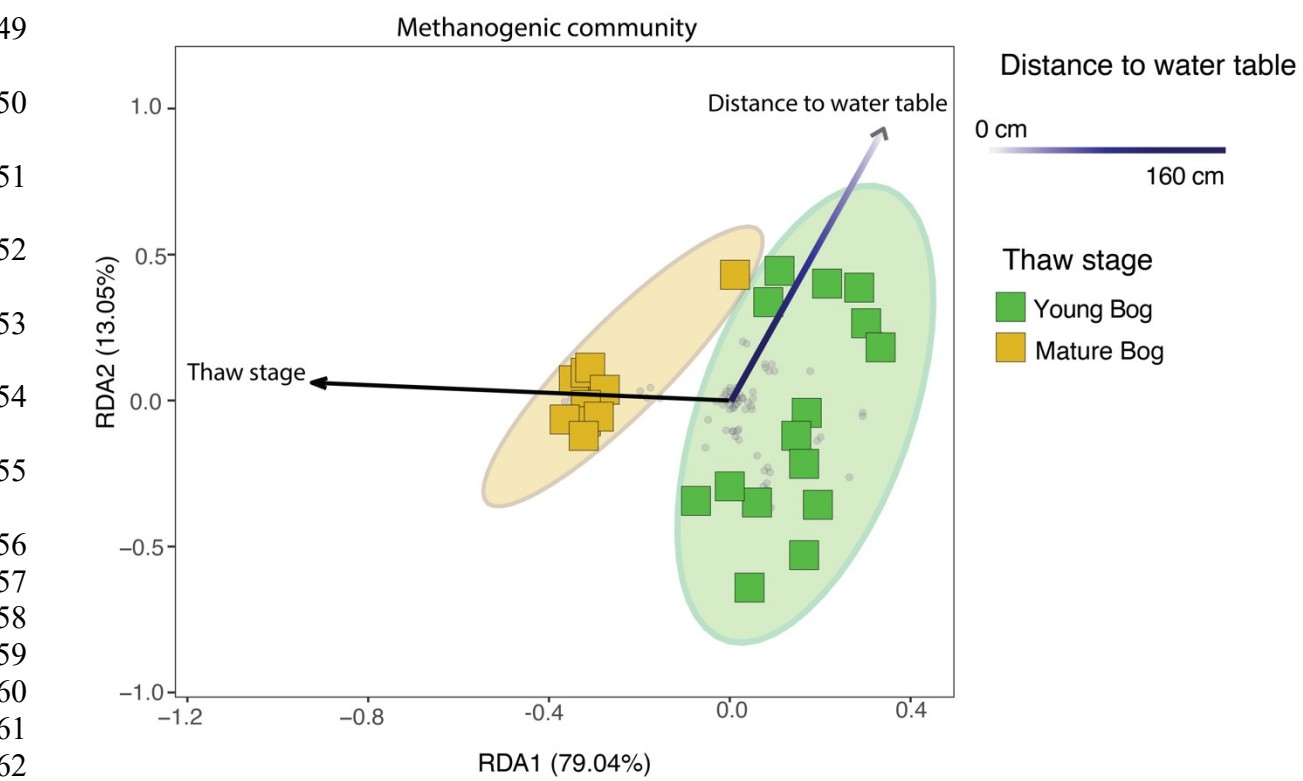

**Figure 6.** Redundancy analysis (RDA) exploring significant biotic and abiotic variables
influencing the total methanogenic community (adjusted $R^2 = 27.6\%$), as determined by a
backward stepping RDA model in the peat and pore water samples. All parameters that were
used in model are described in section 2.10 of the Methods. Grey dots in the panel
demonstrate the distribution of all ASVs in the methanogenic dataset. Shaded ellipses
represent the 95% confidence intervals for microbial community structure according to
peatland thaw stage (young bog vs mature bog). Only significant (ANOVA, $P < 0.05$)
variables are shown. Using variation partitioning, we found that peatland thaw stage
significantly explains about 18.4% of methanogenic community variation whereas distance to
water table explained 4.3%. Both axes are significant (ANOVA, $P < 0.05$).

## 4. Discussion

Our study shows that high $CH_4$ emissions from thermokarst bogs in the initial decades

following permafrost thaw (young bog) are not only linked to environmental conditions
(wetness, soil temperature, vegetation), but also driven by relatively increased microbial $CH_4$
production through the energetically more favourable acetoclastic methanogenesis pathway.
Evidence of putatively acetoclastic methanogens and $CH_4$ produced via the acetoclastic
metabolic pathway was found in the young bog both near the surface and at depths below the
thaw transition (i.e., in peat that accumulated prior to permafrost thaw). We are unable to
determine whether these greater $CH_4$ emissions in the initial decades following thaw are due
to the mineralization of labile organic matter released from previously frozen peat, or are
driven solely by fresh, labile DOM derived from surface vegetation leached throughout the
peat profile. However, previous work in the discontinuous permafrost region in the Interior
Plains of western Canada has found a limited contribution of previously frozen organic
matter contributing to surface $CH_4$ emissions in thermokarst bogs (Cooper et al., 2017).
Elevated $CH_4$ emissions then slow over the following centuries with succession into a mature
thermokarst bog stage where $CH_4$ production is almost exclusively through the
hydrogenotrophic pathway.
*4.1 Shift in microbial community assemblages along a permafrost thaw gradient*
Microbial communities varied along the permafrost thaw gradient; among different thaw
stages (permafrost peat plateau, young bog, and mature bog), with peat depth (surface down
to 160 cm), and between different sample types (solid peat and pore water). We found clear
differences in microbial communities between the young bog and mature bog, despite similar
peat stratigraphy up to the surficial vegetation (Heffernan et al., 2020), where dominant
*Sphagnum* species varied. The greater height of the peat surface above the water table and
drier conditions in the mature bog, due to the slow accumulation of new peat over centuries,
leads to a shift in vegetation composition from hydrophilic *Sphagnum* and graminoids
towards more drought resistant *Sphagnum* spp. and ericaceous shrubs. This shift in water
table position and vegetation community, along with a decrease in temperatures (Figure S1a)
due to the thermal insulating properties of *Sphagnum* peat (Kujala, Seppälä, & Holappa,
2008) appears to have caused the observed differences in microbial communities between the
young and mature bog, even at depths >1 m. Microbial communities were most dissimilar
between the peat plateau and young bog. This was unsurprising given the abrupt shift from
the elevated, frozen, and relatively dry peat plateau forest to the young bog where the surface
was saturated, dominated by hydrophilic vegetation and had warmer temperatures. We
further noted that the microbial community of the mature bog was more similar with the peat
plateau than with the young bog. Paleo-records in the region (Heffernan et al., 2020; Pelletier
et al., 2017; Zoltai, 1993) show that many peatlands have undergone cyclical permafrost
developments, as thermal insulating properties of *Sphagnum* peat in mature bogs leads to the
re-aggradation of permafrost peat plateaus. Our study suggests that the peat plateau microbial
community is influenced by the preceding mature bog microbial community as permafrost
aggrades.

The most dissimilar microbial community composition was observed between

samples near the surface and those at depth (i.e., down to 160 cm), as has also been
observed in other permafrost ecosystems (Frey et al., 2016; Monteux et al., 2018). Shifts in
microbial community composition along the thaw gradient were most evident nearer the
surface, whereas communities found at depth were similar between the young bog and mature
bog (Figure 4). At the surface, microbial community structure is influenced by the
successional vegetation community (Hodgkins et al., 2014) and the role that vegetation,
particularly graminoids which are found in the young bog, has on microbial community
structure has been well documented in northern peatlands (Robroek et al., 2015, 2021;
Bragazza et al., 2015). Moderately acidic, saturated peatlands with hydrophilic vegetation,
similar to the young bog, have been shown to harbour acid tolerant fermenting bacteria that
produce substrates for methanogenesis and are trophically linked with methanogens (Wüst et
al., 2009). Thus, the interaction between water table position, pH, and vegetation community
influences the substrates available to the microbial community, which in turn impacts the
surface community's structure (Kotiaho et al., 2013). In contrast, communities at depth are
known to be influenced by peat properties, such as peat chemistry and degree of
decomposition, and the paleoenvironment under which they originally colonized (Lee et al.,
2012; Holm et al., 2020). In the young and mature bog both peat properties (humification
indices including FTIR 1630/1090 $cm^{-3}$ and C:N ratios) and the paleoenvironment at depth
are similar (Heffernan et al., 2020), which may explain the observed convergence of
microbial community structure. Nonetheless, although there are some similarities at depth
between both young and mature bog, microbial communities inhabiting either are still distinct
(Figure 4). This is emphasized by the differing abundance of Archaea that participate in
hydrogenotrophic or acetoclastic methanogenesis (Figure 5) in both stages down the peat
profile.

As has been shown previously in other thermokarst peatlands (McCalley et al., 2014),

the young and mature bog stages were dominated by hydrogenotrophic methanogens.
However, putatively acetoclastic methanogens were relatively more abundant in the young
bog (Figure 5), particularly at or below the transition in peat that accumulated prior to
permafrost thaw. Thaw stage and distance from the water table were found to influence the
methanogenic community composition (Figure 6), with distance from the water table
dictating where anoxic conditions persist (Blodau et al., 2004) and thus where methanogenic
colonization can occur. The influence of vegetation communities associated with different
thermokarst peatland stages on methanogenic community composition has previously been
attributed to the role of plant derived DOM serving as the substrate for $CH_4$ production
(Liebner et al., 2015; McCalley et al., 2014). The presence of hydrophilic vegetation,
particularly graminoids, in the saturated young bog provides the precursors for fermentation,
yielding acetate (Liebner et al., 2015; Ström et al., 2003, 2012, 2015) and serving as the
substrate for acetoclastic $CH_4$ production. The downward transport from the surface of plant
derived DOM in the young bog (Chanton et al., 2008) likely provides sufficient acetate for
the establishment of acetoclastic methanogens at depth in this environment.
*4.2 . Production and emissions of $CH_4$ along a peatland thaw gradient*
Isotopic signatures ($\delta^{13}C$) of dissolved $CO_2$ and $CH_4$ and $\alpha_C$ values in porewater and
the of $\delta^{13}C$ signature of $CH_4$ emitted to the atmosphere provided further evidence of
relatively elevated acetoclastic methanogenesis in the young bog stage. The general increase
in $\delta^{13}C$-$CO_2$ with depth observed at both sites (Figure 2d) indicates accumulation of
isotopically heavier $\delta^{13}C$-$CO_2$ which is likely explained by the preferential use of isotopically
lighter $\delta^{13}C$-$CO_2$ during hydrogenotrophic methanogenesis (Hornibrook et al., 2000). As a
result, $CH_4$ tends to become lighter with depth and this was particularly apparent in the
mature bog (Figure 2c). This leads to the average $\alpha_C$ values of 1.064 ($\delta^{13}C$-$CH_4$; -68.7‰) in
the mature bog, which were significantly higher than the 1.058 ($\delta^{13}C$-$CH_4$; -62.4‰) observed
in the young. Together, the $\delta^{13}C$-$CH_4$ and $\delta^{13}C$-$CO_2$ data and the resulting $\alpha_C$ depth profiles
suggest that the majority of $CH_4$ is produced via the hydrogenotrophic methanogenic
pathway, which supports the findings of the microbial community analysis (Figure 5). Our
isotope data also suggests that a greater proportion of $CH_4$ is produced via acetoclastic
methanogenesis throughout the profile in the young bog compared to the mature bog (Figure
2c – f). This is evident from lower average $\alpha_C$ values found in the young bog compared to the
mature bog, and greater number of these young bog $\alpha_C$ values falling between 1.040 – 1.065
which represents acetoclastic methanogenesis (Whiticar, 1999). These findings again agree
with the relatively greater abundance of acetoclastic methanogens observed at that site
(Figure 5).
In this study we found that average $CH_4$ emissions in the initial decades following thaw,
in the young bog stage, were 2.5 – 3 times greater than emissions measured in the mature bog
stage which had thawed ~200 years ago (Figure 3a). Furthermore, the proportion of $CH_4$ to
overall C emissions (Figure 3b) was considerably greater in the young bog than in the mature
bog. In the mature bog the lower water table position leads to both increased $CO_2$ emissions
and decreased $CH_4$ emissions, resulting in a reduced fraction of C emissions as $CH_4$. Previous
studies have shown similarly increased $CH_4$ emissions in the initial decades following thaw
(Johnston et al., 2014; Wickland et al., 2006). While our pore water chemistry data is
inconclusive with regards to organic carbon characteristics, other work in thermokarst bogs in
the Interior Plains of western Canada has shown that the organic matter derived from the
young bog vegetation community is highly labile (Burd et al., 2020). Previous work at our
study site has shown that the vegetation community in the young bog is associated with
greater potential enzymatic degradation of organic matter (Heffernan et al., 2021). Hydrolysis
of plant derived organic matter by extracellular enzymes leads to the formation of monomers
(Kotsyurbenko, 2005). These monomers can be further degraded to form acetate and other
percussors for methanogenesis when present with anaerobic fermenting bacteria (Hamberger
et al., 2008) and near the surface and vegetation inputs (Hädrich et al., 2012). Our study
shows that these higher $CH_4$ emissions are likely linked to increased wetness, temperatures,
and a vegetation community associated with more labile organic matter which favour a
greater proportion of $CH_4$ produced via acetoclastic methanogenesis, as shown by our $\delta^{13}C$-
$CH_4$, $\alpha_c$ depth profiles and microbial community composition analyses.
Many factors, including environmental conditions and microbial community structure
likely contribute to the differences in net $CH_4$ emissions from the young and mature bog
(Figure 3a). Methane oxidation has been shown to be an important regulator of post-thaw
$CH_4$ emissions (Perryman et al., 2020) and to result in isotopically heavier (i.e., less negative)
$\delta^{13}C$-$CH_4$ and lighter (i.e., more negative) $\delta^{13}C$-$CO_2$ (Whiticar, 1999). Our data suggests the
role of $CH_4$ oxidation was different between sites. Methane oxidation was apparent in the
$\delta^{13}C$-$CH_4$ and $\delta^{13}C$-$CO_2$ signatures above the water table in the mature bog but no $CH_4$
oxidation is evident in the young bog (Figure 2c, d). The difference in gas flux $\delta^{13}C$
signatures (Figure 3c) also suggests a greater prevalence of $CH_4$ oxidation in the mature bog.
However, increased oxidation above the water table in the mature bog is likely not fully
responsible for the observed differences in $CH_4$ surface emissions and depth profiles between
the young and mature bog. Lower soil temperatures, a vegetation community associated with
reduced substrate availability, the dominance of hydrogenotrophic methanogenesis
throughout the peat profile, and a deeper water table position all contribute to the lower $CH_4$
production and higher $CH_4$ oxidation observed in the mature bog. Nonetheless, using this
interdisciplinary approach, we are unable to determine the relative contribution of
acetoclastic methanogenesis at each depth to the overall emissions at the surface.

Our results, and those of others (Euskirchen et al., 2014; Johnston et al., 2014), have

shown that $CH_4$ emissions exhibit seasonal variation (Figure S3a, c) . However, in contrast to
some previous findings (Ebrahimi & Or, 2017), we did not observe a corresponding seasonal
response in the microbial community composition (Figure S2). This may be a sampling
design effect since our study spanned only two months (June and September), compounded
by the fact that we did not have replicate samples to test the robustness of this finding.
However, other studies have also shown that soil microbial community growth is not
impacted by seasonal variations in temperature (Simon et al., 2020) and that microbial
communities require a longer time scale (years-decades-centuries) to respond to temperature
following thaw (Feng et al., 2020). Our results corroborate these observations, suggesting a
long-term response in the microbial community composition to the ecological shifts
associated with autogenic peatland succession following permafrost thaw. Autogenic
peatland succession following thaw occurs on the decade to century timescale, shifting from
recently thawed to mature thermokarst bogs (Camill, 1999). Both recently thawed (young)
and mature thermokarst bogs have distinct hydrological regimes, vegetation communities,
and peat chemistry. Following thaw, associated changes in vegetation and litter input alters
microbial community composition and activity (Adamczyk et al., 2020; Kirkwood et al.,
2021). Such changes in microbial community structure thus impact $CH_4$ emissions from
thermokarst peatlands. Under predicted climatic warming scenarios differences in microbial
community composition have been shown to be increasingly driven by seasonally
independent variables such as substrate quality and the legacy effects of soil temperatures
(Luláková et al., 2019). This study suggests that the environmental conditions required for
increased methanogenic activity at depth is limited to the initial decades following thaw, after
which the microbial community structure changes in response to lowering of the water table,
lower soil temperatures and shifts in the vegetation community.

## 5. Conclusion

This study demonstrates that higher $CH_4$ emissions in thermokarst bogs in the initial

decades following thaw are driven by shifts in vegetation communities that produce organic
matter inputs of varying lability (Burd et al., 2020) and prevalence of anoxic conditions,
which was associated with an increase of acetoclastic methanogenesis in our site. The
influence of this pathway was apparent at depth throughout the peat profile. With succession
following thaw towards a mature thermokarst bog, a shift in water table position and
vegetation composition seems to reduce the role of acetoclastic methanogenesis pathway.
Previous work at this site (Heffernan et al., 2021) and other thermokarst peatlands in the
discontinuous permafrost zone of boreal western Canada (Burd et al., 2020) have indicated
that the vegetation community found in the initial decades following permafrost thaw is
associated with increased potential enzymatic degradation and biodegradability of organic
matter compared to that found in the mature bog. Average growing season $CH_4$ emissions
were 2.5 – 3 times greater in the recently thawed young bog. Overall, C emissions in the
young bog contained proportionally more $CH_4$ than those from the mature bog, due to greater
$CH_4$ production and also reduced $CO_2$ emissions. These greater $CH_4$ emissions in the young
bog are driven by a higher contribution to surface emissions from $CH_4$ produced throughout
the peat profile by acetoclastic methanogens. The response of the microbial community to
permafrost thaw is tied to the shifting environmental conditions associated with peatland
autogenic succession. Warmer and wetter conditions in the initial decades following thaw, in
conjunction with a vegetation community associated with greater availability of labile plant
leachates (Bragazza et al., 2015), provides favourable conditions for acetoclastic
methanogens throughout the peat profile. Given the projected increases in thermokarst
peatland formation (Olefeldt et al., 2016), our study suggests that we can expect a pulse of
$CH_4$ emissions from current regions of the discontinuous permafrost zone. This pulse will be
driven, in part, by increased acetoclastic methanogenesis from labile substrates in recently
thawed thermokarst peatlands. However, this rapid increase in $CH_4$ emissions will only
remain at the decadal to century scale as autogenic peatland succession results in relatively
drier mature thermokarst bogs, where lower temperatures and less labile substrate availability
leads to a dominance of hydrogenotrophic methanogenesis.

**Data availability**
All biogeochemical and enzyme datasets generated and analyzed during this study are
available in the UAL Dataverse repository, [https://doi.org/10.5683/SP3/5TSH9V]. Microbial
sequences used in this study can be accessed from the NCBI database, using accession
number PRJNA660023.

## Author contributions

All authors contributed to the conception of the work. LH and CEA performed the field work
component. LH performed the biogeochemistry measurements. MAC performed the
microbial measurements. LH and MAC analyzed the data and wrote the manuscript draft. All
authors reviewed and edited the manuscript.

## Competing interests

The authors declare that they have no conflict of interest.

## Acknowledgements

The authors wish to thank McKenzie Kuhn, Maya Frederickson, Jördis Stührenberg, and
Trisha Elliot for assistance with field and lab work. We also thank Sophie Dang, at MBSU
for providing guidance throughout 16S rRNA gene library building and for subsequently
sequencing these libraries at the MBSU facility.

## Financial support

Funding and support were provided to D. Olefeldt and M. Bhatia by the Natural Science and
Engineering Research Council of Canada, Discovery grant (RGPIN-2016-04688 to DO and
RGPIN-2020-05975 to MB) and the Campus Alberta Innovates Program (CAIP).

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
