# Peer review of "High peatland methane emissions following permafrost thaw: enhanced acetoclastic"

_Biogeosciences, 2021_

## Referee Comment (RC1)

**Review of "High peatland methane emissions following permafrost thaw: enhanced acetoclastic methanogenesis during early successional stages" by Liam Heffernan and others.**

**Summary:**

The goal of this manuscript is to advance our understanding of the underlying controls of methane emissions from permafrost thaw in northern peatlands. Specifically, the authors assess how shifting ecological conditions (e.g., collapse of peat plateau and thermokarst bog formation) affect microbial communities, the amount of $CH_4$ released, and the $\delta^{13}C$ isotope composition of released $CH_4$. The authors also aim to determine how long elevated surface $CH_4$ emissions persist after thaw.

To answer these questions, the authors study peatland methanogenic community composition and methane emissions along a thaw gradient (intact peat plateau, thermokarst bog formed 30 years ago, and thermokarst bog formed 200 years ago) in discontinuous permafrost in western Canada. The authors analyzed methanogenic community composition down to 160 cm, measured dissolved $CH_4$ and $CO_2$ concentrations and $\delta^{13}C$ values down to 245 cm in the bogs sites, in addition to rates and $\delta^{13}C$ values of land-atmosphere $CH_4$ and $CO_2$ fluxes. Results from these analyses show that methanogenesis is primarily hydrogenotrophic, rather than acetoclastic, at both the young and mature sites. Young bog had isotopically heavier methane $\delta^{13}C$ values than the mature bog, suggesting that acetoclastic methanogenesis was more enhanced in the young bog. Young bog $CH_4$ emissions were 3x greater than the mature bog. These results imply that $CH_4$ emissions by acetoclastic methanogenesis will increase with continued thermokarst peat plateau collapse and thaw depth lowering in discontinuous permafrost over the next century. As thermokarst bogs mature and dry out, lower temperatures and lower substrate availability will lead to a dominance of hydrogenotrophic methanogenesis.

**Recommendation:**

This is an interesting study that aligns with the research focus of *Biogeosciences*, but further analysis, clarification, and a more robust discussion are needed before this manuscript can be accepted for publication. Below I describe my overall points of concern, and provide suggestions for the authors to improve the manuscript. I also found that the text needs editing and revision to be more easily understood by the reader, and thus I provide detailed line-by-line comments that are more editorial in nature. Therefore, I recommend major revisions.

**Major comments:**

While the data and analyses reported in this manuscript appear robust and offer insight into the effect of thermokarst peat plateau collapse on greenhouse gas emissions, I feel that the authors have not presented any new ideas or conceptual models that help us relate Arctic landscape change to changes in carbon cycling. In= find the discussion and conclusions to be very generalized, attributing the observed differences between young and mature bogs to "hydrological regimes, vegetation communities, and peat chemistry." Statements like this do not provide any insight to the specific mechanisms driving microbial community change. Specifically, it would be useful to quantify the relationship between rate of water table lowering and CO2 and CH4 production rates/magnitudes. I suggest using the ages of the bogs to

determine rate of change in environmental parameters, like thaw depth lowering, water table lowering, and temperature change.

The data for this study were collected from three very localized sites, and it is not clear whether the processes driving CO2 and CH4 production are representative of the greater Arctic landscape. I think the authors need to use their dataset to dive a little deeper into the mechanisms of methanogenesis and transport to the surface (e.g., Throckmorton et al., 2015). I would also like to know exactly which archaeal communities are most important for greenhouse gas production and how they are changing along the thaw gradient (e.g., Høj et al., 2008).

The 16S rRNA data appear to be underutilized, whereas the data could be used to test hypotheses presented by other studies (e.g., Hultman et al., 2015). The study relies heavily on statistical analysis, but it is not clear that the authors are testing specific hypotheses with their analyses. Further hypothesis testing will help to elucidate some of the other processes driving carbon cycling along the thaw gradient.

I also find that the authors make comparative statements that are not supported by statistically significant differences (e.g., in dissolved chemistry parameters). This section of the results is misleading, and also leads to some misleading interpretations of the data (e.g., L853-854).

**Detailed comments:**

*Abstract*
L 32: "(~30 and 200 years since that, respectively)"

L 34-35: "high throughput 16S rRNA gene sequencing"

L 39-40: It would be helpful to give values or the difference between the mean values of the young vs. mature sites

L 42: It would also be useful to give the measured CH4 fluxes in the abstract

L43-45: Be more specific on what the interactions are. I assume that different interactions between ecological conditions and methanogen communities can also reduce CH4 emissions. What exactly are favorable conditions for methanogens and what is the implication for future CH4 emissions as these thermokarst bogs continue to age and turn more hydrogenotrophic?

*Introduction*

L51: "…are *thought to be* driven by…"

L 64: Can thermokarst formation also expose frozen C to aerobic microbial decomposition(e.g., to CO2)? Do we know whether aerobic or anaerobic decomposition result in greater greenhouse emissions? What about the role of methane oxidation by aerobic bacteria or anaerobic archaea (e.g., In't Zandt et al., 2020)?

L133: Does colonization cause fresh, labile inputs of carbon here? I'm not sure what specific process is increasing the amount and temperature sensitivity of CH2 emissions.

L130: Can you be more specific about the "shifting ecological conditions"? Only be more specific if your results allow you to link methane emissions to specific conditions.

L197: "…is *drier than the young bog*, with …"

L217: "…young and mature bog stages, ~1 m from the nearest collar."

L219: remove "deep"

L221: "…devices were installed in each bog…" (since there are only two bogs, you don't need to keep repeating "young and mature bogs")

L221-222: "…where three dissolved gas samples were collected, two from 5-95 cm depth and a third from 115-245 cm depth."

In general, the writing in section 2.2 needs to be improved to make the methods more clear for the reader.

L272-280: It is better practice to first present he equation, the define and give values for all variables. Here, there is a mix of information given before and after the equation. I recommend changing to: "The  CH4 and CO2 land-atmosphere fluxes (F) were calculated following:

$$F = S*(PV/RTA) \quad (1)$$

Where S is the slope of the linear regression fitted to the gas concentration measurements over time inside the flux chamber (units). P is the atmospheric pressure (0.96 atm), . . .

L 285-286: You should write this out using the equation tool.

L 302-303: It would sound better to say: "We measured the $\delta$13C values of gas samples from both the flux chambers and atmospheric background."

Please use proper notation for stable carbon isotope values ($\delta^{13}C$), rather than saying "13C signatures"

L 307: "measured", not "quantified"

L302-311: This paragraph could be written more clearly and concisely. Use of passive voice here makes it difficult to read.

L315: should this be "$(1/[CH_4])$" to denote that it is the concentration of CH4?

L321: Use "collected" rather than "taken"

L 329: "concentration measurements" rather than "concentrations"

L 330-331: Again, $\delta^{13}C$ values, rather than "13C signatures"

L321-340: Again, needs to be written more clearly.

L335: "concentration range"

L337: "measurable range of the system"

L349: "Focusing"

L372: please write out what PVDF stands for(Polyvinylidene difluoride)

L424: "We performed these tests to assess whether thaw stage…"

L430: "Similarly, we tested for significant differences between the depth profiles in the young versus old bogs with respect to dissolved CH4 and CO2 concentrations, d13C values, and alpha-c values."

L412: Better subtitle would be "Statistical analyses"

L434: do you need to mention the instrument again? The illumine miseq is already mentioned in the methods section

L448: There are other key studies that should have been used in the comparison

L492-497: These are not statistically significant differences, so you cannot say that measurements were higher in the mature bog than the young bog. These parameters are statistically identical between the two bogs.

L503: "below the water table", rather than "under"

L504, 505: "Dissolved CH4 concentrations", rather than "concentrations of CH4"

L507:What was the peak concentration in the mature bog? It's difficult to compare the concentrations between the two bogs because you report different types of measurements.

L509-510: It's not clear that the mature bog had higher CO2 concentrations, so this is confusing.

L517: Again, use the delta notation rather than writing "13C isotopic signatures."

L527: "Distinct" is a strong word to use here, given that there are many d13C measurements with overlapping uncertainty in both CO2 and CH4 d13C profiles.

L533: It is not immediately apparent that "F" is the ANOVA F-test statistic. In the methods section, please introduce that you will use the F-test static to compare the two profiles statistically. It's also not clear what the (1, 99) and (1,92) subscripts indicate.

L553-554: Each d13C measurements represents a mixture of two sources (acetoclastic and hydrogenotrophic). It would be much more informative to make a two end-member mixing model and estimate the relative contributions of the two methanogeneis pathways to each measurements. Using these estimates would allow for more quantitative comparison between methanogenesis pathways between the young and mature bogs.

L567-568: Rather than saying "maximum ecosystem respiration in the mature bog was found…", better to say "Ecosystem respiration rates were elevated from June to August, and decreased in September."

L570-577: There are some grammatical mistakes and misuse of punctuation that make this difficult to read.

L674-678: If these variables only explain 18.4 and 4.3% of methanogenic community structure variation, then what other variable are important here? It seems like the analysis needs to go farther/data are inconclusive.

L709-710: Would be worth mentioning that 14C measurements of CO2 and CH4 would help answer the question of whether the emissions are derived from decomposition of fresh, labile DOM or old, previously frozen peat.

L718: up "to" the surficial…

L720: "drier", rather than "relatively drier." Using the word "drier" already implies a comparison

L738: "also been observed"

**Conclusions:**

L853-854: The authors cite a "greater availability of plant leachates" but the DOC and DN data suggest no statistically significant differences in plant leachates between the young and mature bog. This is not a good explanation for the observed shifts in methanogenic communities.

L861: lower temperatures in the mature bog? I assume that the water in the young bog helps to absorb more heat, but this should be made clear in the manuscript.

In the conclusions, it would be useful to mention the relative contributions of acetoclastic and hydrogenotrophic methanogenesis to methane emissions in the two bogs.

**Figures:**

Figure 1: would be helpful to label photos d and e with "mature" and "young"

Figure 2: You need a legend showing which colors represent young vs. mature bog. It is better practice to not rely on explanation in the figure caption, but to give the reader essential information in the figure itself. I would suggest using a different shape for the data points for one of the bogs too.

Figure 2(a): It would be helpful to write "CH4 concentrations" and "CO2 concentrations" or use concentration notation ([CO2], [CH4]). I also don't understand the arrows. Why are they pointing to specific points? Don't these represent the top of the permafrost/bottom of active layer? It would make more sense to have additional horizontal lines rather than arrows, unless that looks too busy with the water table levels.

Figure 2(f): To help guide the reader, I suggest using background shading and labels to identify the regions of acetoclastic and hydrogenotrophic methanogesis

Figure 4: I like this figure, but it could be arranged differently to take up less space on the page – the circles and triangle legend could go below the color bars, or the entire legend could go on top of the plot and the text can wrap around the right side of the figure.

Figure 6: In the text, you say that you assess only thaw stage and distance to water table, but in the caption you say that you explore both biotic and abiotic factors. What else was included in

this analysis that is not shown in the plot that seemingly explains more of the methanogenic community variation?

**References cited above:**

Høj, L., Olsen, R. & Torsvik, V. Effects of temperature on the diversity and community structure of known methanogenic groups and other archaea in high Arctic peat. *ISME J* **2,** 37–48 (2008). https://doi.org/10.1038/ismej.2007.84

Throckmorton, H.M., Heikoop, J.M., Newman, B.D., Altmann, G.L., Conrad, M.S., Muss, J.D., Perkins, G.B., Smith, L.J., Torn, M.S., Wullschleger, S.D. and Wilson, C.J., 2015. Pathways and transformations of dissolved methane and dissolved inorganic carbon in Arctic tundra watersheds: Evidence from analysis of stable isotopes. *Global Biogeochemical Cycles*, *29*(11), pp.1893-1910.
*Data show a temporal shift in methanogenesis pathways, from acetoclastic in July to hydrogenotrophic in September.*

Hultman, J., Waldrop, M., Mackelprang, R. *et al.* Multi-omics of permafrost, active layer and thermokarst bog soil microbiomes. *Nature* **521,** 208–212 (2015). https://doi.org/10.1038/nature14238
*Active layer communities expressed genes and proteins involved in obtaining energy and nutrients from a diversity of aerobic and anaerobic processes and were equipped with functions for survival under freeze–thaw conditions. The bog represented a different scenario with a very high measured rate of methanogenesis and correspondingly high relative abundances of genes, transcripts and proteins involved in methanogenesis, thus demonstrating the potential linkage between molecular data and ecosystem level process rates*

---

## Referee Comment (RC3)

The objectives of this study were to assess the impacts of time following permafrost thaw stage on methane emissions and methanogenic community composition. To do this, the authors identified two bog sites with permafrost that thawed 30 and 200 years ago. Analyses conducted at these sites included (1) metagenomic assessments, (2) dissolved gas concentrations (CO2 and CH4), (3) surface emissions,(4) d13C signatures for both CH4 and CO2 (used to assess the relative contribution of acetoclastic methanogenesis to total methanogenesis).

Overall, this paper effectively approaches that goal.

I have two primary concerns:

(1) My main concern with this paper is centered around the use of isotope d13C signatures. The alpha value is an accepted method for discerning the relative contributions of acetoclastic vs. hydrogenotrophic methanogenesis. While the authors used alpha values for dissolved gas analysis, they limited their assessments of acetoclastic input to 13C signatures of methane (without concomitant 13C-CO2 signatures). Is the 13C-CH4 signature on its own a sufficient indicator of acetoclastic contribution? If so, please provide citations that indicate so.

Furthermore—I was not clear on whether the apparent difference in alpha values between mature vs. young sites was indeed statistically significant. This comparison needs to be made explicit. If significant differences between sites can only be found at specific depth intervals, then that should also be stated explicitly.

(2) Your goal was to examine the effects of thaw stage on methane fluxes/methanogenic community composition. It is difficult to wrap my head around this goal since thaw succession causes shifts in so many different environmental factors (soil temperature, thickness of the unsaturated peat column, availability of labile organics). This makes your results difficult to build off of/apply to other settings. Perhaps you could perform an ordinary least squares regression analysis (OLS) to try to tease apart the relative influence of these numerous factors on a dependent variable of interest (perhaps total methane emissions, or acetoclastic methane emissions).

Specific Questions/Recommendations

(1) I am unclear on what is mean by the term "ecological" in the context of this manuscript (e.g. L27-28). I get the impression that it references vegetation primarily. I am unsure about that, however, because "ecological" could also be used to describe microbial community composition. Please clarify.
(2) Fig 1: I'd recommend explicitly stating how far apart the mature and young bog sites are from one another in panel f (or the caption). You list only one GPS coordinate from the whole site in the figure caption.
(3) L158-159: How did you determine that this complex is representative? If the succeeding sentences are meant to serve as evidence for this claim, make that explicit.
(4) L551-554: "Overall, the isotopic data indicates a general dominance of hydrogenotrophic methanogenesis in both sites, but a greater contribution of acetoclastic methanogenesis in the

young bog relative to the mature bog." Was this difference statistically significant? I suggest adding a p-value after this statement.

(5) L 572-575: *"The δ13C-CH4 signature of CH4 emissions (intercept values from Keeling plots), in the young bog were significantly greater than those observed in the mature bog (Figure 3c; F (1, 4) = 20.67, P< 0.05)., suggesting a greater influence of acetoclastic CH4 production."*

Why is there no alpha value for the flux measurements? Please provide a source indicating that 13C-CH4 measurements alone (i.e. without concomitant 13C-CO2) are sufficient to discern the relative influence of acetoclastic methanogenesis on total methane production.

(6) L704-706: *"Evidence of acetoclastic methanogens and CH4 produced via the acetoclastic metabolic pathway was found in the young bog both near the surface and at depths below the thaw transition (i.e., in peat that accumulated prior to permafrost thaw)."*

I have two notes on this:

(1) If the difference in alpha values was not significant in the subsurface (which I am not 100% clear on), this needs to be noted and discussed.
(2) See my comments regarding L572-575. Make sure your methods for discerning the acetoclastic influence on surface CH4 emissions is sound.

(7) L765-766: "*The presence of hydrophilic vegetation, particularly graminoids, in the saturated young bog provides the precursors for fermentation..*"

I am confused by this statement. "Precursors" could be interpreted as "reactants", which are primarily sugars. Sugars are ultimately delivered to porewater from other plants too (i.e. Sphagnum spp.). Are you saying that the sugars derived from gramminoids are more labile than those derived from Sphagnum? I would agree with this, but it is necessary to clarify.

(8) L805-813: I find the thread of this paragraph hard to follow. Please make the connections between sentences clearer.
(9) Fig 2: Please add in a legend.

---

## Author Comment (AC1)

Review of "High peatland methane emissions following permafrost thaw: enhanced acetoclastic methanogenesis during early successional stages" by Liam Heffernan and others.

Summary:

The goal of this manuscript is to advance our understanding of the underlying controls of methane emissions from permafrost thaw in northern peatlands. Specifically, the authors assess how shifting ecological conditions (e.g., collapse of peat plateau and thermokarst bog formation) affect microbial communities, the amount of $CH_4$ released, and the $d_{13}C$ isotope composition of released $CH_4$. The authors also aim to determine how long elevated surface $CH_4$ emissions persist after thaw.
To answer these questions, the authors study peatland methanogenic community composition and methane emissions along a thaw gradient (intact peat plateau, thermokarst bog formed 30 years ago, and thermokarst bog formed 200 years ago) in discontinuous permafrost in western Canada. The authors analyzed methanogenic community composition down to 160 cm, measured dissolved $CH_4$ and $CO_2$ concentrations and $d_{13}C$ values down to 245 cm in the bogs sites, in addition to rates and $d_{13}C$ values of land-atmosphere $CH_4$ and $CO_2$ fluxes. Results from these analyses show that methanogenesis is primarily hydrogenotrophic, rather than acetoclastic, at both the young and mature sites. Young bog had isotopically heavier methane $d_{13}C$ values than the mature bog, suggesting that acetoclastic methanogenesis was more enhanced in the young bog. Young bog $CH_4$ emissions were 3x greater than the mature bog. These results imply that $CH_4$ emissions by acetoclastic methanogenesis will increase with continued thermokarst peat plateau collapse and thaw depth lowering in discontinuous permafrost over the next century. As thermokarst bogs mature and dry out, lower temperatures and lower substrate availability will lead to a dominance of hydrogenotrophic methanogenesis.

Recommendation:
This is an interesting study that aligns with the research focus of Biogeosciences, but further analysis, clarification, and a more robust discussion are needed before this manuscript can be accepted for publication. Below I describe my overall points of concern, and provide suggestions for the authors to improve the manuscript. I also found that the text needs editing and revision to be more easily understood by the reader, and thus I provide detailed line-byline comments that are more editorial in nature. Therefore, I recommend major revisions.

*Thank you for your feedback! We below respond to your comments and suggestions to better revise our manuscript.*

Major comments:
While the data and analyses reported in this manuscript appear robust and offer insight into the effect of thermokarst peat plateau collapse on greenhouse gas emissions, I feel that the authors have not presented any new ideas or conceptual models that help us relate Arctic landscape change to changes in carbon cycling. I find the discussion and conclusions to be very generalized, attributing the observed differences between young and mature bogs to "hydrological regimes, vegetation communities, and peat chemistry." Statements like this do not provide any insight to the specific mechanisms driving microbial community change.

*With this manuscript, we present the first study to combine microbial and biogeochemical data to assess the influence of permafrost thaw on methanogenesis and CH₄ emissions along a space-for-time thermokarst bog transect. We will review and modify the discussion/conclusion to avoid very generalized statements and try to better link the biogeochemical and microbial community data.*

Specifically, it would be useful to quantify the relationship between rate of water table lowering and CO2 and CH4 production rates/magnitudes. I suggest using the ages of the bogs to determine rate of change in environmental parameters, like thaw depth lowering, water table lowering, and temperature change.

*This is an interesting thought; however, we do not think it reasonable to make any further extrapolation on how shifting ecological conditions will impact emissions using this specific dataset, which may be out of the scope of this study. A similar point was also made by RC3, wondering if we could determine how different conditions following thaw may influence total CH₄ emissions. To determine the relationship between magnitude of fluxes and site conditions we would need a larger, more comprehensive dataset that consists of either multiple years of data, or multiple sites, or both. This is the objective of a yet to be published study from this site, that includes 3 years of flux data and a secondary site. The objective of this study was to compare sites with different thaw histories, and thus differencing current ecological conditions. We will make the scope and objectives of the study clearer in the introduction to address both reviewers' concerns.*

The data for this study were collected from three very localized sites, and it is not clear whether the processes driving CO2 and CH4 production are representative of the greater Arctic landscape. I think the authors need to use their dataset to dive a little deeper into the mechanisms of methanogenesis and transport to the surface (e.g., Throckmorton et al., 2015). I would also like to know exactly which archaeal communities are most important for greenhouse gas production and how they are changing along the thaw gradient (e.g., H̃ j et al., 2008).

*The study site is considered to be representative of boreal peatlands in the discontinuous permafrost zone in the Mackenzie River Basin of western Canada; see below for some references supporting this. In short, this area is comprised of intact peat plateaus interspersed with permafrost free bogs, fens, and ponds. Permafrost peatlands in this area are very similar to those found in the Hudson Bay Lowlands (Kuhry, 2008) and Alaska (Jones et al., 2017)*

*We do not think there is a single site or ecosystem that is representative of the greater Arctic landscape or northern circumpolar permafrost region. However, our study system does represent a globally significant organic carbon store that is vulnerable to permafrost thaw and potential mineralization into greenhouse gases. Peatlands in the Mackenzie River Basin are one of the three largest stores of organic carbon found in peatlands within the permafrost zone, the other two being the Hudson Bay Lowlands and the West Siberian Lowlands (Hugelius et al., 2020; Olefeldt et al., 2021). Within the sporadic and discontinuous permafrost zone of our study*

region >15% of the total peat plateau area has thawed and formed thermokarst bogs in the last 30 years (Baltzer et al., 2014; Gibson et al., 2018). Projections for this area suggests total permafrost lost from plateaus by 2050 (Chasmer and Hopkins, 2017). Thus, we consider the results of this study results to be important.

Regarding the mechanisms of methanogenesis and transport pathways to the surface, initially we considered the mass balance approach used by Thockmorton et al., (2015) as well by Corbett et al., (2013). We agree that this is a very interesting approach to answer questions regarding the pathways of anaerobic fermentation and decomposition, and vertical transport of the end-products of this decomposition. We use a similar approach in determining the pathways of methanogenesis responsible for dissolved concentrations of $CH_4$ at depth to that described in these papers. However, we do not follow a similar approach in assessing the transport of the resulting dissolved gases of anaerobic decomposition. We decided to not take such an approach as it was beyond the scope and objectives of our study. Our study focuses on assessing how shifting ecological conditions following permafrost thaw influence the structure and activity of the methanogen community, the pathways of methanogenesis, and surface $CH_4$ emissions. We do not focus on how dissolved $CH_4$ reaches the surface, or where in the peat profile the $CH_4$ emitted at the surface was produced. Rather, we focus on how methanogenesis and the microbial community responsible for methanogenesis is affected in the top 160 cm of a peatland following permafrost thaw, whether this results in greater surface $CH_4$ emissions, and for how long these surface emissions may last (decades to centuries). We believe that the combination of microbial data (16S) and biogeochemical data (dissolved concentrations, $\delta^{13}C$ signatures, surface emissions) from areas that have thawed 30 and 200 years ago in a thermokarst bog is novel, timely, and interesting.

Regarding which archaeal communities are most important for greenhouse gas production and how they are changing along the thaw gradient, we also agree that this is a very interesting and timely question. This question, however, is beyond the scope of our study. Here, we aim to address the influence that shifting ecological conditions, following permafrost thaw, has on methanogenesis specifically, not on anaerobic chemoheterotrophy in general. This indeed would be a fascinating topic for a future study that would include not just 16S data but also various other metaOmics. The dataset for this study is open and freely available, we would be more than happy to discuss the contribution of this data to any such studies in the future.

*References – boreal western Canada*
- *Bauer, I. E., Gignac, L. D., & Vitt, D. H. (2003). Development of a peatland complex in boreal western Canada: Lateral site expansion and local variability in vegetation succession and long-term peat accumulation. Canadian Journal of Botany, 81(8), 833–847. https://doi.org/10.1139/b03-076*
- *Beilman, D. W. (2001). Plant community and diversity change due to localized permafrost dynamics in bogs of western Canada. Canadian Journal of Botany, 79(8), 983–993. https://doi.org/10.1139/cjb-79-8-983*
- *Camill, P. (1999). Peat accumulation and succession following permafrost thaw in the Boreal peatlands of Manitoba, Canada. Ecoscience, 6(4), 592–602. https://doi.org/10.1080/11956860.1999.11682561*
- *Pelletier, N., Talbot, J., Olefeldt, D., Turetsky, M., Blodau, C., Sonnentag, O., & Quinton, W. L. (2017). Influence of Holocene permafrost aggradation and thaw on the paleoecology and carbon*

storage of a peatland complex in northwestern Canada. *Holocene, 27(9), 1391–1405.* *https://doi.org/10.1177/0959683617693899*

- Vitt, D. H., Halsey, L. A., & Zoltai, S. C. (1994). The Bog Landforms of Continental Western Canada in Relation to Climate and Permafrost Patterns. Arctic and Alpine Research, 26(1), 1. *https://doi.org/10.2307/1551870*
- Vitt, D. H., Halsey, L. A., Bauer, I. E., & Campbell, C. (2000). Spatial and temporal trends in carbon storage of peatlands of continental western Canada through the Holocene. Canadian Journal of Earth Sciences, 37(5), 683–693. *https://doi.org/10.1139/e99-097*
- Zoltai, S. C. (1972). Palsas and Peat Plateaus in Central Manitoba and Saskatchewan. Canadian Journal of Forest Research, 2(3), 291–302. *https://doi.org/10.1139/x72-046*
- Zoltai, S. C. (1993). Cyclic Development of Permafrost in the Peatlands of Northwestern Alberta, Canada. Arctic and Alpine Research, 25(3), 240. *https://doi.org/10.2307/1551820*

*References - other*

- Kuhry, P. (2008), Palsa and peat plateau development in the Hudson Bay Lowlands, Canada: timing, pathways and causes. Boreas, 37: 316-327. *https://doi.org/10.1111/j.1502-3885.2007.00022.x*
- Jones, M. C., Harden, J., O'Donnell, J., Manies, K., Jorgenson, T., Treat, C., & Ewing, S. (2017). Rapid carbon loss and slow recovery following permafrost thaw in boreal peatlands. Global Change Biology, 23(3), 1109–1127. *https://doi.org/10.1111/gcb.13403*
- Hugelius, G., Loisel, J., Chadburn, S., Jackson, R. B., Jones, M., MacDonald, G., Marushchak, M., Olefeldt, D., Packalen, M., Siewert, M. B., Treat, C., Turetsky, M., Voigt, C., & Yu, Z. (2020). Large stocks of peatland carbon and nitrogen are vulnerable to permafrost thaw. Proceedings of the National Academy of Sciences of the United States of America, 117(34), 20438–20446. https://doi.org/10.1073/pnas.19163 87117
- Olefeldt, D., Heffernan, L., Jones, M. C., Sannel, A. B. K., Treat, C. C., & Turetsky, M. R. (2021). Permafrost thaw in northern peatlands: rapid changes in ecosystem and landscape functions. Ecosystem Collapse and Climate Change, 27-67.
- Baltzer JL, Veness T, Chasmer LE, et al (2014) Forests on thawing permafrost: fragmentation, edge effects, and net forest loss. Global Change Biology 20:824–834. doi: 10.1111/gcb.12349
- Gibson CM, Chasmer LE, Thompson DK, et al (2018) Wildfire as a major driver of recent permafrost thaw in boreal peatlands. Nature Communications 9:3041. doi:10.1038/s41467-018-1034 05457-1
- Chasmer L, Hopkinson C (2017) Threshold loss of discontinuous permafrost and landscape evolution. Global Change Biology 23:2672–2686. doi: 10.1111/gcb.13537
- Corbett, J. E., M. M. Tfaily, D. J. Burdige, W. T. Cooper, P. H. Glaser, and J. P. Chanton (2013), Partitioning pathways of CO2 production in peatlands with stable carbon isotopes, Biogeochemistry, 114(1–3), 327–340.
- Throckmorton, H.M., Heikoop, J.M., Newman, B.D., Altmann, G.L., Conrad, M.S., Muss, J.D., Perkins, G.B., Smith, L.J., Torn, M.S., Wullschleger, S.D. and Wilson, C.J., 2015. Pathways and transformations of dissolved methane and dissolved inorganic carbon in Arctic tundra watersheds: Evidence from analysis of stable isotopes. Global Biogeochemical Cycles, 29(11), pp.1893-1910.

The 16S rRNA data appear to be underutilized, whereas the data could be used to test

hypotheses presented by other studies (e.g., Hultman et al., 2015).

*While it is true that 16S RrNA gene data may be underutilized in studies such as these, there are numerous constraints on what can be done with (and concluded in using) this kind of microbial taxonomic data in tandem with biogeochemical data. We therefore wanted to limit our interpretation and discussion to the methanogenic community so that we do not "overreach" with what our data could tell us about this system ...*

*The hypotheses presented by the study exemplified (Hultman et al., 2015) is a more robust study in that it combines not just 16S microbial taxonomic data, but also metaOmics data such as proteomics, metatranscriptomics and metagenomics to specifically target the functional processes occurring in their system. With our dataset, we can explore the putative metabolisms involved, but with significant limitations, as 16S cannot be directly tied to microbial metabolic function. In an attempt to gain further insight into putative microbial function, we applied FAPROTAX, a bioinformatics tool that can predict ecologically relevant functions from 16S microbial taxonomic data (Louca et al., 2016, Sansupa et al., 2021), to our dataset. However, it was unable to resolve whether particular methanogenic pathways were taking place in different stages of thaw, and thus we chose not include this analysis. Instead the insight gleaned about the archaeal community composition from the 16S rRNA gene analysis, in conjunction with the isotopic signatures for $CH_4$ and $CO_2$, was ultimately more convincing in identifying the dominant methanogen pathways along our thaw gradient. There is previous precedent to combining 16S 16S rRNA gene and biogeochemical data in a similar fashion to this study to gain insight into changing microbial community structure (Ganzert et al., 2007; Saidi-Mehrabad et al., 2020; Cherbunina et al., 2021), as well as others that incorporate both 16SrRNA gene sequencing and targeted qPCR/metaOmics to more definitively determine more microbially-driven processes in permafrost ( Wen et al., 2018, Unger et al., 2021).*

*References:*

- *Lars Ganzert, German Jurgens, Uwe Münster, Dirk Wagner, Methanogenic communities in permafrost-affected soils of the Laptev Sea coast, Siberian Arctic, characterized by 16S rRNA gene fingerprints, FEMS Microbiology Ecology, Volume 59, Issue 2, February 2007, Pages 476–488, https://doi.org/10.1111/j.1574-6941.2006.00205.x*

- *Saidi-Mehrabad, A., Neuberger, P., Hajihosseini, M., Froese, D., Lanoil, B.D. (2020). Permafrost microbial community structure changes across the Pleistocene-Holocene Boundary. Front. Environ. Sci. https://doi.org/10.3389/fenvs.2020.00133*

- *Cherbunina, M.Yu., Karaevskaya, E.S., Vasil'chuk, Yu.K., Tananaev, N.I., Smelev, D.G., Budantseva, N.A., Merkel, A.Y., Rakitin, A.L., Mardanov, A.V., Brouchkov, A.V., Bulat, S.A. (2021). Microbial and Geochemical evidence of permafrost formation at Mamontova Gora and Syrdakh, Central Yakutia. Front. Earth. Sci. https://doi.org/10.3389/feart.2021.739365*

- *Wen, X., Unger, V., Jurasinski, G., Koebsch, F., Horn, F., Rehder, G., Sachs, T., Zak, D., Lischeid, G., Knorr, K., Böttcher, M.E., Winkel, M., Bodelier, P.L., & Liebner, S. (2018).*

*Predominance of methanogens over methanotrophs in rewetted fens characterized by high methane emissions. Biogeosciences.*

- *Unger, V., Liebner,S., Koebsch, F., Yang,S., Horn,F., Sachs, T.,Kallmeyer, J., Klaus-Holger, K., Rehder,G., Gottschalk, P., Jurasinski,G. (2021). Congruent changes in microbial community dynamics and ecosystem methane fluxes following natural drought in two restored fens,*
  *Soil Biology and Biochemistry 160: https://doi.org/10.1016/j.soilbio.2021.108348.*

The study relies heavily on statistical analysis, but it is not clear that the authors are testing specific hypotheses with their analyses. Further hypothesis testing will help to elucidate some of the other processes driving carbon cycling along the thaw gradient.

*The hypotheses that we aim to answer, are specified in the introduction (Lines 130-136). There, we state that we hypothesize "(1) shifting ecological conditions along the permafrost thaw gradient results in a successional microbial community and a restructuring of the methanogenic community, and (2) the warmer conditions in the young bog, along with the exposure of previously frozen peat, will result in a greater relative abundance of acetoclastic methanogens throughout the depth profile, and subsequently greater overall CH4 emissions."*

*To test our first hypothesis, our 16S microbial data was used to test whether there is evidence of distinct groupings of methanogen communities using NMDS and ANOSIM (L434-444) down to 160 cm depth in areas that have thawed 30 and 200 years ago. We then used RDA and variance partitioning (L445-471) to test how biogeochemical and site data from these two different thawed areas influence the 16S data and methanogen community structure. Using our dataset, we unfortunately cannot get more specific than this without further metaOmic data or qPCR data.*

*To test our second hypothesis, we used ANOVAs and Bonferroni post-hoc tests on linear mixed effects models (L422-433) to test for differences in the concentrations and $\delta^{13}C$ signatures of surface gas fluxes and dissolved gas depth profiles down to 245 cm between the two thawed areas.*

*Could the reviewer be more specific regarding what they mean by testing and how they deem these tests to not be sufficient in addressing our hypotheses? We welcome any suggestions for further hypotheses they would consider testing with the dataset available to us.*

I also find that the authors make comparative statements that are not supported by statistically significant differences (e.g., in dissolved chemistry parameters). This section of the results is misleading, and also leads to some misleading interpretations of the data (e.g., L853-854).

*We will better highlight the pore water chemistry parameters that are statistically different between the sites (pH and DOC). We will further discuss those parameters that are not statistically different, and which have large standard deviations associated with them (SUVA and TDN). We will also better highlight that the differences in lability associated with the young and mature bog is also inferred from previous work at this study site (Heffernan et al. 2021), but also from work at other closely related sites with similar vegetation communities (Burd et al., 2020).*

*References*
- *Heffernan, L., Jassey, V. E. J., Frederickson, M., MacKenzie, M. D., & Olefeldt, D. (2021). Constraints on potential enzyme activities in thermokarst bogs: Implications for the carbon balance of peatlands following thaw. Global Change Biology, 27, 4711–4726. https://doi.org/10.1111/gcb.15758*
- *Burd, K., Estop-Aragonés, C., Tank, S. E., & Olefeldt, D. (2020). Lability of dissolved organic carbon from boreal peatlands: Interactions between permafrost thaw, wildfire, and season. Canadian Journal of Soil Science, 13(February), 503–515. https://doi.org/10.1139/cjss-2019-0154*

Detailed comments:

Abstract

L 32: "(~30 and 200 years since that, respectively)"

*We will change L32 to read:"~30 and 200 years since thaw, respectively*

L 34-35: "high throughput 16S rRNA gene sequencing"

*We will change L34-35 to read:"~...high throughput 16S rRNA gene sequencing"...*

L 39-40: It would be helpful to give values or the difference between the mean values of the young vs. mature sites

*We will add these values to the abstract*

L 42: It would also be useful to give the measured CH4 fluxes in the abstract

*We will add the rates of CH4 fluxes to the abstract*

L43-45: Be more specific on what the interactions are. I assume that different interactions between ecological conditions and methanogen communities can also reduce CH4 emissions. What exactly are favorable conditions for methanogens and what is the implication for future CH4 emissions as these thermokarst bogs continue to age and turn more hydrogenotrophic?

*We will be more specific about what interactions we are talking about here. Namely, we will say that warmer temperatures, higher water table, and hydrophilic vegetation in the young bog are favourable for enhanced CH₄ emissions. It will now read as*
*"Our study suggests that interactions between the methanogenic community and hydrophilic vegetation, warmer temperatures, and saturated surface conditions enhance CH4 emissions in young thermokarst bogs, but these favorable conditions only persist for the initial decades after permafrost thaw."*

Introduction

L51: "…are thought to be driven by…"

*We will change L51 to read:  "~…are thought to be driven by", as suggested.*

L 64: Can thermokarst formation also expose frozen C to aerobic microbial decomposition(e.g., to CO2)? Do we know whether aerobic or anaerobic decomposition result in greater greenhouse emissions? What about the role of methane oxidation by aerobic bacteria or anaerobic archaea (e.g., In't Zandt et al., 2020)

*Yes, thermokarst formation can expose previously frozen organic matter to aerobic respiration, resulting in increased CO₂ emissions (Schädel et al., 2016). However, thermokarst formation in peatlands is characterized by ground subsidence, resulting in saturated surface conditions (Camill, 1999). These saturated surface conditions result in previously frozen peat being exposed to anoxic conditions. Peatlands are a wetland ecosystem where anoxia is the dominant redox condition and anaerobic decomposition is the main form of decomposition.*

*In general, aerobic respiration occurs at a faster rate than anaerobic respiration as has been shown for other, non-peatland, thermokarst ecosystems (Schädel et al., 2016). We will add some text to reflect this in the introduction. This new text will read*
*"Redox conditions following thermokarst formation are an important control of decomposition, with 3-4 times as C mineralization occurring as aerobic respiration compared to anaerobic respiration (Schädel et al., 2016)"*

*We address CH₄ oxidation in the manuscript (L798-807) but will elaborate further. Our study objectives were not to explore the relationship between ecological conditions following thaw and aerobic respiration at the surface in peatlands, but rather the anaerobic processes beneath the water table. The In't Zandt et al., 2020 study focuses on thermokarst lakes in ice-rich Yedoma deposits. While the role of CH₄ oxidation within anaerobic lake sediments in these systems is an interesting one, we do not think it entirely relevant to our study. In thermokarst affected permafrost peatlands, CH₄ oxidation has been shown to be closely linked with redox potential associated with the water table position (Perryman et al., 2020). This is included in our discussion of oxidation in the manuscript.*

*References*

- *Schädel, C., Bader, M. K. F., Schuur, E. A. G., Biasi, C., Bracho, R., Capek, P., et al. (2016). Potential carbon emissions dominated by carbon dioxide from thawed permafrost soils. Nature Climate Change, 6(10), 950–953. https://doi.org/10.1038/nclimate3054*
- *Perryman, C. R., McCalley, C. K., Malhotra, A., Fahnestock, M. F., Kashi, N. N., Bryce, J. G., ... Varner, R. K. (2020). Thaw Transitions and Redox Conditions Drive Methane Oxidation in a Permafrost Peatland. Journal of Geophysical Research: Biogeosciences, 125(3). https://doi.org/10.1029/2019JG005526*

L133: Does colonization cause fresh, labile inputs of carbon here? I'm not sure what specific process is increasing the amount and temperature sensitivity of CH2 emissions.

*Yes, colonization of hydrophilic vegetation following thaw is associated with an increase in labile inputs. This increase in labile inputs can increase methanogenesis, and thus, the sensitivity of methanogenesis to temperature. We cite references in the text (L113) that address this and shaped this hypothesis.*

L130: Can you be more specific about the "shifting ecological conditions"? Only be more specific if your results allow you to link methane emissions to specific conditions.

*The shifting ecological conditions associated with permafrost thaw in peatlands is outlined in the text above (L62; L87-91; L102-104) and at our study site in the methods section below (Section 2.1 L147). We will add a new, thorough, and clear definition of what we mean by ecological conditions in relation to the conditions found in the young and mature bog at our study site*

L197: "…is drier than the young bog, with …"

*We will change L197 accordingly*

L217: "…young and mature bog stages, ~1 m from the nearest collar."

*We are unsure about what change is suggested, the suggestion is exactly similar to the current text.*

L219: remove "deep"

*We will remove "deep" from L219.*

L221: "…devices were installed in each bog…" (since there are only two bogs, you don't need to
keep repeating "young and mature bogs")
*We will remove this repetition as suggested.*

L221-222: "…where three dissolved gas samples were collected, two from 5-95 cm depth and a third from 115-245 cm depth."

*We will add this change to the text as suggested.*

In general, the writing in section 2.2 needs to be improved to make the methods more clear for the reader.
*This section will be re-written to improve clarity .*

L272-280: It is better practice to first present he equation, the define and give values for all variables. Here, there is a mix of information given before and after the equation. I recommend changing to: "The rates of CH4 and CO2 land-atmosphere fluxes (F) were calculated following:
F = S*(PV/RTA) (1)
Where S is the slope of the linear regression fitted to the gas concentration measurements over time inside the flux chamber (units). P is the atmospheric pressure (0.96 atm), . . .

*We will change this text accordingly.*

L 285-286: You should write this out using the equation tool.

*We will change this text accordingly.*

L 302-303: It would sound better to say: "We measured the d13C values of gas samples from both the flux chambers and atmospheric background."

*We will change this text accordingly.*

Please use proper notation for stable carbon isotope values (d13C), rather than saying "13C signatures"

*We will change "$^{13}C$ isotopic signatures" to "$\delta^{13}C$" throughout the text.*

L 307: "measured", not "quantified"

*We will change L307 as specified.*

L302-311: This paragraph could be written more clearly and concisely. Use of passive voice here makes it difficult to read.

*We will edit this text to improve clarity.*

L315: should this be "(1/[CH4])" to denote that it is the concentration of CH4?

*We will change this text accordingly, here and throughout the manuscript.*

L321: Use "collected" rather than "taken"

*We will change L321 to read as "Dissolved gas samples were collected…"*

L 329: "concentration measurements" rather than "concentrations"

*We will change this as suggested.*

L 330-331: Again, d13C values, rather than "13C signatures"

*Addressed above.*

L321-340: Again, needs to be written more clearly.

*This section will be edited to improve clarity.*

L335: "concentration range"

*We will change L335 as specified.*

L337: "measurable range of the system"

*We will change L337 as specified.*

L349: "Focusing"

*We will change L349 as specified.*

L372: please write out what PVDF stands for( Polyvinylidene difluoride)

*We will change L372 to "...pore size Polyvinylidene difluoride (PVDF) membrane…"*

L424: "We performed these tests to assess whether thaw stage…"

*We will change L424 accordingly.*

L430: "Similarly, we tested for significant differences between the depth profiles in the young versus old bogs with respect to dissolved CH4 and CO2 concentrations, d13C values, and alphac values."

*We will change L430 to: "Similarly, we tested for significant differences between the depth profiles in the young versus old bogs with respect to dissolved CH4 and CO2 concentrations, d13C values, and alpha c values…"*

L412: Better subtitle would be "Statistical analyses"

*We will change the subtitle to "Statistical Analyses", as suggested.*

L434: do you need to mention the instrument again? The illumine miseq is already mentioned in the methods section

*Good point- we will remove the additional mention of the Illumina Miseq*

L448: There are other key studies that should have been used in the comparison

*We provide additional references, Kendall & Boone (2006) and Zhang et al., (2020). In case further studies exist we are not yet aware of, we would be glad to include them and kindly ask the reviewer to provide more suggestions here.*

*References:*
- *Kendall MM, Boone DR. Cultivation of methanogens from shallow marine sediments at Hydrate Ridge, Oregon. Archaea. 2006 Aug;2(1):31-8. doi: 10.1155/2006/710190. PMID: 16877319; PMCID: PMC2685590.*
- *Zhang, CJ., Pan, J., Liu, Y. et al. Genomic and transcriptomic insights into methanogenesis potential of novel methanogens from mangrove sediments. Microbiome 8, 94 (2020). https://doi.org/10.1186/s40168-020-00876-z*

L492-497: These are not statistically significant differences, so you cannot say that measurements were higher in the mature bog than the young bog. These parameters are statistically identical between the two bogs.

*Statistically, pH and DOC are different (ANOVA; $p < 0.05$). We will clearly outline that these are statistically different and provide the test results. For the rest of the results that are not significantly different, we will clearly state this and also provide the test results. However, we will still state which averages are higher, particularly for SUVA and TDN, which have large standard deviations associated with them. While statistically not different, the differences in SUVA and TDN between stages of thaw may be sufficient to impact the microbial community structure (Bradley et al 2017). Nevertheless, we agree that we must point out more clearly where we found significant differences and where there were only insignificant trends or tendencies. References*
- *Bradley JA, Anesio AM and Arndt S (2017) Microbial and Biogeochemical Dynamics in Glacier Forefields Are Sensitive to Century-Scale Climate and Anthropogenic Change. Front. Earth Sci. 5:26. doi: 10.3389/feart.2017.00026*

L503: "below the water table", rather than "under"

*We will change L503 accordingly.*

L504, 505: "Dissolved CH4 concentrations", rather than "concentrations of CH4"

*We will change L504 & L505 accordingly.*

L507: What was the peak concentration in the mature bog? It's difficult to compare the concentrations between the two bogs because you report different types of measurements.

*Peak concentration in the mature bog was 6,800 µmol $L^{-1}$. This will be added to the text*

L509-510: It's not clear that the mature bog had higher CO2 concentrations, so this is confusing.

*Figure 2 shows that the mature bog has higher $CO_2$ concentrations, and we state in Lines 510-511 that the peak values were higher in the mature bog and provide these concentrations. " Again, the mature bog had overall higher concentrations, peaking at 1,500 $\mu mol$ $L-1$ at 85 cm while the young bog peaked at 1,200 $\mu mol$ $L-1$ at 95 cm (Figure 2b)."*

L517: Again, use the delta notation rather than writing "13C isotopic signatures."

*Addressed above.*

L527: "Distinct" is a strong word to use here, given that there are many d13C measurements with overlapping uncertainty in both CO2 and CH4 d13C profiles.

*The depth profiles of both $\delta^{13}C\text{-}CO_2$ and $\delta^{13}C\text{-}CH_4$ are statistically different from one another both above (L533; ANOVA (F (1, 92) = 17.25, P < 0.001) and below the water table (L536); ANOVA F (1, 99) = 5.33, P < 0.05) thus we consider distinct to be an appropriate word.*

L533: It is not immediately apparent that "F" is the ANOVA F-test statistic. In the methods section, please introduce that you will use the F-test static to compare the two profiles statistically. It's also not clear what the (1, 99) and (1,92) subscripts indicate.

*We will clearly state in the methods section that we are using ANOVA and will report the F statistic throughout when reporting the results from our ANOVA. We will also add "ANOVA" to the results section when reporting the F statistic.*

L553-554: Each d13C measurements represents a mixture of two sources (acetoclastic and hydrogenotrophic). It would be much more informative to make a two end-member mixing model and estimate the relative contributions of the two methanogenesis pathways to each measurements. Using these estimates would allow for more quantitative comparison between methanogenesis pathways between the young and mature bogs.

*We agree that an end member mixing model would be an interesting way to answer the questions we are addressing in the manuscript. Unfortunately, we do not have the correct dataset to do so. To perform an end-member mixing model we would need the specific fractionation factors associated with acetoclastic and hydrogenotrophic methanogenesis at our site. These fractionation factors vary considerably across sites (Conrad, 2005), thus we cannot use values from the literature. To determine these fractionation factors we would need to either perform an in situ labelling experiment or have a high resolution of $\delta^{13}C$ data of organic matter at each depth from where we have dissolved $CO_2$ and $CH_4$ concentrations. As we do not have either of these an end member mixing model approach is unfortunately not suitable to our study.*

*References:*

- *Conrad, Quantification of methanogenic pathways using stable carbon isotopic signatures: a review and a proposal, Organic Geochemistry, Volume 36, Issue 5, 2005, Pages 739-752, ISSN 0146-6380, https://doi.org/10.1016/j.orggeochem.2004.09.006.*

L567-568: Rather than saying "maximum ecosystem respiration in the mature bog was found…", better to say "Ecosystem respiration rates were elevated from June to August, and decreased in September."

*We will change this line to "Ecosystem respiration rates were elevated from June to August, and decreased in September", as suggested.*

L570-577: There are some grammatical mistakes and misuse of punctuation that make this difficult to read.

*L570-L577 will be corrected to read as: "...emissions (sum of CH4 and CO2 emissions) released as CH4 were an order or magnitude greater in the young bog than in the mature bog stage, at 18 and 2% respectively. This resulted from both the young bog's higher CH4 emissions and lower ecosystem respiration (Figure S3). The δ13C-CH4 signature of CH4 emissions (intercept values from Keeling plots) in the young bog were significantly greater than those observed in the mature bog (Figure 3c; F (1, 4) = 20.67, P < 0.05), suggesting a greater influence of acetoclastic CH4 production. The average isotopic signature in young bog CH4 emissions (n = 4) was -66.5 ± 1.4‰ (Figure 3c), whereas the average from mature bog emissions (n = 4) was -78.5 ± 5.6‰ (95% CI)."*

L674-678: If these variables only explain 18.4 and 4.3% of methanogenic community structure variation, then what other variables are important here? It seems like the analysis needs to go farther/data are inconclusive.

*It is important to note that these variables are significant in influencing methanogenic community structure as determined by our backward stepping model, As such these two variables (distance to water table and thaw stage) were only mentioned because these were most important and relevant in determining methanogenic community structure, utilizing our backward stepping model. The analysis would not be as statistically robust if we were to include other, non-significant variables that were used as part of this model (i.e., DOC, temperature, enzymatic activity estimate depth, and etc, as described in L453 of the methods). We will add a line to mention that the remaining variation may be constrained by these non-significant parameters, and likely others not measured here. We have added the caveat that microbial community structure cannot always be fully explained by the discrete set of environmental parameters measured in any one study, and unconstrained variation may be further explained by plant-microbe and individual microbe-microbe interactions that we did not quantify (Boon et al., 2014).*

*Reference:*
- *Boon, E., Meehan, C. J., Whidden, C., Wong, D. H., Langille, M. G., & Beiko, R. G. (2014). Interactions in the microbiome: communities of organisms and communities of*

*genes. FEMS microbiology reviews, 38(1), 90–118. https://doi.org/10.1111/1574-6976.12035*

L709-710: Would be worth mentioning that 14C measurements of CO2 and CH4 would help answer the question of whether the emissions are derived from decomposition of fresh, labile DOM or old, previously frozen peat.

*We will add in text citing previous 14C-CO2 work that was performed at the study site (Estop-Aragonés et al 2018), which showed little to no evidence of aged carbon contributing to surface CO2 emissions. Other studies at similar thermokarst bog sites in western Canada have also found little to no evidence of aged carbon contributing to CH4 emissions at the surface (Cooper et al 2017) which we will also include in the text.*
*References*

- *Cooper, M. D. A., Estop-Aragonés, C., Fisher, J. P., Thierry, A., Garnett, M. H., Charman, D. J., et al. (2017). Limited contribution of permafrost carbon to methane release from thawing peatlands. Nature Climate Change, 7(7), 507–511. https://doi.org/10.1038/nclimate3328*
- *Estop-Aragonés, C., Czimczik, C. I., Heffernan, L., Gibson, C., Walker, J. C., Xu, X., & Olefeldt, D. (2018). Respiration of aged soil carbon during fall in permafrost peatlands enhanced by active layer deepening following wildfire but limited following thermokarst. Environmental Research Letters, 13(8). https://doi.org/10.1088/1748-9326/aad5f0*

L718: up "to" the surficial…

*We will change L718 accordingly.*

L720: "drier", rather than "relatively drier." Using the word "drier" already implies a Comparison

*We will change L720 accordingly, to avoid repetition.*

L738: "also been observed"

*We will change L738 accordingly.*

Conclusions:

L853-854: The authors cite a "greater availability of plant leachates" but the DOC and DN data suggest no statistically significant differences in plant leachates between the young and mature bog. This is not a good explanation for the observed shifts in methanogenic communities.

*We will better differentiate how we use our data as well as that from the literature to discuss our results. In this study, we only measured a small suite of DOM parameters and thus, rely upon previous literature at this site and similar locations to provide further support for our interpretations. We have two distinct vegetation communities and surface inundation conditions. Previously, both of these factors have been shown to influence the quality and quantity of plant derived DOM, which in turn has been shown to significantly influence microbial community composition (Laiho, 2003; 2006; Robroek., et al 2016; Bragazza et al., 2015; Ernakovich et al., 2017; Burd et al., 2020). Thus, we believe this to be a logical and appropriate explanation for the differences we observe within the microbial community. We will make these connections clearer in the text.*

*References*
- *Laiho, R. (2006). Decomposition in peatlands: Reconciling seemingly contrasting results on the impacts of lowered water levels. Soil Biology and Biochemistry, 38(8), 2011–2024. https://doi.org/10.1016/j.soilbio.2006.02.017*
- *Laiho, R., Vasander, H., Penttilä, T., & Laine, J. (2003). Dynamics of plant-mediated organic matter and nutrient cycling following water-level drawdown in boreal peatlands. Global Biogeochemical Cycles, 17(2). https://doi.org/10.1029/2002g b002015*
- *Robroek, B. J. M., Albrecht, R. J. H., Hamard, S., Pulgarin, A., Bragazza, L., Buttler, A., & Jassey, V. E. (2016). Peatland vascular plant functional types affect dissolved organic matter chemistry. Plant and Soil, 407(1-2), 135–143. https://doi.org/10.1007/s1110 4-015-2710-3*
- *Bragazza, L., Bardgett, R. D., Mitchell, E. A. D., & Buttler, A. (2015). Linking soil microbial communities to vascular plant abundance along a climate gradient. New Phytologist, 205(3), 1175–1182. https://doi.org/10.1111/nph.13116*
- *Ernakovich, J. G., Lynch, L. M., Brewer, P. E., Calderon, F. J., & Wallenstein, M. D. (2017). Redox and temperature-sensitive changes in microbial communities and soil chemistry dictate greenhouse gas loss from thawed permafrost. Biogeochemistry, 134(1–2), 183–200. https://doi.org/10.1007/s1053 3-017-0354-5*
- *Burd, K., Estop-Aragonés, C., Tank, S. E., & Olefeldt, D. (2020). Lability of dissolved organic carbon from boreal peatlands: Interactions between permafrost thaw, wildfire, and season. Canadian Journal of Soil Science, 13(February), 503–515. https://doi.org/10.1139/cjss-2019-0154*

L861: lower temperatures in the mature bog? I assume that the water in the young bog helps to absorb more heat, but this should be made clear in the manuscript.

*We have soil temperature measurements from the site (Figure S1) to show this and provide results from L483-491 on soil temperatures. We also highlight how the thermal properties of the drier surface-peat reduce the temperature in the mature bog (L723).*

In the conclusions, it would be useful to mention the relative contributions of acetoclastic and hydrogenotrophic methanogenesis to methane emissions in the two bogs.

*This has been addressed above (comment for lines 553-554).*

Figures:

Figure 1: would be helpful to label photos d and e with "mature" and "young"

*We will add these labels to enable easier interpretation of the plots.*

Figure 2: You need a legend showing which colors represent young vs. mature bog. It is better practice to not rely on explanation in the figure caption, but to give the reader essential information in the figure itself. I would suggest using a different shape for the data points for one of the bogs too.

*We will add a legend to this figure in panel Figure 2b.*

Figure 2(a): It would be helpful to write "CH4 concentrations" and "CO2 concentrations" or use concentration notation ([CO2], [CH4]). I also don't understand the arrows. Why are they pointing to specific points? Don't these represent the top of the permafrost/bottom of active layer? It would make more sense to have additional horizontal lines rather than arrows, unless that looks too busy with the water table levels.

*We will change to Dissolved $CH_4$/$CO_2$. The unit ($\mu mol\ L^{-1}$) demonstrates that these are concentrations.*

*The arrows are explained in the figure caption on L523-525; they indicate the thaw transition depth at both sites. We tried using horizontal lines to indicate this, however it made the figure too crowded and messy. We do not focus too much on this transition within our results and discussion, but rather, the water table position as it is more important to our study. Thus, the water table position is prominent in the figure but the thaw transition is not.*

Figure 2(f): To help guide the reader, I suggest using background shading and labels to identify the regions of acetoclastic and hydrogenotrophic methanogesis

*We appreciate this suggestion, and had previously tried this, but ultimately, the shading made the figure too busy and unclear. Because the range of $\alpha_C$ values for acetoclastic and hydrogenotrophic methanogenesis overlap, the shading becomes difficult. We instead provide a label for each line within the figure and the range associated with each pathway in the text. This is in the same format as Hornibrook et al (1997, 2000).*

*References*
- *Hornibrook, E. R. C., Longstaffe, F. J., & Fyfe, W. S. (1997). Spatial distribution of microbial methane production pathways in temperate zone wetland soils: Stable carbon and hydrogen isotope evidence. Geochimica et Cosmochimica Acta, 61(4), 745–753.* [https://doi.org/https://doi.org/10.1016/S0016-7037(96)00368-7](https://doi.org/https://doi.org/10.1016/S0016-7037(96)00368-7)
- *Hornibrook, E. R. C., Longstaffe, F. J., & Fyfe, W. S. (2000). Evolution of stable carbon isotope compositions for methane and carbon dioxide in freshwater wetlands and other anaerobic environments. Geochimica et Cosmochimica Acta, 64(6).* [https://doi.org/10.1016/S0016-7037(99)00321-X](https://doi.org/10.1016/S0016-7037(99)00321-X)

Figure 4: I like this figure, but it could be arranged differently to take up less space on the page – the circles and triangle legend could go below the color bars, or the entire legend could go on top of the plot and the text can wrap around the right side of the figure.

*We will edit this figure, as suggested.*

Figure 6: In the text, you say that you assess only thaw stage and distance to water table, but in the caption you say that you explore both biotic and abiotic factors. What else was included in this analysis that is not shown in the plot that seemingly explains more of the methanogenic community variation?

*We describe the other factors that went into this analysis in the methods section at L453 of the methods. We will reference this section in the figure caption.*

References cited above:

H  j, L., Olsen, R. & Torsvik, V. Effects of temperature on the diversity and community structure of known methanogenic groups and other archaea in high Arctic peat. ISME J 2, 37–48 (2008). https://doi.org/10.1038/ismej.2007.84
Throckmorton, H.M., Heikoop, J.M., Newman, B.D., Altmann, G.L., Conrad, M.S., Muss, J.D., Perkins, G.B., Smith, L.J., Torn, M.S., Wullschleger, S.D. and Wilson, C.J., 2015. Pathways and transformations of dissolved methane and dissolved inorganic carbon in Arctic tundra watersheds: Evidence from analysis of stable isotopes. Global Biogeochemical Cycles, 29(11), pp.1893-1910.
Data show a temporal shift in methanogenesis pathways, from acetoclastic in July to hydrogenotrophic in September.
Hultman, J., Waldrop, M., Mackelprang, R. et al. Multi-omics of permafrost, active layer and thermokarst bog soil microbiomes. Nature 521, 208–212 (2015). https://doi.org/10.1038/nature14238
Active layer communities expressed genes and proteins involved in obtaining energy and nutrients from a diversity of aerobic and anaerobic processes and were equipped with functions for survival under freeze–thaw conditions. The bog represented a different scenario with a very high measured rate of methanogenesis and correspondingly high relative abundances of genes, transcripts and proteins involved in methanogenesis, thus demonstrating the potential linkage between molecular data and ecosystem level process rates

---

## Author Comment (AC2)

Review on bg-2021-337
Anonymous Referee #2

Referee comment on "High peatland methane emissions following permafrost thaw: enhanced acetoclastic methanogenesis during early successional stages" by Liam Heffernan et al., Biogeosciences Discuss., https://doi.org/10.5194/bg-2021-337-RC2, 2022

The manuscript by Heffernan et al. looks at the effect of permafrost thaw on methane emission, pathways of methanogenesis and microbial community. They compare depth profiles of young and mature thermokarst bogs and the uncollapsed plateau. Based on isotope values of methane and methanogenic archaeal community composition it is concluded that acetoclastic methanogenesis is more important in the young bog with higher methane emission than in the mature bog.

The major strength of the manuscript of the manuscript is the multifaceted approach: CH4 and CO2 emissions during the whole growing season, isotope values of the gases, depth profiles of dissolved gases, depth profiles microbial communities in peat and porewater at two time points. These all help build a thorough picture of the large methane emission during thermokast formation where changes through the growing season and the peat profile are taken into account, together with the microbial successional dynamics. The manuscript is easy to read ang the figures are clear. I especially like Figure 1 on the experimental setup that shows well both the horizontal and vertical aspects of the sampling setup.

*Thank you very much for your comments and feedback!*

A potential weakness of the study is that the conclusions of the microbial community analysis focus on methanogens, but the analysis was carried out by primers that amplify both bacteria and archaea. This means that archaea and further methanogens form only a small fraction of the sequence reads. However, the read numbers and the proportion of archaea and methanogens in the dataset are reported well and suggest that there is on average around 900 methanogen reads per sample (I hope I got this right), which should be sufficient to cover methanogen diversity.

*Yes, we used universal primers, targeting both archaea and bacteria. We made this choice in order to enable exploration of both the bacterial and archaeal populations before narrowing our focus on the methanogenic community for this manuscript. Because archaea are still adequately captured in our dataset, as the reviewer points out, on (average 1021 methanogen-related reads were captured per sample), we believe that our approach is sufficient for covering methanogen diversity. We will add these values to the methods section.*

Major comments:
1. Based on Fig. S2, Methanosarcinales/Methanosarcinaceae/Methanosarcina were defined as acetoclastic methanogens Please clarify the basis of this definition. Methanosarcinales contains methanogens that can use acetate, $H_2+CO_2$ and methylated compounds. Even within genus Methanosarcina, not all species use acetate (Kendall & Boone 2006 https://doi.org/10.1007/0-387-30743-5_12). The family Methanotrichaceae consists of obligate acetoclastic methanogens, but based on Fig. S2 they were not detected?

*Yes, as the reviewer points out, members of the Methanosarcinales do indeed perform multiple kinds of methanogenesis. We labelled them here as "acetoclastic" since they were only methanogenic members that we detected that were associated with acetoclastic methanogenesis. We did not detect Methanotrichaceae in our samples, using either SILVA or Greengenes (see response below re: SILVA). However, we agree with the reviewer that our labelling of Methanosarcinales as solely acetoclastic is mis-leading. We will re-label these as acetoclastic / hydrogenotrophic in Fig S2.*

*Reference:*

*Kendall M.M., Boone D.R. (2006) The Order Methanosarcinales. In: Dworkin M., Falkow S., Rosenberg E., Schleifer KH., Stackebrandt E. (eds) The Prokaryotes. Springer, New York, NY. https://doi.org/10.1007/0-387-30743-5_12*

2. Do I understand correctly that the microbial analyses were based on one peat core per site per sampling month (so no replication within sampling month)? I understand that in such a multifaceted study it is not possible to cover everything perfectly, but how is it possible to test the effect of sampling month (L620-621) without replication?

*Yes, correct. We only had one peat core per type of peat (YB, MB, peat plateau) per sampling month as well as per depth. We combined all samples between months together (i.e., all samples from YB, MB and peat plateau in June vs all samples from YB, MB and peat plateau in September) for statistical analyses because, utilizing a PERMANOVA, we found that these samples were not statistically significant between sampling months. However, using this approach, we were unable to more robustly confirm if sampling month had a significant impact on microbial community structure. We will add the caveat that we did not have replicate samples to test the robustness of this finding, and as such, additional study, with more samples, is necessary to verify this result.*

Specific comments:
L84-85 Please clarify how the statement that two-thirds of CH4 comes from acetoclastic methanogenesis applies to peatlands. As far as I understand, Conrad 1999 is a general prediction, and Kotsyurbenko et al. 2007 cites several references to say most of methane in peatlands and even 100% comes from hydrogenotrophic methanogenesis?

*Thank you for catching this! We will modify the text to: "according to a study conducted by Oremland in 1988, two thirds of methane produced in natural systems is attributed to acetoclastic methanogenesis. However, northern peatlands in particular have shown that the acetoclastic methanogenesis pathway is less relevant in producing $CH_4$ (Rooney-Varga et al., 2007), except for minerotrophic fens, dominated by vegetation such as Carex sp, where, especially in the upper layers of peat, acetoclastic methanogenesis dominates (Galand et al., 2005)".*

*References:*

*Oremland R.S (1988) Biogeochemistry of methanogenic bacteria. Biology of Anaerobic Microorganisms (Zehnder AJB, ed), pp. 641–705. John Wiley, New York.*

*Galand, P. E., Fritze, H., Conrad, R., & Yrjälä, K. (2005). Pathways for methanogenesis and diversity of methanogenic archaea in three boreal peatland ecosystems. Applied and environmental microbiology, 71(4), 2195–2198. https://doi.org/10.1128/AEM.71.4.2195-2198.2005*

*Juliette N. Rooney-Varga, Michael W. Giewat, Khrystyne N. Duddleston, Jeffrey P. Chanton, Mark E. Hines, Links between archaeal community structure, vegetation type and methanogenic pathway in Alaskan peatlands, FEMS Microbiology Ecology, Volume 60, Issue 2, May 2007, Pages 240–251, https://doi.org/10.1111/j.1574-6941.2007.00278.x*

L410-411 I think the Greengenes database hasn't been updated for a very long time? This might not be a big problem because methanogen nomenclature has not changed that much recently. However, I am still left wondering if using a newer reference database would have improved the taxonomic affiliations (for example by providing more detailed affiliations or affiliations to unidentified OTUs).

*The reviewer is correct, the Greengenes database hasn't been updated since May 2013, however we also used the SILVA database to assign taxonomy to our ASVs and found that both SILVA and Greengenes captured a similar number of archaea (total of 51187 methanogenic read counts attributed to SILVA vs 51141 methanogenic read counts attributed to Greengenes). We will add these lines to the methods section as well. Also, the taxonomic resolution between both databases was also similar, identifying the same kinds of phyla, families and genus, and methanogens (i.e. methanoregula, methanosarcinales, etc,..) Given, these similarities, and the fact that methanogen nomenclature has not changed significantly as the reviewer points out, we ultimately chose to use Greengenes because it was able to resolve more methanogenic families belonging to methanocelalles and Methanomassiliicoccaceae compared to SILVA. We also note that the Greengenes database is still commonly used to explore methanogenic archaeal communities (Vanwonterghem et al., 2016, Lin et al., 2017, Carson et al., 2019).*

*References*

- *Lin, Y., Liu, D., Yuan, J., Ye, G,m Ding, W. (2017). Methanogenic community was stable in two contrasting freshwater marshes exposed to elevated atmospheric CO2. Front Microbiol.  https://doi.org/10.3389/fmicb.2017.00932*

- *Michael A Carson, Suzanna Bräuer, Nathan Basiliko, Enrichment of peat yields novel methanogens: approaches for obtaining uncultured organisms in the age of rapid sequencing, FEMS Microbiology Ecology, Volume 95, Issue 2, February 2019, fiz001, https://doi.org/10.1093/femsec/fiz001*

- *Vanwonterghem, I., Evans, P., Parks, D. et al. Methylotrophic methanogenesis discovered in the archaeal phylum Verstraetearchaeota. Nat Microbiol 1, 16170 (2016). https://doi.org/10.1038/nmicrobiol.2016.170*

L600, L606, L611: Are these PERMANOVA results or ANOSIM results? In the methods only ANOSIM is mentioned (L444), and L617 and L621 mentions ANOSIM instead of PERMANOVA. Were both ANOSIM and PERMANOVA used and why? PERMANOVA should be the more robust alternative (see vegan documentation). Please also give the R or R2 values for PERMANOVA/ANOSIM results in addition to p values to give the reader an idea on the magnitude of the difference.

*We used both ANOSIM and PERMANOVA as a method to test significance, since they are similar analyses (although one is more robust, as the reviewer points out). The fact that PERMANOVA was left out of the methods was an oversight. All statistical tests using ANOSIM will be converted to PERMANOVA, with the corresponding R2 reported. Furthermore, the results from the PERMANOVA matched those from the ANOSIM test, and so our conclusions do remain unchanged.*
s
L611 Figure S2b is cited here but Fig. S2 has no a or b panels?

*Thank you for catching this typo! Figure S2 is a standalone figure with no a) or b) panels. The text will be modified to reflect this change (referring to Figure S2 rather than S2b).*

L620-621, L693 Microbial community diversity -> microbial community composition or microbial community structure (because 'diversity' often refers to alpha diversity).

*We will change the corresponding lines to "microbial community structure / microbial community composition" as suggested, to avoid referral towards alpha diversity.*

L718 Check missing letter in 'up t the'.

*Thank you for catching this typo! This text will be edited to "…despite similar peat stratigraphy amongst the surficial vegetation…"*

---

## Author Comment (AC3)

Reviewer 3 Summary:

The objectives of this study were to assess the impacts of time following permafrost thaw stage on methane emissions and methanogenic community composition. To do this, the authors identified two bog sites with permafrost that thawed 30 and 200 years ago. Analyses conducted at these sites included (1) metagenomic assessments, (2) dissolved gas concentrations (CO2 and CH4), (3) surface emissions,(4) d13C signatures for both CH4 and CO2 (used to assess the relative contribution of acetoclastic methanogenesis to total methanogenesis).

Overall, this paper effectively approaches that goal.

I have two primary concerns:
(1) My main concern with this paper is centered around the use of isotope d13C signatures. The alpha value is an accepted method for discerning the relative contributions of acetoclastic vs. hydrogenotrophic methanogenesis. While the authors used alpha values for dissolved gas analysis, they limited their assessments of acetoclastic input to 13C signatures of methane (without concomitant 13C-CO2 signatures). Is the 13C-CH4 signature on its own a sufficient indicator of acetoclastic contribution? If so, please provide citations that indicate so.

*We agree that the presentation of the isotope data can be clearer and more consistent throughout. We do not calculate alpha values for our fluxes as a significant proportion of the $\delta^{13}C\text{-}CO_2$ signature will be heavily influenced by autotrophic respiration. This would significantly influence the alpha values and lead to a bias towards acetoclastic methanogenesis, thus we do not calculate it. In the results section and throughout the manuscript we will change how we present the isotope data. The new format we will follow will be to first present the alpha vale with the delta values in parentheses. We will keep Figure 3c the same, showing results from the Keeling plot method as this is a commonly used approach to assess the processes involved in determining the isotopic composition of atmospheric $CH_4$. (Keeling, 1958). During methanogenesis, fractionation by hydrogenotrophic and acetoclastic methanogens produces $CH_4$ with a $\delta^{13}C\text{-}CH_4$ of -110‰ to -60‰ and -65‰ to -50‰, respectively (Hornibrook et al., 1997, 2000).*
*Many studies have used these plots, along with the known range of $\delta^{13}C\text{-}CH_4$ signatures associated with the methanogenic pathways, to identify the source signature, some of these are listed below.*

*References*
- *Keeling, C. D. (1958). The concentration and isotopic abundances of atmospheric carbon dioxide in rural areas. Geochimica et Cosmochimica Acta, 13(4). https://doi.org/10.1016/0016-7037(58)90033-4*
- *Hornibrook, E. R. C., Longstaffe, F. J., & Fyfe, W. S. (1997). Spatial distribution of microbial methane production pathways in temperate zone wetland soils: Stable carbon and hydrogen isotope evidence. Geochimica et Cosmochimica Acta, 61(4), 745–753. https://doi.org/https://doi.org/10.1016/S0016-7037(96)00368-7*
- *Hornibrook, E. R. C., Longstaffe, F. J., & Fyfe, W. S. (2000). Evolution of stable carbon isotope compositions for methane and carbon dioxide in freshwater wetlands and other anaerobic environments. Geochimica et Cosmochimica Acta, 64(6). https://doi.org/10.1016/S0016-7037(99)00321-X*

*Studies using the keeling plot method to identify $CH_4$ source*
- *Fisher, R. E., et al. (2017), Measurement of the 13C isotopic signature of methane emissions from northern European wetlands, Global Biogeochem. Cycles, 31, 605– 623, doi:10.1002/2016GB005504*
- *Marushchak, M. E., Friborg, T., Biasi, C., Herbst, M., Johansson, T., Kiepe, I., Liimatainen, M., Lind, S. E., Martikainen, P. J., Virtanen, T., Soegaard, H., and Shurpali, N. J.: Methane dynamics in the subarctic tundra: combining stable isotope analyses, plot- and ecosystem-scale flux measurements, Biogeosciences, 13, 597–608, https://doi.org/10.5194/bg-13-597-2016, 2016.*

- *Santoni, G. W., Lee, B. H., Goodrich, J. P., Varner, R. K., Crill, P. M., McManus, J. B., Nelson, D. D., Zahniser, M. S., and Wofsy, S. C. (2012), Mass fluxes and isofluxes of methane (CH4) at a New Hampshire fen measured by a continuous wave quantum cascade laser spectrometer, J. Geophys. Res., 117, D10301, doi:10.1029/2011JD016960.*
- *McCalley, C., Woodcroft, B., Hodgkins, S. et al. Methane dynamics regulated by microbial community response to permafrost thaw. Nature 514, 478–481 (2014). https://doi.org/10.1038/nature13798*
- *S. Sriskantharajah, R. E. Fisher, D. Lowry, T. Aalto, J. Hatakka, M. Aurela, T. Laurila, A. Lohila, E. Kuitunen & E. G. Nisbet (2012) Stable carbon isotope signatures of methane from a Finnish subarctic wetland, Tellus B: Chemical and Physical Meteorology, 64:1, DOI: 10.3402/tellusb.v64i0.18818*

Furthermore—I was not clear on whether the apparent difference in alpha values between mature vs. young sites was indeed statistically significant. This comparison needs to be made explicit. If significant differences between sites can only be found at specific depth intervals, then that should also be stated explicitly.

*We agree, this has not been made clear in the text and will rectify it. The depth profiles of dissolved $CH_4$, dissolved $CO_2$, $\delta^{13}C$-$CH_4$, and $\alpha_C$ are significantly different between the young and mature bog. We will clearly state that these depth profiles in the young and mature bog are distinct from one another. Within our analysis, we control for depth as we are not interested in differences at specific depths, but rather how the depth profiles overall differ between the thawed sites. Direct comparison of specific depths between the two sites can be complicated and misleading as depths in the young and mature bog do not correspond to one another with regard to depth from the water table, age, time spent since thaw occurred, and peat composition. Thus, we are interested in how these entire depth profiles differ between thaw sites. We can split the depth profiles into peat that accumulated before and after the most recent thaw event at the site (the depths these are found at are indicated by arrows in Figure 1a). Similar to the entire depth profile, dissolved $CO_2$, $\delta^{13}C$-$CH_4$, and $\alpha_C$ are significantly different between the young and mature bog for peat that accumulated before and after the most recent thaw event. This additional information and statistical analysis will be added to the text.*

(2) Your goal was to examine the effects of thaw stage on methane fluxes/methanogenic community composition. It is difficult to wrap my head around this goal since thaw succession causes shifts in so many different environmental factors (soil temperature, thickness of the unsaturated peat column, availability of labile organics). This makes your results difficult to build off of/apply to other settings. Perhaps you could perform an ordinary least squares regression analysis (OLS) to try to tease apart the relative influence of these numerous factors on a dependent variable of interest (perhaps total methane emissions, or acetoclastic methane emissions).

*We agree that peatland succession following permafrost thaw presents a dynamic, complex landscape that may influence microbial community composition and its activity in a myriad of ways. Here, we present data from two areas that have thawed 30 and 200 years ago. Each site has a distinct water table position, vegetation community, soil temperatures, and volume of peat accumulated at the surface following thaw. Each site does however have identical histories in the peat layers that accumulated prior to permafrost thaw (Heffernan et al., 2020). The objective of this study was to address how the combined effect of the ecological conditions (exposure of previously frozen peat, water table position, vegetation community, soil temperatures) found in the decades following thaw (young bog) and those found centuries following thaw (mature bog) influences the soil methanogen community, its activity, and the resulting $CH_4$ fluxes to the atmosphere. We found that no single factor drives differences, but rather it is the overall ecological conditions and interactions between these and microbial community members that influences the microbial community structure, its activity, pathways of methanogenesis, and $CH_4$ surface fluxes. .*

*We explored potential relationships between environmental factors and methanogen community composition using a distance-based redundancy analysis (RDA; Figure 6). Included in our RDA was an initial backward stepwise regression (L453-457) to determine what environmental factors were significantly influencing the methanogenic community and should be included in our RDA. These included dissolved concentrations of*

*$CO_2$, $CH_4$, DOC, temperature, enzymatic activity estimate, thaw stage, depth, and distance to water table. This stepwise regression serves a similar purpose to the prosed ordinary least squares and allows us use redundancy analysis once significant variables have been identified.*

*While we could use the environmental data at each site (water table depth, soil temperatures, time since thaw) to model our total $CH_4$ emissions, and in doing so determine the main drivers of our $CH_4$ emissions over a growing season, this was not the objective of this study. This study aimed to relate the methanogen community to surface $CH_4$ fluxes, and to comment on how long elevated $CH_4$ emissions may persist following thaw. The 2018 growing season data presented in this study is being prepared, along with multiple other years, in a separate study to achieve this.*

*References*
- *Heffernan, L., Estop-Aragonés, C., Knorr, K.-H., Talbot, J., & Olefeldt, D. (2020). Long-term impacts of permafrost thaw on carbon storage in peatlands: deep losses offset by surficial accumulation. Journal of Geophysical Research: Biogeosciences, 2011(2865), e2019JG005501. https://doi.org/10.1029/2019JG005501*

Specific Questions/Recommendations

(1) I am unclear on what is mean by the term "ecological" in the context of this manuscript (e.g. L27-28). I get the impression that it references vegetation primarily. I am unsure about that, however, because "ecological" could also be used to describe microbial community composition. Please clarify.

*By 'ecological' we mean the shifts associated with autogenic ecological succession seen following thaw in thermokarst bogs, this includes vegetation community, water table position, and temperature. We will be more explicit with this and define what we mean by ecological shifts in the introduction.*

(2) Fig 1: I'd recommend explicitly stating how far apart the mature and young bog sites are from one another in panel f (or the caption). You list only one GPS coordinate from the whole site in the figure caption.

*Figure 1c includes a scale of 10 m to indicate this distance and we have also roughly defined the distance between these sites in the methods by stating how far they are from the plateau (young bog – L192 and mature bog – L202). We do not have a single exact measurement of the distance between these as cores and dissolved gas depth profiles were taken within specific areas (circles in Figure 1c) rather than a single location.*

(3) L158-159: How did you determine that this complex is representative? If the succeeding sentences are meant to serve as evidence for this claim, make that explicit.

*This is based on other studies of peatland complexes within the discontinuous zone in the Interior Plains of western Canada. Some of these are added below and we will add some of these references to the text on L159*

*References*
- *Bauer, I. E., Gignac, L. D., & Vitt, D. H. (2003). Development of a peatland complex in boreal western Canada: Lateral site expansion and local variability in vegetation succession and long-term peat accumulation. Canadian Journal of Botany, 81(8), 833–847. https://doi.org/10.1139/b03-076*
- *Beilman, D. W. (2001). Plant community and diversity change due to localized permafrost dynamics in bogs of western Canada. Canadian Journal of Botany, 79(8), 983–993. https://doi.org/10.1139/cjb-79-8-983*
- *Camill, P. (1999). Peat accumulation and succession following permafrost thaw in the Boreal peatlands of Manitoba, Canada. Ecoscience, 6(4), 592–602. https://doi.org/10.1080/11956860.1999.11682561*

- *Pelletier, N., Talbot, J., Olefeldt, D., Turetsky, M., Blodau, C., Sonnentag, O., & Quinton, W. L. (2017). Influence of Holocene permafrost aggradation and thaw on the paleoecology and carbon storage of a peatland complex in northwestern Canada. Holocene, 27(9), 1391–1405. https://doi.org/10.1177/0959683617693899*
- *Vitt, D. H., Halsey, L. A., & Zoltai, S. C. (1994). The Bog Landforms of Continental Western Canada in Relation to Climate and Permafrost Patterns. Arctic and Alpine Research, 26(1), 1. https://doi.org/10.2307/1551870*
- *Vitt, D. H., Halsey, L. A., Bauer, I. E., & Campbell, C. (2000). Spatial and temporal trends in carbon storage of peatlands of continental western Canada through the Holocene. Canadian Journal of Earth Sciences, 37(5), 683–693. https://doi.org/10.1139/e99-097*
- *Zoltai, S. C. (1972). Palsas and Peat Plateaus in Central Manitoba and Saskatchewan. Canadian Journal of Forest Research, 2(3), 291–302. https://doi.org/10.1139/x72-046*
- *Zoltai, S. C. (1993). Cyclic Development of Permafrost in the Peatlands of Northwestern Alberta, Canada. Arctic and Alpine Research, 25(3), 240. https://doi.org/10.2307/1551820*

(4) L551-554: "Overall, the isotopic data indicates a general dominance of hydrogenotrophic methanogenesis in both sites, but a greater contribution of acetoclastic methanogenesis in the young bog relative to the mature bog." Was this difference statistically significant? I suggest adding a p-value after this statement.

*This statement is intended to cover the entire section of results discussing isotopic data and includes dissolved gas depth profiles of $\delta^{13}C$-$CH_4$, $\delta^{13}C$-$CO_2$, and $\alpha_C$ (L527-552). It summarises these results to close that section, thus we do not perform any statistical analysis or have any p value for it. We can remove the statement if the reviewer deems it unfit for the results section and move it to the discussion. We will make sure to add all relevant p values comparing alpha and delta values in the results section above this to better highlight statistical differences that led to this statement.*

*(5) L 572-575: "The $\delta13C$-$CH4$ signature of $CH4$ emissions (intercept values from Keeling plots), in the young bog were significantly greater than those observed in the mature bog (Figure 3c; $F (1, 4) = 20.67$, $P< 0.05$)., suggesting a greater influence of acetoclastic $CH4$ production."*

Why is there no alpha value for the flux measurements? Please provide a source indicating that 13C-CH4 measurements alone (i.e. without concomitant 13C-CO2) are sufficient to discern the relative influence of acetoclastic methanogenesis on total methane production.

*Please see above where we have addressed why we do not calculate an alpha value for fluxes*

*(6) L704-706: "Evidence of acetoclastic methanogens and $CH4$ produced via the acetoclastic metabolic pathway was found in the young bog both near the surface and at depths below the thaw transition (i.e., in peat that accumulated prior to permafrost thaw)."*

I have two notes on this:
(1) If the difference in alpha values was not significant in the subsurface (which I am not 100% clear on), this needs to be noted and discussed.

*Please see our response above which addresses this. We will add results from statistical analysis comparing both alpha and delta values from below ground dissolved gas samples to show that results from the peat layer that accumulated after thaw are distinct between the young and mature bog.*

(2) See my comments regarding L572-575. Make sure your methods for discerning the acetoclastic influence on surface CH4 emissions is sound.

*Please see above where we have addressed this.*

*(7)* L765-766: "*The presence of hydrophilic vegetation, particularly graminoids, in the saturated young bog provides the precursors for fermentation..*"

I am confused by this statement. "Precursors" could be interpreted as "reactants", which are primarily sugars. Sugars are ultimately delivered to porewater from other plants too (i.e. Sphagnum spp.). Are you saying that the sugars derived from gramminoids are more labile than those derived from Sphagnum? I would agree with this, but it is necessary to clarify.

*Yes, this is exactly what we mean. The references below (Ström et al., 2003; 2012) show how graminoids enhance substrate quality and availability, leading to greater methanogenesis. We will add these citations to the text. We use precursors as a catchall term for all plant derived monomeric compounds formed following extracellular enzyme hydrolysis used in this fermentation step. Precursor is the most suitable term for these, as it is defined by Merriam-Webster dictionary, "a substance, cell, or cellular component from which another substance, cell, or cellular component is formed"*

*References*
*Ström et al., Presence of Eriophorum scheuchzeri enhances substrate availability and methane emission in an Arctic wetland, Soil Biology and Biochemistry, Volume 45, 2012, Pages 61-70, ISSN 0038-0717, https://doi.org/10.1016/j.soilbio.2011.09.005.*
*Ström, L., Ekberg, A., Mastepanov, M. and Røjle Christensen, T. (2003), The effect of vascular plants on carbon turnover and methane emissions from a tundra wetland. Global Change Biology, 9: 1185-1192. https://doi.org/10.1046/j.1365-2486.2003.00655.x*

*(8)* L805-813: I find the thread of this paragraph hard to follow. Please make the connections between sentences clearer.

*These sentences will be restructured to better highlight the message we are trying to convey with them. This section attempts to summarise that there are multiple factors influencing the observed differences in the $\delta^{13}C$-$CH_4$ signature of $CH_4$ emissions between the young and mature bog. This will be edited to make this clearer and will now read*

*"However, increased oxidation above the water table in the mature bog is likely not fully responsible for the observed differences in CH4 surface emissions and depth profiles between the young and mature bog. Lower soil temperatures, a vegetation community associated with reduced substrate availability, the dominance of hydrogenotrophic methanogenesis throughout the peat profile, and a deeper water table position all contribute to the lower CH4 production and higher CH4 oxidation observed in the mature bog.*

*(9)* Fig 2: Please add in a legend.
*We have added a legend to this figure in panel Figure 2b*

---

## Author Response (AR1)

*Here we provide our response to each reviewer that includes a point-by-point response to each review. We would like to thank each reviewer for their time and consideration in the review of our manuscript. We believe that after this review the manuscript is much improved and we hope the reviewers find it to be so.*

*The original review of each reviewer is in black text, our response is in blue text, and the changes we have made to the manuscript are shown in blue, bold, and italicized text.*

Response to reviewer 1 (RC1)

Review of "High peatland methane emissions following permafrost thaw: enhanced acetoclastic methanogenesis during early successional stages" by Liam Heffernan and others.

Summary:

The goal of this manuscript is to advance our understanding of the underlying controls of methane emissions from permafrost thaw in northern peatlands. Specifically, the authors assess how shifting ecological conditions (e.g., collapse of peat plateau and thermokarst bog formation) affect microbial communities, the amount of $CH_4$ released, and the $d_{13}C$ isotope composition of released $CH_4$. The authors also aim to determine how long elevated surface $CH_4$ emissions persist after thaw.

To answer these questions, the authors study peatland methanogenic community composition and methane emissions along a thaw gradient (intact peat plateau, thermokarst bog formed 30 years ago, and thermokarst bog formed 200 years ago) in discontinuous permafrost in western Canada. The authors analyzed methanogenic community composition down to 160 cm, measured dissolved $CH_4$ and $CO_2$ concentrations and $d_{13}C$ values down to 245 cm in the bogs sites, in addition to rates and $d_{13}C$ values of land-atmosphere $CH_4$ and $CO_2$ fluxes. Results from these analyses show that methanogenesis is primarily hydrogenotrophic, rather than acetoclastic, at both the young and mature sites. Young bog had isotopically heavier methane $d_{13}C$ values than the mature bog, suggesting that acetoclastic methanogenesis was more enhanced in the young bog. Young bog $CH_4$ emissions were 3x greater than the mature bog. These results imply that $CH_4$ emissions by acetoclastic methanogenesis will increase with continued thermokarst peat plateau collapse and thaw depth lowering in discontinuous permafrost over the next century. As thermokarst bogs mature and dry out, lower temperatures and lower substrate availability will lead to a dominance of hydrogenotrophic methanogenesis.

Recommendation:
This is an interesting study that aligns with the research focus of Biogeosciences, but further analysis, clarification, and a more robust discussion are needed before this manuscript can be accepted for publication. Below I describe my overall points of concern, and provide suggestions for the authors to improve the manuscript. I also found that the text needs editing and revision to be more easily understood by the reader, and thus I provide detailed line-byline comments that are more editorial in nature. Therefore, I recommend major revisions.

*Thank you for your feedback! We below respond to your comments and suggestions to better revise our manuscript.*

Major comments:

While the data and analyses reported in this manuscript appear robust and offer insight into the effect of thermokarst peat plateau collapse on greenhouse gas emissions, I feel that the authors have not presented any new ideas or conceptual models that help us relate Arctic landscape change to changes in carbon cycling. I find the discussion and conclusions to be very generalized, attributing the observed differences between young and mature bogs to "hydrological regimes, vegetation communities, and peat chemistry." Statements like this do not provide any insight to the specific mechanisms driving microbial community change.

*With this manuscript, we present the first study to combine microbial and biogeochemical data to assess the influence of permafrost thaw on methanogenesis and $CH_4$ emissions along a space-for-time thermokarst bog transect. We have reviewed and modified the discussion/conclusion to avoid very generalized statements and try to better link the biogeochemical and microbial community data.*

Specifically, it would be useful to quantify the relationship between rate of water table lowering and CO2 and CH4 production rates/magnitudes. I suggest using the ages of the bogs to determine rate of change in environmental parameters, like thaw depth lowering, water table lowering, and temperature change.

*This is an interesting thought; however, we do not think it reasonable to make any further extrapolation on how shifting ecological conditions will impact emissions using this specific dataset, which may be out of the scope of this study. A similar point was also made by RC3, wondering if we could determine how different conditions following thaw may influence total $CH_4$ emissions. To determine the relationship between magnitude of fluxes and site conditions we would need a larger, more comprehensive dataset that consists of either multiple years of data, or multiple sites, or both. This is the objective of a yet to be published study from this site, that includes 3 years of flux data and a secondary site. The objective of this study was to compare sites with different thaw histories, and thus differencing current ecological conditions. We have made the scope and objectives of the study clearer in the introduction to address both reviewers' concerns.*

The data for this study were collected from three very localized sites, and it is not clear whether the processes driving CO2 and CH4 production are representative of the greater Arctic landscape. I think the authors need to use their dataset to dive a little deeper into the mechanisms of methanogenesis and transport to the surface (e.g., Throckmorton et al., 2015). I would also like to know exactly which archaeal communities are most important for greenhouse gas production and how they are changing along the thaw gradient (e.g., H j et al., 2008).

*The study site is considered to be representative of boreal peatlands in the discontinuous permafrost zone in the Mackenzie River Basin of western Canada; see below for some references supporting this. In short, this area is comprised of intact peat plateaus interspersed with*

*permafrost free bogs, fens, and ponds. Permafrost peatlands in this area are very similar to those found in the Hudson Bay Lowlands (Kuhry, 2008) and Alaska (Jones et al., 2017)*

*We do not think there is a single site or ecosystem that is representative of the greater Arctic landscape or northern circumpolar permafrost region. However, our study system does represent a globally significant organic carbon store that is vulnerable to permafrost thaw and potential mineralization into greenhouse gases. Peatlands in the Mackenzie River Basin are one of the three largest stores of organic carbon found in peatlands within the permafrost zone, the other two being the Hudson Bay Lowlands and the West Siberian Lowlands (Hugelius et al., 2020; Olefeldt et al., 2021). Within the sporadic and discontinuous permafrost zone of our study region >15% of the total peat plateau area has thawed and formed thermokarst bogs in the last 30 years (Baltzer et al., 2014; Gibson et al., 2018). Projections for this area suggests total permafrost lost from plateaus by 2050 (Chasmer and Hopkins, 2017). Thus, we consider the results of this study results to be important.*

*To address these comments, we have now added additional text to demonstrate the importance and relevance of these sites for circumpolar north carbon cycling*

*L226– "... similar to those found in the Hudson Bay Lowlands (Kuhry, 2008) and Alaska (Jones et al., 2017)."*

*L238-245– "Peatlands in the Interior Plains in western Canada are one of the three largest stores of organic carbon found in peatlands within the permafrost zone, the other two being the Hudson Bay Lowlands and the West Siberian Lowlands (Hugelius et al., 2020; Olefeldt et al., 2021). Within the sporadic and discontinuous permafrost zone of our study region >15% of the total peat plateau area has thawed and formed thermokarst bogs in the last 30 years (Baltzer et al., 2014; Gibson et al., 2018). Projections for this area suggests total permafrost lost from plateaus by 2050 (Chasmer and Hopkins, 2017)."*

*Regarding the mechanisms of methanogenesis and transport pathways to the surface, initially we considered the mass balance approach used by Thockmorton et al., (2015) as well by Corbett et al., (2013). We agree that this is a very interesting approach to answer questions regarding the pathways of anaerobic fermentation and decomposition, and vertical transport of the end-products of this decomposition. We use a similar approach in determining the pathways of methanogenesis responsible for dissolved concentrations of $CH_4$ at depth to that described in these papers. However, we do not follow a similar approach in assessing the transport of the resulting dissolved gases of anaerobic decomposition. We decided to not take such an approach as it was beyond the scope and objectives of our study. Our study focuses on assessing how shifting ecological conditions following permafrost thaw influence the structure and activity of the methanogen community, the pathways of methanogenesis, and surface $CH_4$ emissions. We do not focus on how dissolved $CH_4$ reaches the surface, or where in the peat profile the $CH_4$ emitted at the surface was produced. Rather, we focus on how methanogenesis and the microbial community responsible for methanogenesis is affected in the top 160 cm of a peatland following permafrost thaw, whether this results in greater surface $CH_4$ emissions, and for how long these surface emissions may last (decades to centuries). We believe that the combination of microbial data (16S) and biogeochemical data (dissolved concentrations, $\delta^{13}C$ signatures, surface*

*emissions) from areas that have thawed 30 and 200 years ago in a thermokarst bog is novel, timely, and interesting.*

*Regarding which archaeal communities are most important for greenhouse gas production and how they are changing along the thaw gradient, we also agree that this is a very interesting and timely question. This question, however, is beyond the scope of our study. Here, we aim to address the influence that shifting ecological conditions, following permafrost thaw, has on methanogenesis specifically, not on anaerobic chemoheterotrophy in general. This indeed would be a fascinating topic for a future study that would include not just 16S data but also various other metaOmics. The dataset for this study is open and freely available, we would be more than happy to discuss the contribution of this data to any such studies in the future.*

The 16S rRNA data appear to be underutilized, whereas the data could be used to test hypotheses presented by other studies (e.g., Hultman et al., 2015).

*While it is true that 16S rRNA gene data may be underutilized in studies such as these, there are numerous constraints on what can be done with (and concluded in using) this kind of microbial taxonomic data in tandem with biogeochemical data. We therefore wanted to limit our interpretation and discussion to the methanogenic community so that we do not "overreach" with what our data could tell us about this system.*

*The hypotheses presented by the study exemplified (Hultman et al., 2015) is a more robust study in that it combines not just 16S microbial taxonomic data, but also metaOmics data such as proteomics, metatranscriptomics and metagenomics to specifically target the functional processes occurring in their system. With our dataset, we can explore the putative metabolisms involved, but with significant limitations, as 16S cannot be directly tied to microbial metabolic function. In an attempt to gain further insight into putative microbial function, we applied FAPROTAX, a bioinformatics tool that can predict ecologically relevant functions from 16S microbial taxonomic data (Louca et al., 2016, Sansupa et al., 2021), to our dataset. However, it was unable to resolve whether particular methanogenic pathways were taking place in different stages of thaw, and thus we chose not include this analysis. Instead, the insight gleaned about the archaeal community composition from the 16S rRNA gene analysis, in conjunction with the isotopic signatures for $CH_4$ and $CO_2$, was ultimately more convincing in identifying the dominant methanogen pathways along our thaw gradient. There is previous precedent to combining 16S rRNA gene and biogeochemical data in a similar fashion to this study to gain insight into changing microbial community structure (Ganzert et al., 2007; Saidi-Mehrabad et al., 2020; Cherbunina et al., 2021), as well as others that incorporate both 16SrRNA gene sequencing and targeted qPCR/metaOmics to more definitively determine more microbially-driven processes in permafrost (Wenal., 2018, Unger et al., 2021).*

*References:*

- *Lars Ganzert, German Jurgens, Uwe Münster, Dirk Wagner, Methanogenic communities in permafrost-affected soils of the Laptev Sea coast, Siberian Arctic, characterized by 16S rRNA gene fingerprints, FEMS Microbiology Ecology, Volume 59, Issue 2, February 2007, Pages 476–488, https://doi.org/10.1111/j.1574-6941.2006.00205.x*

- *Saidi-Mehrabad, A., Neuberger, P., Hajihosseini, M., Froese, D., Lanoil, B.D. (2020). Permafrost microbial community structure changes across the Pleistocene-Holocene Boundary. Front. Environ. Sci. https://doi.org/10.3389/fenvs.2020.00133*

- *Cherbunina, M.Yu., Karaevskaya, E.S., Vasil'chuk, Yu.K., Tananaev, N.I., Smelev, D.G., Budantseva, N.A., Merkel, A.Y., Rakitin, A.L., Mardanov, A.V., Brouchkov, A.V., Bulat, S.A. (2021). Microbial and Geochemical evidence of permafrost formation at Mamontova Gora and Syrdakh, Central Yakutia. Front. Earth. Sci. https://doi.org/10.3389/feart.2021.739365*

- *Wen, X., Unger, V., Jurasinski, G., Koebsch, F., Horn, F., Rehder, G., Sachs, T., Zak, D., Lischeid, G., Knorr, K., Böttcher, M.E., Winkel, M., Bodelier, P.L., & Liebner, S. (2018). Predominance of methanogens over methanotrophs in rewetted fens characterized by high methane emissions. Biogeosciences.*

- *Unger, V., Liebner,S., Koebsch, F., Yang,S., Horn,F., Sachs, T.,Kallmeyer, J., Klaus-Holger, K., Rehder,G., Gottschalk, P., Jurasinski,G. (2021). Congruent changes in microbial community dynamics and ecosystem methane fluxes following natural drought in two restored fens, Soil Biology and Biochemistry 160: https://doi.org/10.1016/j.soilbio.2021.108348.*

The study relies heavily on statistical analysis, but it is not clear that the authors are testing specific hypotheses with their analyses. Further hypothesis testing will help to elucidate some of the other processes driving carbon cycling along the thaw gradient.

*The hypotheses that we aim to answer, are specified in the introduction (Lines 191-193). There, we state that we hypothesize* **"(1) shifting environmental conditions along the permafrost thaw gradient results in a successional microbial community and a restructuring of the methanogenic community, and (2) the warmer conditions and hydrophilic vegetation community in the young bog, along with the exposure of previously frozen peat, will result in a greater relative abundance of acetoclastic methanogens throughout the depth profile, and subsequently greater overall CH4 emissions."**

*To test our first hypothesis, our 16S microbial data was used to test whether there was evidence of distinct groupings of methanogen communities using NMDS and PERMANOVA (L750-757) down to 160 cm depth in areas that have thawed 30 and 200 years ago- the underlined statements are those that have been added to better highlight what test was used to address each hypothesis:*

    **"…to address our first hypothesis, we assessed differences in community composition across both peat and pore water and to determine whether seasonality impacted microbial**

*community structure in both sample matrices. Here, Bray Curtis dissimilarity matrices for overall microbial community data were used, at 999 permutations, to identify distinct groupings assessed at the 95% confidence interval in NMDS ordinations. These distinct groupings were further evaluated for significance using the non-parametric permutational analysis of variance (PERMANOVA) test."*

We then used RDA and variance partitioning (L777-787) to test how biogeochemical and site data from these two different thawed areas influence the 16S data and methanogen community structure:

*"We utilized our methanogenic community data to construct redundancy analyses (RDA) and relative abundance bar plots. RDAs were conducted using a Hellinger-transformed methanogenic community. Explanatory variables (i.e., dissolved concentrations of $CO_2$, $CH_4$, DOC, temperature, enzymatic activity estimate, thaw stage, depth, and distance to water table) were scaled about the mean. These explanatory variables had variance standardized, were checked for collinearity (parameters with variance inflation value > 10 were removed) and selected for significance using backward selection, set at 1,000 permutations. The significance of the RDA model, and of each axis was tested using ANOVAs, set at 999 permutations. Variance partitioning analyses were conducted to assess the contribution of significant environmental parameters (i.e., thaw stage and distance to water table) on the structuring of the Hellinger-transformed methanogenic community…"*

Using our dataset, we unfortunately cannot get more specific than this without further metaOmic data or qPCR data.

To test our second hypothesis, we used ANOVAs and Bonferroni post-hoc tests on linear mixed effects models (L728-739) to test for differences in the concentrations and $\delta^{13}C$ signatures of surface gas fluxes and dissolved gas depth profiles down to 245 cm between the two thawed areas- the underlined statements are those that have been added to better highlight what test was used to address each hypothesis:

*"We used ANOVAs and Bonferroni post-hoc tests on linear mixed effects models to address our second hypothesis and to evaluate significant differences and seasonal trends in greenhouse gas fluxes and dissolved gas depth profiles. We performed these tests to assess whether thaw stage (young bog or mature bog) influenced greenhouse gas fluxes and dissolved gas depth profiles…"*

Could the reviewer be more specific regarding what they mean by testing and how they deem these tests to not be sufficient in addressing our hypotheses? We welcome any suggestions for further hypotheses they would consider testing with the dataset available to us.

I also find that the authors make comparative statements that are not supported by statistically significant differences (e.g., in dissolved chemistry parameters). This section of the results is misleading, and also leads to some misleading interpretations of the data (e.g., L853-854).

*To address this, we have re-analyzed and re-written our results section that presents our pore water chemistry results, the new section (L820-831) reads as*

*"**Across all depths and sampling occasions, average pH was higher (ANOVA: $F_{(1, 77)}$ = 35.2, P < 0.001) in the young bog than in the mature bog at 4.1 ± 0.2 and 3.9 ± 0.2 respectively. In contrast, DOC at 69.2 ± 18.4 and 53.8 ± 5.4 mg C $L^{-1}$ (ANOVA: $F_{(1, 82)}$ = 38.7, P < 0.001) and total dissolved nitrogen at 1.5 ± 1.4 and 0.9 ± 0.1 mg $L^{-1}$ (ANOVA: $F_{(1, 82)}$ = 12.8, P < 0.01) were higher in the mature bog than in the young bog, respectively.  Average SUVA values were higher (ANOVA: $F_{(1, 82)}$ = 103.5, P < 0.001) in the young bog (3.2 ± 0.4 L mg $C^{-1}$ $m^{-1}$) compared to the mature bog (2.6 ± 0.4 L mg C-1 m-1), indicating DOM with a greater aromatic content in the young bog. However, average spectral slope (S250 – 465) values were also greater (ANOVA: $F_{(1, 81)}$ = 6.9, P < 0.05) in the young bog (-0.016 ± 0.002 $nm^{-1}$) compared to the mature bog (-0.017 ± 0.003 $nm^{-1}$), indicating lower molecular weight and decreasing aromaticity. Average phenolics (0.6 ± 0.2 and 0.6 ± 0.2 mg $L^{-1}$) and phosphate ($PO_4^{3-}$: 9.0 ± 14.3 and 6.7± 3.0 µg $L^{-1}$) were similar between the young bog and mature bog, respectively, across all depths and sampling occasions. Full details of DOM chemistry results can be found in Heffernan et al., (2021). Of note is the fact that the pore water chemistry was compared across all depths in this study, in contrast to Heffernan et al., (2021) in which pore water found above and below the transition indicating permafrost thaw was compared.**"*

*We have also added text to the discussion (L1245-1246) to provide a further explanation and references on how vegetation community will impact the microbial community structure through ways we do not measure. This section now reads as:*

*"**At the surface, microbial community structure is influenced by the successional vegetation community (Hodgkins et al., 2014) and the role that vegetation, particularly graminoids which are found in the young bog, has on microbial community structure has been well documented in northern peatlands (Robroek et al., 2015, 2021; Bragazza et al., 2015). Moderately acidic, saturated peatlands with hydrophilic vegetation, similar to the young bog, have been shown to harbour acid tolerant fermenting bacteria that produce substrates for methanogenesis and are trophically linked with methanogens (Wüst et al., 2009). Thus, the interaction between water table position, pH, and vegetation community influences the substrates available to the microbial community, which in turn impacts the surface community's structure (Kotiaho et al., 2013).**"*

*We have also added further text to the discussion to explain the differences in lability associated with the young and mature bog is based on previous work at this study site, but also from work at other closely related sites with similar vegetation communities, and the interaction of these observed in other studies. Again, these are explanations that we need to infer from the literature. This section is found at L1340-1341 and reads as:*

*"While our pore water chemistry data is inconclusive with regards to organic carbon characteristics, other work in thermokarst bogs in the Interior Plains of western Canada has shown that the organic matter derived from the young bog vegetation community is highly labile (Burd et al., 2020). Previous work at our study site has shown that the vegetation community in the young bog is associated with greater potential enzymatic degradation of organic matter (Heffernan et al., 2021). Hydrolysis of plant derived organic matter by extracellular enzymes leads to the formation of monomers (Kotsyurbenko, 2005). These monomers are further degraded to form acetate and other percussors for methanogenesis when present with anaerobic fermenting bacteria (Hamberger et al., 2008) and near the surface and vegetation inputs (Hädrich et al., 2012)."*

Detailed comments:

Abstract

L 32: "(~30 and 200 years since that, respectively)"

*We have changed L32 to read:*
*"~30 and ~200 years since thaw, respectively"*

L 34-35: "high throughput 16S rRNA gene sequencing"

*We have changed L34-35 to read:*
*" high throughput 16S rRNA gene sequencing"...*

L 39-40: It would be helpful to give values or the difference between the mean values of the young vs. mature sites

*We have added these values to the abstract and it now reads as*
*"However, mean δ13C-CH4 signatures of both dissolved gases profiles and surface CH4 emissions were found to be isotopically heavier in the young bog (-63 ‰ and -65 ‰, respectively) compared to the mature bog (-69 ‰ and -75 ‰, respectively), suggesting that acetoclastic methanogenesis was relatively more enhanced throughout the young bog peat profile."*

L 42: It would also be useful to give the measured CH4 fluxes in the abstract

*We have added the rates of CH4 fluxes to the abstract and it now reads as:*
*"Furthermore, mean young bog CH4 emissions of 82 mg CH4 m-2 day-1, were ~ three times greater than the 32 mg CH4 m-2 day-1, observed in the mature bog."*

L43-45: Be more specific on what the interactions are. I assume that different interactions between ecological conditions and methanogen communities can also reduce CH4 emissions. What exactly are favorable conditions for methanogens and what is the implication for future CH4 emissions as these thermokarst bogs continue to age and turn more hydrogenotrophic?

*We added more specific lines regarding what interactions we are talking about here (Lines 44-48). Namely, we have made the below changes:*
**"Our study suggests that interactions between the methanogenic community and hydrophilic vegetation, warmer temperatures, and saturated surface conditions enhance CH4 emissions in young thermokarst bogs, but that these favorable conditions only persist for the initial decades after permafrost thaw."**

Introduction

L51: "…are thought to be driven by…"

*We have changed L51 (now L88) to read:*
*"are thought to be driven by"*

L 64: Can thermokarst formation also expose frozen C to aerobic microbial decomposition(e.g., to CO2)? Do we know whether aerobic or anaerobic decomposition result in greater greenhouse emissions? What about the role of methane oxidation by aerobic bacteria or anaerobic archaea (e.g., In't Zandt et al., 2020)

*Yes, thermokarst formation can expose previously frozen organic matter to aerobic respiration, resulting in increased $CO_2$ emissions (Schädel et al., 2016). However, thermokarst formation in peatlands is characterized by ground subsidence, resulting in saturated surface conditions (Camill, 1999). These saturated surface conditions result in previously frozen peat being exposed to anoxic conditions. Peatlands are a wetland ecosystem where anoxia is the dominant redox condition and anaerobic decomposition is the main form of decomposition.*

*In general, aerobic respiration occurs at a faster rate than anaerobic respiration as has been shown for other, non-peatland, thermokarst ecosystems (Schädel et al., 2016). We will add some text to reflect this in the introduction. This new text now reads*
**"Redox conditions following thermokarst formation are an important control of decomposition, with 3-4 times as C mineralization occurring as aerobic respiration compared to anaerobic respiration (Schädel et al., 2016)"** *(L102-104)*

*We address $CH_4$ oxidation in the manuscript (L905-909). Our study objectives were not to explore the relationship been ecological conditions following thaw and aerobic respiration at the surface in peatlands, but rather the anaerobic processes beneath the water table. The In't Zandt et al., 2020 study focuses on thermokarst lakes in ice-rich Yedoma deposits. While the role of $CH_4$ oxidation within anaerobic lake sediments in these systems is an interesting one, we do not think it entirely relevant to our study. In thermokarst affected permafrost peatlands, $CH_4$ oxidation has been shown to be closely linked with redox potential associated with the water table position (Perryman et al., 2020). This is included in our discussion of oxidation in the manuscript.*

L133: Does colonization cause fresh, labile inputs of carbon here? I'm not sure what specific process is increasing the amount and temperature sensitivity of CH2 emissions.

*Yes, colonization of hydrophilic vegetation following thaw is associated with an increase in labile inputs. This increase in labile inputs can increase methanogenesis, and thus, the sensitivity of methanogenesis to temperature. We cite references in the text (L193;1218) that address this and shaped this hypothesis.*

L130: Can you be more specific about the "shifting ecological conditions"? Only be more specific if your results allow you to link methane emissions to specific conditions.

*We have changed "ecological conditions" to "environmental conditions" throughout the manuscript and have provided our definition of environmental conditions on (L187-189) which reads as*
**"Thermokarst formation has resulted in distinct environmental conditions at each stage along this thaw gradient that we herein define as water table position and surface wetness, soil temperatures, and vegetation community."**

L197: "…is drier than the young bog, with …"

*We have changed this to*
**"is drier than"**

L217: "…young and mature bog stages, ~1 m from the nearest collar."

*We are unsure about what change is suggested, the suggestion is the same as the current text.*

L219: remove "deep"

*We have removed "**deep**" from this line.*

L221: "…devices were installed in each bog…" (since there are only two bogs, you don't need to
keep repeating "young and mature bogs")

*We have removed this repetition as suggested. This has been changed to*
**"both thermokarst bog stages"**

L221-222: "…where two dissolved gas samples were collected, two from 5-95 cm depth and a third from 115-245 cm depth."

*We have changed this to*
**"Three diffusive gas sampling devices were installed in each thermokarst bog stage, where two collected dissolved soil gas samples from 5 – 95 cm deep and a third from 115 – 245 cm."**

In general, the writing in section 2.2 needs to be improved to make the methods more clear for the reader.

*This section has been re-written and now reads as*

*"The Lutose peatland study site was established in 2015 and a boardwalk was constructed to minimize disturbances along the peat plateau - thermokarst bog transect. Three collars for surface greenhouse gas flux (39 cm diameter) measurements were permanently installed to a depth of 20 cm in both the young and mature thermokarst bog stages. The top of each collar was aligned with the peat surface. PVC wells (2 cm diameter) were installed directly next to each collar and were used to manually monitor the water table position during each gas flux measurement. We monitored soil temperature (°C) at 10, 30, 50, 75, 100, 150, 200, and 250 cm every 30 min from May – September 2018 using permanently installed loggers (Hobo 8k Pendant Onset Computer, Bourne, MA, USA) in both thermokarst bog stages. Temperature depth profiles were established centrally among collars in each thermokarst bog stage, in areas that had similar vegetation, water table position, and distance from the thawing edge as the collars.*

*Custom made plexiglass pore water suction (Heffernan et al., 2021) and diffusive equilibration gas sampling devices (Knorr et al., 2009) were installed in July 2016 in the young and mature bog. These devices were installed in both thermokarst bog stages ~1 m from the nearest flux measurement collar. Pore water suction devices were installed to a depth of 160 cm and consisted of 15 sampling depths, with each sampling depth connected to the surface via silicone tubing. This allowed for repeated non-destructive pore water sampling. Three diffusive gas sampling devices were installed in each thermokarst bog stage, where two collected dissolved soil gas samples from 5 – 95 cm deep and a third from 115 – 245 cm. Each diffusive gas sampler consisted of a PVC pipe with a 10 cm long sampling section centred at each sampling depth. Sampling sections consisted of ~2 m of silicon tubing (3 mm i.d., 5 mm o.d.) wrapped around the PVC pipe and kept in place by PVC-spacers at the top and bottom of each interval. Silicone tubes were sealed at one end whereas the other end was connected to polyurethane tubing (1.8 mm i.d.) that ran back up inside the PVC tube to reach the peat surface where it was sealed with a three-way stopcock. Silicone tubing has been shown to be permeable to gases such as CO2 and CH4 within a number of hours, while remaining impermeable to water, making it suitable for sampling of dissolved soil gases (Kammann et al., 2001)."*

L272-280: It is better practice to first present he equation, the define and give values for all variables. Here, there is a mix of information given before and after the equation. I recommend changing to: "The rates of CH4 and CO2 land-atmosphere fluxes (F) were calculated following:
F = S*(PV/RTA) (1)
Where S is the slope of the linear regression fitted to the gas concentration measurements over time inside the flux chamber (units). P is the atmospheric pressure (0.96 atm), . . .

*We have changed this text to*
*"The rates of CH4 and CO2 land-atmosphere fluxes (Flux) were calculated using the change in gas concentration over time inside the chamber (linear regression), the ideal gas law*

*following, average air temperature inside the chamber during the measurement, and a constant atmospheric pressure value of 0.96 atm in Eq. (1)::*
*Flux=slope (P.V)/(R.T.A)*
  *(1)*
*where slope is the linear rate of change of gas concentration (μmol mol-1 second-1) over the measurement period inside the chamber; P is an atmospheric pressure (atm) constant of 0.96 atm; V is chamber volume (L); R is the universal gas constant (L atm K-1 mol-1); T is the average temperature (K) inside the chamber during the measurement; and A is the chamber basal area (m2)."*

L 285-286: You should write this out using the equation tool.

*We have added the equation (now L416)*

L 302-303: It would sound better to say: "We measured the d13C values of gas samples from both the flux chambers and atmospheric background."

*We have changed this text to*
*"We measured the δ13C values of gas samples from both the chamber fluxes and atmospheric background"*

Please use proper notation for stable carbon isotope values (d13C), rather than saying "13C signatures"

*We have changed "$^{13}C$ isotopic signatures" to "$δ^{13}C$" throughout the text.*

L 307: "measured", not "quantified"

**We have removed "quantified".**

L302-311: This paragraph could be written more clearly and concisely. Use of passive voice here makes it difficult to read.

*We have edited this text (457-460) to improve clarity and now it reads*
*"We measured the δ13C values of gas samples from both the chamber fluxes and atmospheric background. To assess whether the gas concentration of each sample fit within the measurement range required for δ13C analysis we measured CO2 and CH4 concentrations using 1 – 3 mL from each vial. Following these concentration measurements, the remaining sample (17 – 19 ml) was diluted with nitrogen gas to a final volume of 20 mL and injected into a Small Sample Introduction Module (SSIM, Picarro, California, USA) system to measure δ13C signatures. The δ13C-CO2 and δ13C-CH4 signature was measured in-line with a cavity ring-down spectrometer (G2201-L, Picarro, California, USA) that had been calibrated using certified standards."*

L315: should this be "$(1/[CH_4])$" to denote that it is the concentration of CH4?

*We have changed this to (l/[CH₄]) as recommended (L 469):*

**"against the inverse of CH4 gas concentrations (1/[CH4])…"**

L321: Use "collected" rather than "taken"

*We have changed this "**Dissolved gas samples were collected**…" (L500)*

L 329: "concentration measurements" rather than "concentrations"

*We have changed this as suggested (L 506):*
**"CH4 dissolved gas concentration measurements were made by injecting 1 – 3 mL of gas into a gas chromatograph with an FID…"**

L 330-331: Again, d13C values, rather than "13C signatures"

*We have addressed all mention of 13C signatures, as requested.*

L321-340: Again, needs to be written more clearly.

*This section has been edited to improve clarity (L500-514):*

**"Dissolved gas samples were collected using diffusive equilibration gas sampling devices. Samples were taken from the following 15 depths: every 10 cm down to 95 cm starting at 5 – 15 cm, and then at 115 cm, 140 cm, 165 cm, 195 cm, and 245 cm. Once a month from May – September 2018 a ~7 mL gas sample was drawn from each depth using a 10 mL plastic syringe. These gas samples were immediately injected into a 10 mL sealed glass-vial that had been flushed with nitrogen gas prior to sealing , and then were stored at 4 °C until analysis. A total of 214 $CO_2$ and 211 $CH_4$ dissolved gas concentration measurements were made by injecting 1 – 3 mL of gas into a gas chromatograph with an FID and $CO_2$ methanizer (8610C Gas Chromatograph, SRI Instruments, California, USA). We measured $\delta^{13}C$-$CO_2$ and $\delta^{13}C$-$CH_4$ signatures using the previously mentioned cavity ringdown spectrometer and SSIM system. As with surface chamber gas samples, dissolved gas samples were diluted with $N_2$ to 20 ml. However, dissolved gas concentrations were considerably higher than gas concentrations found in the surface chambers, and some were well above the optimal concentration range required for accurate $\delta^{13}C$ analysis for the SSIM system even after dilution. To fit within the optimal operational $CH_4$ concentration range of the SSIM system used…"**

L335: "concentration range"

*We have changed L335 (now L513) as specified:*

*"*…**optimal concentration range required for accurate $\delta^{13}C$ analysis for the SSIM system…"**

L337: "measurable range of the system"

*We have edited this line as suggested (L514):*
**"…To fit within measurement range of the system, further…"**

L349: "Focusing"

*We have applied this change to L349 (now L640):*
**"Focusing on peat samples, microbial community composition in the active layer…"**

L372: please write out what PVDF stands for( Polyvinylidene difluoride)

*We have specified what PVDF stands for: "… Polyvinylidene difluoride (PVDF) membrane sterivex filters (MilliporeSigma)." (L665)*

L424: "We performed these tests to assess whether thaw stage…"

*We have changed this line accordingly (L734):*

**"We performed these tests to assess whether thaw stage (young bog or mature bog) influenced greenhouse gas fluxes…"**

L430: "Similarly, we tested for significant differences between the depth profiles in the young versus old bogs with respect to dissolved CH4 and CO2 concentrations, d13C values, and alpha$_c$ values."

*We have changed these lines as specified (L747):*
**"Similarly, we tested for significant differences between the young and mature bog depth profiles with respect to dissolved CH4 and CO2 concentrations, δ13C-CH4 and δ13C-CO2 values, αc values, and pore water chemistry. In these models, sampling month and peatland stage were defined as fixed effects while sample depth was defined as a random effect."**

L412: Better subtitle would be "Statistical analyses"

*We have changed this subtitle to "Statistical Analyses", as suggested (L722):*

**2.1 "Statistical analyses"**

L434: do you need to mention the instrument again? The illumine miseq is already mentioned in the methods section

*Good point- we have removed the additional mention of the Illumina Miseq (L754):*

**"Following microbial 16S rRNA gene sequencing, sample reads were."**

L448: There are other key studies that should have been used in the comparison

*We provide additional references, Kendall & Boone (2006) and Zhang et al., (2020) in the revised manuscript. In case further studies exist that we are not yet aware of, we would be glad to include them and kindly ask the reviewer to provide more suggestions here.*

*L769-770: "…determined by comparing our findings with the literature (Berghuis et al., 2019; Stams et al, 2019; Kendall & Boone, 2006; Zhang et al., 2020)."*

*References:*
- *Kendall MM, Boone DR. Cultivation of methanogens from shallow marine sediments at Hydrate Ridge, Oregon. Archaea. 2006 Aug;2(1):31-8. doi: 10.1155/2006/710190. PMID: 16877319; PMCID: PMC2685590.*
- *Zhang, CJ., Pan, J., Liu, Y. et al. Genomic and transcriptomic insights into methanogenesis potential of novel methanogens from mangrove sediments. Microbiome 8, 94 (2020). https://doi.org/10.1186/s40168-020-00876-z*

L492-497: These are not statistically significant differences, so you cannot say that measurements were higher in the mature bog than the young bog. These parameters are statistically identical between the two bogs.

*Statistically, pH and DOC are different (ANOVA; p < 0.05). We have more clearly outlined that these are statistically different and have provided the test results. For the rest of the results that are not significantly different, we have more clearly stated this and also have provided the test results. However, we do still state which averages are higher, particularly for SUVA and TDN, which have large standard deviations associated with them. While statistically not different, the differences in SUVA and TDN between stages of thaw may be sufficient to impact the microbial community structure (Bradley et al 2017). Nevertheless, we agree that we must point out more clearly where we found significant differences and where there were only insignificant trends or tendencies, and thus show our edited changes in L839-850:*

**"Across all depths and sampling occasions, average pH was higher (ANOVA: F (1, 77) = 35.2, P < 0.001) in the young bog than in the mature bog at 4.1 ± 0.2 and 3.9 ± 0.2 respectively. In contrast, DOC at 69.2 ± 18.4 and 53.8 ± 5.4 mg C L-1 (ANOVA: F (1, 82) = 38.7, P < 0.001) and total dissolved nitrogen at 1.5 ± 1.4 and 0.9 ± 0.1 mg L-1 (ANOVA: F (1, 82) = 12.8, P < 0.01) were higher in the mature bog than in the young bog, respectively. Average SUVA values were higher (ANOVA: F (1, 82) = 103.5, P < 0.001) in the young bog (3.2 ± 0.4 L mg C-1 m-1) compared to the mature bog (2.6 ± 0.4 L mg C-1 m-1), indicating DOM with a greater aromatic content in the young bog. However, average spectral slope (S250 – 465) values were also greater (ANOVA: F (1, 81) = 6.9, P < 0.05) in the young bog (-0.016 ± 0.002 nm-1) compared to the mature bog (-0.017 ± 0.003 nm-1), indicating lower molecular weight and decreasing aromaticity."**

*References*

- *Bradley JA, Anesio AM and Arndt S (2017) Microbial and Biogeochemical Dynamics in Glacier Forefields Are Sensitive to Century-Scale Climate and Anthropogenic Change. Front. Earth Sci. 5:26. doi: 10.3389/feart.2017.00026*

L503: "below the water table", rather than "under"

*We have changed L503 accordingly (now L857):*

**"Dissolved CH$_4$ increased with depth below the water table…"**

L504, 505: "Dissolved CH4 concentrations", rather than "concentrations of CH4"

*We have changed L857 & L858 accordingly:*

**"Dissolved CH$_4$ concentrations in the young bog increased with depth, from 19 μmol L$^{-1}$ at 5 cm depth, to a peak of 5,400 μmol L$^{-1}$ at 195 cm. Dissolved CH$_4$ concentrations in the mature bog remained…"**

L507: What was the peak concentration in the mature bog? It's difficult to compare the concentrations between the two bogs because you report different types of measurements.

*Peak concentration in the mature bog was 6,800 μmol L$^{-1}$. We have added this to the text (L862).*

L509-510: It's not clear that the mature bog had higher CO2 concentrations, so this is confusing.

*Figure 2 shows that the mature bog has higher CO$_2$ concentrations, but we have edited L864-866 to demonstrate this more clearly in the text:*

**"Again, the mature bog had overall higher concentrations, with mean average values ranging from 340 – 1,295 μmol L$^{-1}$ and peaking at 1,500 μmol L$^{-1}$ at 85 cm. Whereas the young bog average values ranged from 113 – 960 μmol L$^{-1}$ and peaked at 1,200 μmol L$^{-1}$ at 95 cm (Figure 2b)."**

L517: Again, use the delta notation rather than writing "13C isotopic signatures."

*We have addressed this throughout the text, as suggested.*

L527: "Distinct" is a strong word to use here, given that there are many d13C measurements with overlapping uncertainty in both CO2 and CH4 d13C profiles.

*The depth profiles of both $\delta^{13}$C-CO$_2$ and $\delta^{13}$C-CH$_4$ are statistically different from one another both above (L533; ANOVA (F (1, 92) = 17.25, P < 0.001) and below the water table (L536); ANOVA F (1, 99) = 5.33, P < 0.05) thus we consider distinct to be an appropriate word (now L919) and have maintained its use in the text.*

L533: It is not immediately apparent that "F" is the ANOVA F-test statistic. In the methods section, please introduce that you will use the F-test static to compare the two profiles

statistically. It's also not clear what the (1, 99) and (1,92) subscripts indicate.

*We have more clearly stated in the methods section that we are using ANOVA and have consistently reported the F statistic throughout when reporting the results from our ANOVA, as suggested. We have also added "ANOVA" to the results section when reporting the F statistic. (L919-929):*

 ***"The young bog and mature bog had distinct profiles of δ13C values for both CH4 and CO2 (Figure 2c, d). The young bog had no apparent trend with depth for both δ13C-CH4 (ANOVA; F (14, 45) = 1.75, P = 0.08) and δ13C-CO2 (ANOVA; F (14, 46) = 1.79, P = 0.07), averaging -62.4 ± 7.0 ‰ and -6.8 ± 1.6 ‰, respectively (Figure 2c, d). In the mature bog we observed significant depth trends for both δ13C-CH4 (ANOVA: F (14, 43) = 3.19, P < 0. 01) and δ13C-CO2 (ANOVA: F (14, 49) = 6.22, P < 0.001). These significant depth trends are due to isotopically heavy δ13C-CH4 and light δ13C-CO2 above the water table, which suggests an influence from CH4 oxidation. When comparing δ13C depth profiles between the thermokarst bogs we focused on those values taken from under the water table to avoid the effect of CH4 oxidation observed above the water table in the mature bog. Under the water table, δ13C-CH4 values in the mature bog were significantly lighter (ANOVA: F (1, 64) = 18.72, P < 0.001) compared to the young bog at an average of -68.7 ± 5.0 ‰ and -62.4 ± 7.0 ‰, respectively. Conversely, the mature bog had isotopically heavier δ13C-CO2 than the young bog below the water table (ANOVA: F (1, 71) = 13.86, P < 0.001)…"***

L553-554: Each d13C measurements represents a mixture of two sources (acetoclastic and hydrogenotrophic). It would be much more informative to make a two end-member mixing model and estimate the relative contributions of the two methanogenesis pathways to each measurements. Using these estimates would allow for more quantitative comparison between methanogenesis pathways between the young and mature bogs.

*We agree that an end member mixing model would be an interesting way to answer the questions we are addressing in the manuscript. Unfortunately, we do not have the correct dataset to do so. To perform an end-member mixing model we would need the specific fractionation factors associated with acetoclastic and hydrogenotrophic methanogenesis at our site. These fractionation factors vary considerably across sites (Conrad, 2005); thus we cannot use values from the literature. To determine these fractionation factors, we would need to either perform an in situ labelling experiment or have a high resolution of $\delta^{13}C$ data of organic matter at each depth from where we have dissolved $CO_2$ and $CH_4$ concentrations. As we do not have either of these an end member mixing model approach is unfortunately not suitable to our study.*

*References:*
- *Conrad, Quantification of methanogenic pathways using stable carbon isotopic signatures: a review and a proposal, Organic Geochemistry, Volume 36, Issue 5, 2005, Pages 739-752, ISSN 0146-6380, https://doi.org/10.1016/j.orggeochem.2004.09.006.*

L567-568: Rather than saying "maximum ecosystem respiration in the mature bog was found…", better to say "Ecosystem respiration rates were elevated from June to August, and

decreased in September."

*We have modified this line to reflect the changes requested by the reviewer (L1014-1015):*

**"Ecosystem respiration rates in the mature bog were elevated from June to August (monthly averages between 2.1 and 2.6 g CO2 m-2 day-1), and decreased in September (0.8 g CO2 m-2 day-1)"**

L570-577: There are some grammatical mistakes and misuse of punctuation that make this difficult to read.

*L570-L577 (now L1017-1022) have been corrected for grammar and punctation:*

**"This was a result of both higher CH4 emissions and lower ecosystem respiration (Figure S3) in the young bog. The δ13C-CH4 signature of CH4 emissions (intercept values from Keeling plots), in the young bog were significantly greater than those observed in the mature bog (Figure 3c; ANOVA: F (1, 4) = 20.67, P < 0.05). The average δ13C-CH4 signature of CH4 emissions in the young bog (n = 4) was -66.5 ± 1.4‰ (95% CI) and 78.5 ± 5.6‰ (95% CI; Figure 3c) in the mature bog emissions (n = 4)."**

L674-678: If these variables only explain 18.4 and 4.3% of methanogenic community structure variation, then what other variables are important here? It seems like the analysis needs to go farther/data are inconclusive.

*It is important to note that these variables are significant in influencing methanogenic community structure as determined by our backward stepping model, As such these two variables (distance to water table and thaw stage) were only mentioned because these were most important and relevant in determining methanogenic community structure, utilizing our backward stepping model. The analysis would not be as statistically robust if we were to include other, non-significant variables that were used as part of this model (i.e., DOC, temperature, enzymatic activity estimate depth, and etc, as described in L800 of the methods). We have added a line to mention that the remaining variation may be constrained by these non-significant parameters, and likely others not measured here in L1161-1166. (Boon et al., 2014):*

**"Although these were the only two parameters that were identified as significant variables impacting microbial community structure when using a backward stepping model, it should be noted that there may be more variation in the community that could be significantly explained by unconstrained variation brought about from plant-microbe and microbe-microbe interactions (Boon et al., 2014) that our experimental design does not take into account."**

*Reference:*
- *Boon, E., Meehan, C. J., Whidden, C., Wong, D. H., Langille, M. G., & Beiko, R. G. (2014). Interactions in the microbiome: communities of organisms and communities of genes. FEMS microbiology reviews, 38(1), 90–118. https://doi.org/10.1111/1574-6976.12035*

L709-710: Would be worth mentioning that 14C measurements of CO2 and CH4 would help answer the question of whether the emissions are derived from decomposition of fresh, labile DOM or old, previously frozen peat.

*We have added in text citing previous 14C-CO2 work at similar thermokarst bog sites in western Canada have also found little to no evidence of aged carbon contributing to CH4 emissions at the surface (Cooper et al 2017) which we have also included in the text (L1223-11225):*

**"However, previous work in the discontinuous permafrost region in the Interior Plains of western Canada has found a limited contribution of previously frozen organic matter contributing to surface CH4 emissions in thermokarst bogs (Cooper et al., 2017)."**

*References*
- *Cooper, M. D. A., Estop-Aragonés, C., Fisher, J. P., Thierry, A., Garnett, M. H., Charman, D. J., et al. (2017). Limited contribution of permafrost carbon to methane release from thawing peatlands. Nature Climate Change, 7(7), 507–511. https://doi.org/10.1038/nclimate3328*

L718: up "to" the surficial…

*We have changed L718 (now L1234) accordingly.*

L720: "drier", rather than "relatively drier." Using the word "drier" already implies a Comparison

*We have changed L720 (L1236) accordingly, to avoid repetition.*

L738: "also been observed"

*We have changed L738 (L1260) accordingly:*

**"…as has also been observed in other permafrost ecosystems (Frey et al., 2016; Monteux et al., 2018)."**

Conclusions:

L853-854: The authors cite a "greater availability of plant leachates" but the DOC and DN data suggest no statistically significant differences in plant leachates between the young and mature bog. This is not a good explanation for the observed shifts in methanogenic communities.

*We have better differentiated how we use our data as well as that from the literature to discuss our results. In this study, we only measured a small suite of DOM parameters and thus, rely upon previous literature at this site and similar locations to provide further support for our interpretations. We have two distinct vegetation communities and surface inundation conditions. Previously, both of these factors have been shown to influence the quality and quantity of plant derived DOM, which in turn has been shown to significantly influence microbial community composition (Laiho, 2003; 2006; Robroek., et al 2016; Bragazza et al., 2015; Ernakovich et al., 2017; Burd et al., 2020). Thus, we believe this to be a logical and appropriate explanation for the differences we observe within the microbial community. We have made these connections clearer in the text (L1442-1448):*

***"The influence of this pathway was apparent at depth throughout the peat profile. With succession following thaw towards a mature thermokarst bog, a shift in water table position and vegetation composition seems to reduce the role of acetoclastic methanogenesis pathway. Previous work at this site (Heffernan et al., 2020) and other thermokarst peatlands in the discontinuous permafrost zone of boreal western Canada (Burd et al., 2020) have indicated that the vegetation community found in the initial decades following permafrost thaw is associated with increased potential enzymatic degradation and biodegradability of organic matter compared to that found in the mature bog."***

*References*
- *Laiho, R. (2006). Decomposition in peatlands: Reconciling seemingly contrasting results on the impacts of lowered water levels. Soil Biology and Biochemistry, 38(8), 2011–2024. https://doi.org/10.1016/j.soilbio.2006.02.017*
- *Laiho, R., Vasander, H., Penttilä, T., & Laine, J. (2003). Dynamics of plant-mediated organic matter and nutrient cycling following water-level drawdown in boreal peatlands. Global Biogeochemical Cycles, 17(2). https://doi.org/10.1029/2002g b002015*
- *Robroek, B. J. M., Albrecht, R. J. H., Hamard, S., Pulgarin, A., Bragazza, L., Buttler, A., & Jassey, V. E. (2016). Peatland vascular plant functional types affect dissolved organic matter chemistry. Plant and Soil, 407(1-2), 135–143. https://doi.org/10.1007/s1110 4-015-2710-3*
- *Bragazza, L., Bardgett, R. D., Mitchell, E. A. D., & Buttler, A. (2015). Linking soil microbial communities to vascular plant abundance along a climate gradient. New Phytologist, 205(3), 1175–1182. https://doi.org/10.1111/nph.13116*
- *Ernakovich, J. G., Lynch, L. M., Brewer, P. E., Calderon, F. J., & Wallenstein, M. D. (2017). Redox and temperature-sensitive changes in microbial communities and soil chemistry dictate greenhouse gas loss from thawed permafrost. Biogeochemistry, 134(1–2), 183–200. https://doi.org/10.1007/s1053 3-017-0354-5*
- *Burd, K., Estop-Aragonés, C., Tank, S. E., & Olefeldt, D. (2020). Lability of dissolved organic carbon from boreal peatlands: Interactions between permafrost thaw, wildfire, and season. Canadian Journal of Soil Science, 13(February), 503–515. https://doi.org/10.1139/cjss-2019-0154*

L861: lower temperatures in the mature bog? I assume that the water in the young bog helps to absorb more heat, but this should be made clear in the manuscript.

*We have soil temperature measurements from the site (Figure S1) to show this and provide results from L830-838) on soil temperatures. We have also highlighted how the thermal properties of the drier surface-peat reduce the temperature in the mature bog (L1387):*

**"Lower soil temperatures, a vegetation community associated with reduced substrate availability, the dominance of hydrogenotrophic methanogenesis throughout the peat profile, and a deeper water table position all contribute to the lower CH4 production and higher CH4 oxidation observed in the mature bog."**

In the conclusions, it would be useful to mention the relative contributions of acetoclastic and hydrogenotrophic methanogenesis to methane emissions in the two bogs.

*We have striven to address this in our conclusions as mentioned in our response to the reviewer's comment for lines 553-554.*

Figures:

Figure 1: would be helpful to label photos d and e with "mature" and "young"

*We have added these labels to enable easier interpretation of the plots.*

Figure 2: You need a legend showing which colors represent young vs. mature bog. It is better practice to not rely on explanation in the figure caption, but to give the reader essential information in the figure itself. I would suggest using a different shape for the data points for one of the bogs too.

*We have added a legend to this figure in panel Figure 2f.*

Figure 2(a): It would be helpful to write "CH4 concentrations" and "CO2 concentrations" or use concentration notation ([CO2], [CH4]). I also don't understand the arrows. Why are they pointing to specific points? Don't these represent the top of the permafrost/bottom of active layer? It would make more sense to have additional horizontal lines rather than arrows, unless that looks too busy with the water table levels.

*We have added the changes to the description for Figure 2a as suggested: "Dissolved $CH_4$ and dissolved $CO_2$" The unit ($\mu mol\ L^{-1}$) further clarifies that these are concentrations.*

*The arrows are explained in the figure caption on L915-917; they indicate the thaw transition depth at both sites. We tried using horizontal lines to indicate this, however it made the figure too crowded and messy. We do not focus too much on this transition within our results and discussion, but rather, the water table position as it is more important to our study. Thus, the water table position is prominent in the figure, but the thaw transition is not.*

Figure 2(f): To help guide the reader, I suggest using background shading and labels to identify the regions of acetoclastic and hydrogenotrophic methanogesis

*We appreciate this suggestion, and had previously tried this, but ultimately, the shading made the figure too busy and unclear. Because the range of α$_C$ values for acetoclastic and hydrogenotrophic methanogenesis overlap, the shading becomes difficult. We have instead provided a label for each line within the figure and the range associated with each pathway in the text. This is in the same format as Hornibrook et al (1997, 2000).*

*References*

- *Hornibrook, E. R. C., Longstaffe, F. J., & Fyfe, W. S. (1997). Spatial distribution of microbial methane production pathways in temperate zone wetland soils: Stable carbon and hydrogen isotope evidence. Geochimica et Cosmochimica Acta, 61(4), 745–753. https://doi.org/https://doi.org/10.1016/S0016-7037(96)00368-7*
- *Hornibrook, E. R. C., Longstaffe, F. J., & Fyfe, W. S. (2000). Evolution of stable carbon isotope compositions for methane and carbon dioxide in freshwater wetlands and other anaerobic environments. Geochimica et Cosmochimica Acta, 64(6). https://doi.org/10.1016/S0016-7037(99)00321-X*

Figure 4: I like this figure, but it could be arranged differently to take up less space on the page – the circles and triangle legend could go below the color bars, or the entire legend could go on top of the plot and the text can wrap around the right side of the figure.

*We have edited this figure, as suggested.*

Figure 6: In the text, you say that you assess only thaw stage and distance to water table, but in the caption you say that you explore both biotic and abiotic factors. What else was included in this analysis that is not shown in the plot that seemingly explains more of the methanogenic community variation?

*We describe the other factors that went into this analysis in the methods section at L800-801of the methods. We have referenced this section in the figure caption.*

References cited above:

H j, L., Olsen, R. & Torsvik, V. Effects of temperature on the diversity and community structure of known methanogenic groups and other archaea in high Arctic peat. ISME J 2, 37–48 (2008). https://doi.org/10.1038/ismej.2007.84
Throckmorton, H.M., Heikoop, J.M., Newman, B.D., Altmann, G.L., Conrad, M.S., Muss, J.D., Perkins, G.B., Smith, L.J., Torn, M.S., Wullschleger, S.D. and Wilson, C.J., 2015. Pathways and transformations of dissolved methane and dissolved inorganic carbon in Arctic tundra watersheds: Evidence from analysis of stable isotopes. Global Biogeochemical Cycles, 29(11), pp.1893-1910.
Data show a temporal shift in methanogenesis pathways, from acetoclastic in July to hydrogenotrophic in September.
Hultman, J., Waldrop, M., Mackelprang, R. et al. Multi-omics of permafrost, active layer and thermokarst bog soil microbiomes. Nature 521, 208–212 (2015).

https://doi.org/10.1038/nature14238
Active layer communities expressed genes and proteins involved in obtaining energy and nutrients from a diversity of aerobic and anaerobic processes and were equipped with functions for survival under freeze–thaw conditions. The bog represented a different scenario with a very high measured rate of methanogenesis and correspondingly high relative abundances of genes, transcripts and proteins involved in methanogenesis, thus demonstrating the potential linkage between molecular data and ecosystem level process rates

Response to reviewer 2 (RC2)

Review on bg-2021-337
Anonymous Referee #2

Referee comment on "High peatland methane emissions following permafrost thaw: enhanced acetoclastic methanogenesis during early successional stages" by Liam Heffernan et al., Biogeosciences Discuss., https://doi.org/10.5194/bg-2021-337-RC2, 2022

The manuscript by Heffernan et al. looks at the effect of permafrost thaw on methane emission, pathways of methanogenesis and microbial community. They compare depth profiles of young and mature thermokarst bogs and the uncollapsed plateau. Based on isotope values of methane and methanogenic archaeal community composition it is concluded that acetoclastic methanogenesis is more important in the young bog with higher methane emission than in the mature bog.
The major strength of the manuscript of the manuscript is the multifaceted approach: CH4 and CO2 emissions during the whole growing season, isotope values of the gases, depth profiles of dissolved gases, depth profiles microbial communities in peat and porewater at two time points. These all help build a thorough picture of the large methane emission during thermokast formation where changes through the growing season and the peat profile are taken into account, together with the microbial successional dynamics. The manuscript is easy to read ang the figures are clear. I especially like Figure 1 on the experimental setup that shows well both the horizontal and vertical aspects of the sampling setup.

*Thank you very much for your comments and feedback!*

A potential weakness of the study is that the conclusions of the microbial community analysis focus on methanogens, but the analysis was carried out by primers that amplify both bacteria and archaea. This means that archaea and further methanogens form only a small fraction of the sequence reads. However, the read numbers and the proportion of archaea and methanogens in the dataset are reported well and suggest that there is on average around 900 methanogen reads per sample (I hope I got this right), which should be sufficient to cover methanogen diversity.

*Yes, we used universal primers, targeting both archaea and bacteria. We made this choice in order to enable exploration of both the bacterial and archaeal populations before narrowing our*

*focus on the methanogenic community for this manuscript. Because archaea are still adequately captured in our dataset, as the reviewer points out, on (average 1021 methanogen-related reads were captured per sample), we believe that our approach is sufficient for covering methanogen diversity. We have added this value to lines 718-720:*

**"We also note that the Greengenes database is still commonly used to explore methanogenic archaeal communities (Vanwonterghem et al., 2016, Lin et al., 2017, Carson et al., 2019). Furthermore, because, on average, 1021 methanogenic reads were captured per sample using Greengenes, we believe that our approach is sufficient for covering methanogen diversity."**

Major comments:

1. Based on Fig. S2, Methanosarcinales/Methanosarcinaceae/Methanosarcina were defined as acetoclastic methanogens Please clarify the basis of this definition. Methanosarcinales contains methanogens that can use acetate, H2+CO2 and methylated compounds. Even within genus Methanosarcina, not all species use acetate (Kendall & Boone 2006 https://doi.org/10.1007/0-387-30743-5_12). The family Methanotrichaceae consists of obligate acetoclastic methanogens, but based on Fig. S2 they were not detected?

*Yes, as the reviewer points out, members of the Methanosarcinales do indeed perform multiple kinds of methanogenesis. We labelled them here as "acetoclastic" since they were only methanogenic members that we detected that were associated with acetoclastic methanogenesis. We did not detect Methanotrichaceae in our samples, using either SILVA or Greengenes (see response below re: SILVA). However, we agree with the reviewer that our labelling of Methanosarcinales as solely acetoclastic is mis-leading. We have re-labelled these as acetoclastic / hydrogenotrophic in Fig S2.*

*Reference:*

*Kendall M.M., Boone D.R. (2006) The Order Methanosarcinales. In: Dworkin M., Falkow S., Rosenberg E., Schleifer KH., Stackebrandt E. (eds) The Prokaryotes. Springer, New York, NY. https://doi.org/10.1007/0-387-30743-5_12*

2. Do I understand correctly that the microbial analyses were based on one peat core per site per sampling month (so no replication within sampling month)? I understand that in such a multifaceted study it is not possible to cover everything perfectly, but how is it possible to test the effect of sampling month (L620-621) without replication?

*Yes, correct. We only had one peat core per type of peat (YB, MB, peat plateau) per sampling month as well as per depth. We combined all samples between months together (i.e., all samples from YB, MB and peat plateau in June vs all samples from YB, MB and peat plateau in September) for statistical analyses because, utilizing a PERMANOVA, we found that these samples were not statistically significant between sampling months. However, using this approach, we were unable to more robustly confirm if sampling month had a significant impact*

*on microbial community structure. We have added the caveat that we did not have replicate samples to test the robustness of this finding to lines 1397-1398 of the discussion:*

***"This may be a sampling design effect since our study spanned only two months (June and September), compounded by the fact that we did not have replicate samples to test the robustness of this finding."***

Specific comments:
L84-85 Please clarify how the statement that two-thirds of CH4 comes from acetoclastic methanogenesis applies to peatlands. As far as I understand, Conrad 1999 is a general prediction, and Kotsyurbenko et al. 2007 cites several references to say most of methane in peatlands and even 100% comes from hydrogenotrophic methanogenesis?

*Thank you for catching this! We have modified the text to now read as:*
***"Hydrogenotrophic methanogenesis is thought to be the main pathway of CH4 formation in northern peatlands (Hornibrook et al., 1997). However, the acetoclastic pathway can dominate in the upper layers of more minerotrophic, nutrient rich peatlands (Popp et al., 1999; Chasar et al., 2000) where there are sufficient levels of acetate (Ye et al., 2012)."***

L410-411 I think the Greengenes database hasn't been updated for a very long time? This might not be a big problem because methanogen nomenclature has not changed that much recently. However, I am still left wondering if using a newer reference database would have improved the taxonomic affiliations (for example by providing more detailed affiliations or affiliations to unidentified OTUs).

*The reviewer is correct, the Greengenes database hasn't been updated since May 2013, however we also used the SILVA database to assign taxonomy to our ASVs and found that both SILVA and Greengenes captured a similar number of archaea (total of 51187 methanogenic read counts attributed to SILVA vs 51141 methanogenic read counts attributed to Greengenes). Also, the taxonomic resolution between both databases was also similar, identifying the same kinds of phyla, families and genus, and methanogens (i.e. methanoregula, methanosarcinales, etc,..) Given, these similarities, and the fact that methanogen nomenclature has not changed significantly as the reviewer points out, we ultimately chose to use Greengenes because it was able to resolve more methanogenic families belonging to methanocelalles and Methanomassiliicoccaceae compared to SILVA. We also note that the Greengenes database is still commonly used to explore methanogenic archaeal communities (Vanwonterghem et al., 2016, Lin et al., 2017, Carson et al., 2019). We have added this information to lines 707-720 of the Methods section:*

***"Although Greengenes is not updated as frequently as the SILVA database, we chose to use Greengenes to classify our ASVs as we found that SILVA and Greengenes captured a similar number of archaea (total of 51187 methanogenic read counts attributed to SILVA versus 51141 methanogenic read counts attributed to Greengenes). The taxonomic resolution between both databases was also similar, identifying the same kinds of phyla, families and genus, and methanogens (i.e. methanoregula, methanosarcinales…) Given, these similarities, and the fact that methanogen nomenclature has not changed significantly over time, we***

*ultimately chose to use Greengenes because it was able to resolve more methanogenic families belonging to methanocelalles and Methanomassiliicoccaceae, compared to SILVA. We also note that the Greengenes database is still commonly used to explore methanogenic archaeal communities (Vanwonterghem et al., 2016, Lin et al., 2017, Carson et al., 2019). Furthermore, because, on average, 1021 methanogenic reads were captured per sample using Greengenes, we believe that our approach is sufficient for covering methanogen diversity."*

*References*

- *Lin, Y., Liu, D., Yuan, J., Ye, G,m Ding, W. (2017). Methanogenic community was stable in two contrasting freshwater marshes exposed to elevated atmospheric CO2. Front Microbiol.  https://doi.org/10.3389/fmicb.2017.00932*

- *Michael A Carson, Suzanna Bräuer, Nathan Basiliko, Enrichment of peat yields novel methanogens: approaches for obtaining uncultured organisms in the age of rapid sequencing, FEMS Microbiology Ecology, Volume 95, Issue 2, February 2019, fiz001, https://doi.org/10.1093/femsec/fiz001*

- *Vanwonterghem, I., Evans, P., Parks, D. et al. Methylotrophic methanogenesis discovered in the archaeal phylum Verstraetearchaeota. Nat Microbiol 1, 16170 (2016). https://doi.org/10.1038/nmicrobiol.2016.170*

L600, L606, L611: Are these PERMANOVA results or ANOSIM results? In the methods only ANOSIM is mentioned (L444), and L617 and L621 mentions ANOSIM instead of PERMANOVA. Were both ANOSIM and PERMANOVA used and why? PERMANOVA should be the more robust alternative (see vegan documentation). Please also give the R or R2 values for PERMANOVA/ANOSIM results in addition to p values to give the reader an idea on the magnitude of the difference.

*We used both ANOSIM and PERMANOVA as a method to test significance, since they are similar analyses (although one is more robust, as the reviewer points out). The fact that PERMANOVA was left out of the methods was an oversight. All statistical tests using ANOSIM have been converted to PERMANOVA, with the corresponding R2 reported. Furthermore, the results from the PERMANOVA matched those from the ANOSIM test, and so our conclusions do remain unchanged:*

*L1065: "(PERMANOVA, R2= 0.13, P < 0.05, Figure 4)."*
*L1071: "(plateau peat, young bog and mature bog; Figure 4; PERMANOVA, R2 =0.18, P < 0.05)"*
*L076: "Figure 4, Figure S2, c; PERMANOVA; R2 =0.16, P < 0.05)."*
*L1086: "(Figure 4, PERMANOVA, R2=0.4, P = 0.1)"*

*L1091: " (PERMANOVA; R2=0.02, P = 0.090)"*

L611 Figure S2b is cited here but Fig. S2 has no a or b panels?

*Thank you for catching this typo! We mistakenly referenced the wrong Supplementary figure. We have subsequently corrected this throughout the text:*

*i.e,: "Our results, and those of others (Euskirchen et al., 2014; Johnston et al., 2014), have shown that CH4 emissions exhibit seasonal variation (Figure S3a, c)"*

L620-621, L693 Microbial community diversity -> microbial community composition or microbial community structure (because 'diversity' often refers to alpha diversity).

*We have changed the corresponding lines to "microbial community structure" as suggested, to avoid referral towards alpha diversity:*

*L1428: "Such changes in microbial community structure thus impact…"*
*L1434: "…which the microbial community structure changes…"*

L718 Check missing letter in 'up t the'.

*Thank you for catching this typo! This text has been edited to "…despite similar peat stratigraphy up to the surficial vegetation…"*

Response to reviewer 3 (RC3)

Reviewer 3 Summary:

The objectives of this study were to assess the impacts of time following permafrost thaw stage on methane emissions and methanogenic community composition. To do this, the authors identified two bog sites with permafrost that thawed 30 and 200 years ago. Analyses conducted at these sites included (1) metagenomic assessments, (2) dissolved gas concentrations (CO2 and CH4), (3) surface emissions,(4) d13C signatures for both CH4 and CO2 (used to assess the relative contribution of acetoclastic methanogenesis to total methanogenesis).

Overall, this paper effectively approaches that goal.

I have two primary concerns:
(1) My main concern with this paper is centered around the use of isotope d13C signatures. The alpha value is an accepted method for discerning the relative contributions of acetoclastic vs. hydrogenotrophic methanogenesis. While the authors used alpha values for dissolved gas analysis, they limited their assessments of acetoclastic input to 13C signatures of methane (without concomitant 13C-CO2 signatures). Is the 13C-CH4 signature on its own a sufficient indicator of acetoclastic contribution? If so, please provide citations that indicate so.

*We agree that the presentation of the isotope data can be clearer and more consistent throughout. We did not initially calculate alpha values for our fluxes as a significant proportion of the $\delta^{13}C$-$CO_2$ signature will be heavily influenced by autotrophic respiration. In our view, this would significantly influence the alpha values and lead to a bias towards acetoclastic methanogenesis. However, in light of the reviewer's comments, in the results section and throughout the manuscript we will change how we present the isotope data. The new format we will follow will be to first present the alpha vale with the delta values in parentheses. We have however kept Figure 3c the same, showing results from the Keeling plot method, as this is a commonly used approach to assess the processes involved in determining the isotopic composition of atmospheric $CH_4$. (Keeling, 1958). During methanogenesis, fractionation by hydrogenotrophic and acetoclastic methanogens produces $CH_4$ with a $\delta^{13}C$-$CH_4$ of -110‰ to -60‰ and -65‰ to -50‰, respectively (Hornibrook et al., 1997, 2000).*

*Many studies have used these plots, along with the known range of $\delta^{13}C$-$CH_4$ signatures associated with the methanogenic pathways, to identify the source signature, some of these are listed below.*

*References*

- *Keeling, C. D. (1958). The concentration and isotopic abundances of atmospheric carbon dioxide in rural areas. Geochimica et Cosmochimica Acta, 13(4). https://doi.org/10.1016/0016-7037(58)90033-4*
- *Hornibrook, E. R. C., Longstaffe, F. J., & Fyfe, W. S. (1997). Spatial distribution of microbial methane production pathways in temperate zone wetland soils: Stable carbon and hydrogen isotope evidence. Geochimica et Cosmochimica Acta, 61(4), 745–753. https://doi.org/https://doi.org/10.1016/S0016-7037(96)00368-7*
- *Hornibrook, E. R. C., Longstaffe, F. J., & Fyfe, W. S. (2000). Evolution of stable carbon isotope compositions for methane and carbon dioxide in freshwater wetlands and other anaerobic environments. Geochimica et Cosmochimica Acta, 64(6). https://doi.org/10.1016/S0016- 7037(99)00321-X Studies using the keeling plot method to identify $CH_4$ source*

- *Fisher, R. E., et al. (2017), Measurement of the 13C isotopic signature of methane emissions from northern European wetlands, Global Biogeochem. Cycles, 31, 605– 623, doi:10.1002/2016GB005504*
- *Marushchak, M. E., Friborg, T., Biasi, C., Herbst, M., Johansson, T., Kiepe, I., Liimatainen, M., Lind, S. E., Martikainen, P. J., Virtanen, T., Soegaard, H., and Shurpali, N. J.: Methane dynamics in the subarctic tundra: combining stable isotope analyses, plot- and ecosystem-scale flux measurements, Biogeosciences, 13, 597–608, https://doi.org/10.5194/bg-13-597-2016, 2016.*

- *Santoni, G. W., Lee, B. H., Goodrich, J. P., Varner, R. K., Crill, P. M., McManus, J. B., Nelson, D. D., Zahniser, M. S., and Wofsy, S. C. (2012), Mass fluxes and isofluxes of methane (CH4) at a New Hampshire fen measured by a continuous wave quantum cascade laser spectrometer, J. Geophys. Res., 117, D10301, doi:10.1029/2011JD016960.*
- *McCalley, C., Woodcroft, B., Hodgkins, S. et al. Methane dynamics regulated by microbial community response to permafrost thaw. Nature 514, 478–481 (2014). https://doi.org/10.1038/nature13798*
- *S. Sriskantharajah, R. E. Fisher, D. Lowry, T. Aalto, J. Hatakka, M. Aurela, T. Laurila, A. Lohila, E. Kuitunen & E. G. Nisbet (2012) Stable carbon isotope signatures of methane from a Finnish subarctic wetland, Tellus B: Chemical and Physical Meteorology, 64:1, DOI: 10.3402/tellusb.v64i0.18818*

Furthermore—I was not clear on whether the apparent difference in alpha values between mature vs. young sites was indeed statistically significant. This comparison needs to be made explicit. If significant differences between sites can only be found at specific depth intervals, then that should also be stated explicitly.

*We agree, this has not been made clear in the text and have rectified it. The results section describing this (L919-974) now reads as:*

**"The young bog and mature bog had distinct profiles of δ13C values for both CH4 and CO2 (Figure 2c, d). The young bog had no apparent trend with depth for both δ13C-CH4 (ANOVA; F (14, 45) = 1.75, P = 0.08) and δ13C-CO2 (ANOVA; F (14, 46) = 1.79, P = 0.07), averaging -62.4 ± 7.0 ‰ and -6.8 ± 1.6 ‰, respectively (Figure 2c, d). In the mature bog we observed significant depth trends for both δ13C-CH4 (ANOVA: F (14, 43) = 3.19, P < 0. 01) and δ13C-CO2 (ANOVA: F (14, 49) = 6.22, P < 0.001). These significant depth trends are due to isotopically heavy δ13C-CH4 and light δ13C-CO2 above the water table, which suggests an influence from CH4 oxidation. When comparing δ13C depth profiles between the thermokarst bogs we focused on those values taken from under the water table to avoid the effect of CH4 oxidation observed above the water table in the mature bog. Under the water table, δ13C-CH4 values in the mature bog were significantly lighter (ANOVA: F (1, 64) = 18.72, P < 0.001) compared to the young bog at an average of -68.7 ± 5.0 ‰ and -62.4 ± 7.0 ‰, respectively. Conversely, the mature bog had isotopically heavier δ13C-CO2 than the young bog below the water table (ANOVA: F (1, 71) = 13.86, P < 0.001).**

**The apparent fractionation factor (αC) is a robust parameter to characterize the relative contribution of CH4 production pathways, with values of 1.040 – 1.060 indicating acetoclastic methanogenesis and 1.060 – 1.090 for hydrogenotrophic methanogenesis (Chanton et al., 2005). Similar to the gas δ13C depth-profiles, we found no clear trend with depth for αC values in the young bog (ANOVA; F (14, 44) =0.87, P = 0.59) with an average of 1.058 ± 0.012 and range of 1.018 – 1.079 (Figure 2e). In the mature bog, we found a clear depth trend in αC values (ANOVA: F (14, 43) = 5.71, P < 0.001). Similar to the δ13C depth profiles in the mature bog, this significant depth trend in αC is due to the influence of CH4 oxidation above the water table, with the lowest αC values being those from samples collected above the water table at 5, 15, and 25 cm. The average αC beneath the water table in the mature bog was 1.064 ± 0.017 and ranged from 1.015 – 1.094. When comparing αC values from beneath the water table between the young and mature we found that αC¬ values were significantly lower in the young bog and mature (ANOVA: F (1, 63) = 30.8, P < 0.001)."**

*Within our analysis, we control for depth as we are not interested in differences at specific depths, but rather how the depth profiles overall differ between the thawed sites. Direct comparison of specific depths between the two sites can be complicated and misleading as depths in the young and mature bog do not correspond to one another with regard to depth from the water table, age, time spent since thaw occurred, and peat composition. Thus, we are interested in how these entire depth profiles differ between thaw sites.*

Your goal was to examine the effects of thaw stage on methane fluxes/methanogenic community composition. It is difficult to wrap my head around this goal since thaw succession causes shifts in so many different environmental factors (soil temperature, thickness of the unsaturated peat column, availability of labile organics). This makes your results difficult to build off of/apply to other settings. Perhaps you could perform an ordinary least squares regression analysis (OLS) to try to tease apart the relative influence of these numerous factors on a dependent variable of interest (perhaps total methane emissions, or acetoclastic methane emissions).

*We agree that peatland succession following permafrost thaw presents a dynamic, complex landscape that may influence microbial community composition and its activity in a myriad of ways. Here, we present data from two areas that have thawed 30 and 200 years ago. Each site has a distinct water table position, vegetation community, soil temperatures, and volume of peat accumulated at the surface following thaw. Each site does however have identical histories in the peat layers that accumulated prior to permafrost thaw (Heffernan et al., 2020). The objective of this study was to address how the combined effect of the ecological conditions (exposure of previously frozen peat, water table position, vegetation community, soil temperatures) found in the decades following thaw (young bog) and those found centuries following thaw (mature bog) influences the soil methanogen community, its activity, and the resulting $CH_4$ fluxes to the atmosphere. We found that no single factor drives differences, but rather it is the overall ecological conditions and interactions between these and microbial community members that influences the microbial community structure, its activity, pathways of methanogenesis, and $CH_4$ surface fluxes.*

*We explored potential relationships between environmental factors and methanogen community composition using a distance-based redundancy analysis (RDA; Figure 6). Included in our RDA was an initial backward stepwise regression (L799) to determine what environmental factors were significantly influencing the methanogenic community and should be included in our RDA. These included dissolved concentrations of $CO_2$, $CH_4$, DOC, temperature, enzymatic activity estimate, thaw stage, depth, and distance to water table. This stepwise regression serves a similar purpose to the prosed ordinary least squares and allows us use redundancy analysis once significant variables have been identified. While we could use the environmental data at each site (water table depth, soil temperatures, time since thaw) to model our total $CH_4$ emissions, and in doing so determine the main drivers of our $CH_4$ emissions over a growing season, this was not the objective of this study. This study aimed to relate the methanogen community to surface $CH_4$ fluxes, and to comment on how long elevated $CH_4$ emissions may persist following thaw. The 2018 growing season data presented in this study is being prepared, along with multiple other years, in a separate study to achieve this.*

*References*
*• Heffernan, L., Estop-Aragonés, C., Knorr, K.-H., Talbot, J., & Olefeldt, D. (2020). Long-term impacts of permafrost thaw on carbon storage in peatlands: deep losses offset by surficial accumulation. Journal of Geophysical Research: Biogeosciences, 2011(2865), e2019JG005501. https://doi.org/10.1029/2019JG005501*

Specific Questions/Recommendations

(1) I am unclear on what is mean by the term "ecological" in the context of this manuscript (e.g. L27-28). I get the impression that it references vegetation primarily. I am unsure about that, however, because "ecological" could also be used to describe microbial community composition. Please clarify.

*We have changed the word ecological to environmental throughout the manuscript when we are discussing the distinct environmental conditions found at each thaw stage. We have also added our definition of environmental conditions in the introduction on L134-136 which reads*

***"Thermokarst formation has resulted in distinct environmental conditions at each stage along this thaw gradient. We herein define these distinct environmental conditions as water table position and surface wetness, soil temperatures, and vegetation community."***

Fig 1: I'd recommend explicitly stating how far apart the mature and young bog sites are from one another in panel f (or the caption). You list only one GPS coordinate from the whole site in the figure caption.

*We have added the following text to L282 to indicate the distance*

*"The mature bog is ~10 – 15 m from the young bog"*

L158-159: How did you determine that this complex is representative? If the succeeding sentences are meant to serve as evidence for this claim, make that explicit.

*We have added the references listed below which describe the boreal peatland complexes in this area on L225. We have also added the Heffernan et al., 2020 reference to L229 which describes the developmental history of the study site, demonstrating how this site is representative of those:*

**"The peatland complexes in this area are a fine-scale mosaic of permafrost peat plateaus, and permafrost-free ponds, fens, and bogs (Zoltai, 1993; Bauer et al., 2003; Vitt et al., 2000; Pelletier et al., 2017), and they are similar to those found in the Hudson Bay Lowlands (Kuhry, 2008) and Alaska (Jones et al., 2017). The Lutose peatland complex is representative of the peatlands found in the discontinuous permafrost zone of the Interior Plains in western Canada (Heffernan et al., 2020)."**

*References*

- *Bauer, I. E., Gignac, L. D., & Vitt, D. H. (2003). Development of a peatland complex in boreal western Canada: Lateral site expansion and local variability in vegetation succession and long-term peat accumulation. Canadian Journal of Botany, 81(8), 833–847. https://doi.org/10.1139/b03-076*
- *Pelletier, N., Talbot, J., Olefeldt, D., Turetsky, M., Blodau, C., Sonnentag, O., & Quinton, W. L. (2017). Influence of Holocene permafrost aggradation and thaw on the paleoecology and carbon storage of a peatland complex in northwestern Canada. Holocene, 27(9), 1391–1405. https://doi.org/10.1177/0959683617693899*
- *Vitt, D. H., Halsey, L. A., Bauer, I. E., & Campbell, C. (2000). Spatial and temporal trends in carbon storage of peatlands of continental western Canada through the Holocene. Canadian Journal of Earth Sciences, 37(5), 683–693. https://doi.org/10.1139/e99-097*
- *Zoltai, S. C. (1993). Cyclic Development of Permafrost in the Peatlands of Northwestern Alberta, Canada. Arctic and Alpine Research, 25(3), 240. https://doi.org/10.2307/1551820*

L551-554: "Overall, the isotopic data indicates a general dominance of hydrogenotrophic methanogenesis in both sites, but a greater contribution of acetoclastic methanogenesis in the young bog relative to the mature bog." Was this difference statistically significant? I suggest adding a p-value after this statement.

*We have removed this statement as we do not have a specific statistical test we could perform to provide a p value. This statement was intended to cover the entire section of results discussing isotopic data and includes dissolved gas depth profiles of $\delta^{13}C$-$CH_4$, $\delta^{13}C$-$CO_2$, and $\alpha_C$ (981). This section has been edited to improve clarity:*

*"In the isotopic ratio cross-plot of δ13C-CH4 and δ13C-CO2 (Figure 2f), most of the young bog had αC values of between 1.055 – 1.065 (29 in total), with a greater number of samples (21) between αC =1.040 – 1.055, compared to the mature bog (15). In contrast, a greater proportion of the mature bog samples had αC > 1.065 (42 in the young bog and 52 in the mature bog). There was no clear depth trend in the αC values and no samples in this study had αC > 1.090. Several samples (13) from the young bog and mature bog had αC values of < 1.040, likely due CH4 oxidation (Knorr et al., 2009)."*

L 572-575: *"The δ13C-CH4 signature of CH4 emissions (intercept values from Keeling plots), in the young bog were significantly greater than those observed in the mature bog (Figure 3c; F (1, 4) = 20.67, P< 0.05)., suggesting a greater influence of acetoclastic CH4 production."*

Why is there no alpha value for the flux measurements? Please provide a source indicating that 13C-CH4 measurements alone (i.e. without concomitant 13C-CO2) are sufficient to discern the relative influence of acetoclastic methanogenesis on total methane production.

*Please see above where we have addressed the reviewer's 1st primary concern with the manuscript. Here we explain why we do not calculate an alpha value for fluxes and provide references for studies that have taken a similar approach in the past.*

L704-706: "Evidence of acetoclastic methanogens and CH4 produced via the acetoclastic metabolic pathway was found in the young bog both near the surface and at depths below the thaw transition (i.e., in peat that accumulated prior to permafrost thaw)."

I have two notes on this:
1) If the difference in alpha values was not significant in the subsurface (which I am not 100% clear on), this needs to be noted and discussed.

*Please see our response above to the reviewer's 2nd concern which addresses this. We have re-written our results section (839-855) to be more explicit in what significant trends and differences we found with depth and between the thermokarst bogs. We have changed how we discuss and present our isotope data to present both the alpha and delta values to help improve clarity. We have added text to section 4.2 to help clarify this. This edited paragraph (L1306-1308) now reads as*

*"Isotopic signatures (δ13C) of dissolved CO2 and CH4 and αC values in porewater and the of δ13C signature of CH4 emitted to the atmosphere provided further evidence of relatively elevated acetoclastic methanogenesis in the young bog stage. The general increase in δ13C-CO2 with depth observed at both sites (Figure 2d) indicates accumulation of isotopically heavier δ13C-CO2 which is likely explained by the preferential use of isotopically lighter δ13C-CO2 during hydrogenotrophic methanogenesis (Hornibrook et al., 2000). As a result, CH4 tends to become lighter with depth and this was particularly apparent in the mature bog (Figure 2c). This leads to the average αC values of 1.064 (δ13C-CH4; -68.7‰) in the mature bog, which were significantly higher than the 1.058 (δ13C-CH4; -62.4‰) observed in the young. Together, the δ13C-CH4 and δ13C-CO2 data and the resulting αC depth profiles suggest that the majority of CH4 is produced via the hydrogenotrophic methanogenic pathway, which supports the findings of the microbial community analysis (Figure 5). Our isotope data also suggests that a greater proportion of CH4 is produced via acetoclastic methanogenesis throughout the profile in the young bog compared to the mature bog (Figure 2c – f). This is evident from lower average αC values found in the young bog compared to the mature bog, and greater number of these young bog αC values falling between 1.040 – 1.065 which represents acetoclastic methanogenesis (Whiticar, 1999). These findings again agree with the relatively greater abundance of acetoclastic methanogens observed at that site (Figure 5)."*

2) See my comments regarding L572-575. Make sure your methods for discerning the acetoclastic influence on surface CH4 emissions is sound.

*Please see above where we have addressed this.*

L765-766: "*The presence of hydrophilic vegetation, particularly graminoids, in the saturated young bog provides the precursors for fermentation..*"

I am confused by this statement. "Precursors" could be interpreted as "reactants", which are primarily sugars. Sugars are ultimately delivered to porewater from other plants too (i.e. Sphagnum spp.). Are you saying that the sugars derived from gramminoids are more labile than those derived from Sphagnum? I would agree with this, but it is necessary to clarify.

*Yes, this is exactly what we mean, and we have added the references below (Ström et al., 2003; 2012) to help support this point. These show how graminoids enhance substrate quality and availability, leading to greater methanogenesis. We have also added text on L1268 and L1299-1300 where we discuss how graminoid vegetation is associated with anaerobic fermentation of monomers, providing the precursors to methanogenesis:*

**"The influence of vegetation communities associated with different thermokarst peatland stages on methanogenic community composition has previously been attributed to the role of plant derived DOM serving as the substrate for CH4 production (Liebner et al., 2015; McCalley et al., 2014). The presence of hydrophilic vegetation, particularly graminoids, in the saturated young bog provides the precursors for fermentation, yielding acetate (Liebner et al., 2015; Ström et al., 2003, 2012, 2015) and serving as the substrate for acetoclastic CH4 production."**

*We use precursors as a catchall term for all plant derived monomeric compounds formed following extracellular enzyme hydrolysis used in this fermentation step. Precursor is the most suitable term for these, as it is defined by Merriam-Webster dictionary, "a substance, cell, or cellular component from which another substance, cell, or cellular component is formed"*

*References*
*Ström et al., Presence of Eriophorum scheuchzeri enhances substrate availability and methane emission in an Arctic wetland, Soil Biology and Biochemistry, Volume 45, 2012, Pages 61-70, ISSN 0038-0717, https://doi.org/10.1016/j.soilbio.2011.09.005.*
*Ström, L., Ekberg, A., Mastepanov, M. and Røjle Christensen, T. (2003), The effect of vascular plants on carbon turnover and methane emissions from a tundra wetland. Global Change Biology, 9: 1185-1192. https://doi.org/10.1046/j.1365-2486.2003.00655.x*

L805-813: I find the thread of this paragraph hard to follow. Please make the connections between sentences clearer.

*These sentences have been restructured to better highlight the message we are trying to convey with them. This section attempts to summarise that there are multiple factors influencing the observed differences in the $\delta^{13}C$- $CH_4$ signature of $CH_4$ emissions between the young and mature bog. This has been edited to make this clearer and now reads:*

**"However, increased oxidation above the water table in the mature bog is likely not fully responsible for the observed differences in CH4 surface emissions and depth profiles between the young and**

*mature bog. Lower soil temperatures, a vegetation community associated with reduced substrate availability, the dominance of hydrogenotrophic methanogenesis throughout the peat profile, and a deeper water table position all contribute to the lower CH4 production and higher CH4 oxidation observed in the mature bog"* (L1387-1399).

Fig 2: Please add in a legend.

*We have added a legend to this figure in panel Figure 2f.*